# C-Evolve: Consensus-based Evolution for Prompt Groups

**Tiancheng Li**[1,2,†], **Yuhang Wang**[2,†], **Zhiyang Chen**[2,4],
**Zijun Wang**[1,2], **Liyuan Ma**[2,4], **Guo-Jun Qi**[2,3,4,*]
[1]Zhejiang University
[2]School of Engineering, Westlake University & MAPLE Lab
[3]Matterwave Intelligence
[4]Institute of Advanced Technology, Westlake Institute for Advanced Study
{litiancheng, wangyuhang, chenzhiyang, wangzijun, maliyuan}@westlake.edu.cn
guojunq@gmail.com

## Abstract

Prompt evolution algorithms offer a powerful paradigm for enhancing AI systems based on closed-source models, while few work explores whether aggregating results from multiple prompts to reach a *consensus* can further advance the system capability boundary. In this paper, we introduce *Consensus-Evolve* (C-Evolve), an evolutionary algorithm that discovers a group of prompts whose aggregated outputs after majority voting achieve optimal performance. More specifically, C-Evolve employs an island-based evolutionary algorithm to maintain population diversity, and prompts from distinct islands are selected to form groups to aggregate their outputs. The key difference from single individual evolution is a voting score, which evaluates each individual prompt's contribution within groups. We take this as the fitness score for evolution instead of individual performance. Consequently, C-Evolve is more likely to produce and maintain prompts with higher potential to form a high-performing group and eliminate low-performing ones, gradually improving the group performance after reaching consensus. Our method achieves state-of-the-art performance across a wide range of tasks, including both open-ended tasks like HotpotQA and closed-ended tasks like MATH. On Qwen3-8B, C-Evolve achieves 70.67% on HotpotQA and 43.88% on IFBench, which are 4.95% and 2.73% higher than GEPA, respectively. For GPT-4.1-mini, the accuracy on IFBench is further improved to 47.96% and reaches 95.33% in the MATH benchmark. These results demonstrate the C-Evolve's competitive performance.

## 1 Introduction

The advancement of Large Language Models (LLMs) has significantly accelerated progress in natural language processing (Liu et al., 2024; Thirunavukarasu et al., 2023). However, many state-of-the-art models, such as GPT-4.1 (OpenAI, 2025) and Claude (Anthropic, 2024), remain closed-source and are accessible only via API interfaces. This restricts researchers from performing parameter fine-tuning to adapt these models to specific downstream tasks. To leverage the capabilities of such black-box models, a new paradigm prompt-based optimization methods have emerged (Opsahl-Ong et al., 2024; Câmara et al., 2025; Agarwal et al., 2025; Fernando et al., 2023). By refining system prompts, these methods enhance model performance without updating internal weights.

Among these paradigms, evolutionary algorithms represent a promising direction (Agrawal et al., 2025; Novikov et al., 2025) without access to the weight gradients of black-box models. Indicated with a predefined fitness function, these methods iteratively refine a population of prompts and select the individual with the highest fitness as the globally optimal prompt. However, in complex

---

This project was initiated and supported by MAPLE Lab at Westlake University.

[†] Equal contribution.
[*] Corresponding author.

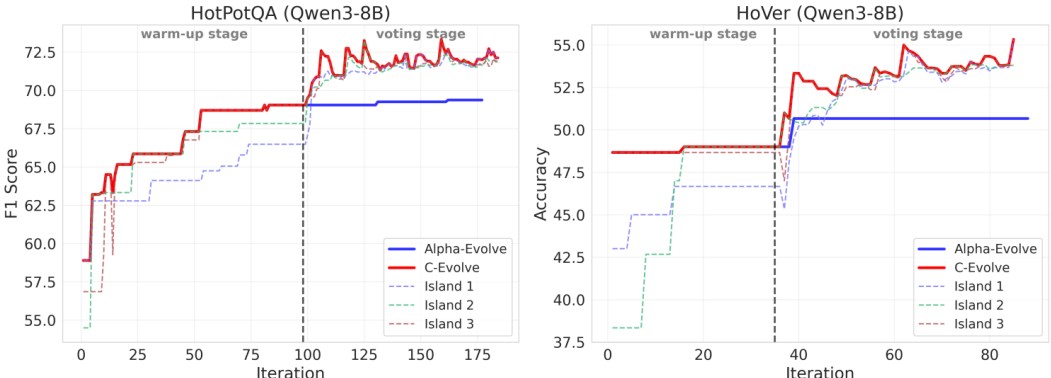

Figure 1: Comparison of our proposed C-Evolve and AlphaEvolve. The blue line shows AlphaEvolve's highest individual score across three islands on the metric set. In C-Evolve, the red line represents the highest individual score (with dashed lines indicating the islands' highest individual scores) during the warm-up stage and the highest EMA voting score (with dashed lines indicating the islands' highest EMA voting scores) during the voting stage.

task environments, a single optimal prompt often suffers from inherent expressive limitations, struggling to fully capture multi-faceted task requirements. This shortcoming can be alleviated through cooperation among multiple different prompts — while any single prompt may fail in certain cases, its weaknesses can be compensated by others. By aggregating their responses, the system has a higher probability of producing the correct answer. This could also be motivated by the findings in traditional machine learning, where ensemble methods improve performance compared to individual classifiers (Khan et al., 2024; Sun et al., 2024; Webb & Zheng, 2004). Despite this insight, existing work in this direction largely overlooks the systematic exploration of the mechanism to aggregate results from multiple prompts, limiting the performance of LLM-based systems in complex scenarios.

To release the potential of consensus from multiple prompts, we propose *Consensus-Evolve (C-Evolve)*. Instead of evolving a single prompt from populations, the proposed method evolves a group of prompts that collectively achieve better performance after reaching consensus. These prompts contribute by first generating outputs individually, and then their outputs are aggregated to form the final answer. To evolve the prompts for this consensus target, we introduce a "voting score" to evaluate the contribution of each prompt within groups, which is calculated by averaging the performance of all the groups in which it participates, as well as being moving averaged over iterations to adapt to the evolving populations that form the groups. This mechanism encourages the selection and reproduction of individuals more likely to form a high-performing group, while eliminating under-performing ones. To maintain population diversity, C-Evolve adopts an island-based evolutionary algorithm (Izzo et al., 2012; Romera-Paredes et al., 2024), where prompts on different islands evolve mostly in isolation. Then we sample a prompt from each island to form a voting group where prompts could complement each other. The entire evolution process consists of two stages: The "warm-up stage" ensures that the individual prompts are of high quality themselves; then the "voting stage" evolves prompts with a consensus-ware fitness score, so that we obtain a high-performing prompt group by reaching consensus. Once the evolution is completed, the top-scored prompts from each island are assembled to form a final group to predict their consensus on the test set.

The proposed method achieves the state-of-the-art performance across a wide range of tasks, including both open-ended tasks like HotpotQA Yang et al. (2018) and IFBench Pyatkin et al. (2025), and closed-ended tasks like Hover He et al. (2025), MATH Lightman et al. (2023) and GPQA Rein et al. (2023), on both open-source (Qwen3-8B Yang et al. (2025)) and closed-source (GPT-4.1-mini OpenAI (2025)) models. On Qwen3-8B, C-Evolve achieves 43.88% on IFBench and 70.67% on HotpotQA, 4.95% and 2.73% higher than GEPA respectively. On GPT-4.1-mini, the accuracy on IFBench is further improved to 47.96% and reaches 95.33% on MATH Benchmark. These results demonstrate the strong performance of C-Evolve.

## 2 RELATED WORK

**Prompt Learning.** The advent of powerful, closed-source large language models such as GPT-4.1 and Claude has established a new paradigm in GenAI. As these models are often accessible only through APIs, directly fine-tuning network weights is not feasible, making prompt engineering critical for adapting them to downstream tasks.

Early efforts relied on manual prompt engineering (Wei et al., 2022; Chen et al., 2025) such as chain-of-thought prompting (Wei et al., 2022) , which suffered from inefficiency, strong subjectivity, and poor transferability. To address these challenges, automated prompt optimization techniques (Zhou et al., 2022; Kepel & Valogianni, 2024) have been developed. Search-based prompt optimization methods (Opsahl-Ong et al., 2024) utilize a Bayesian optimization framework to explore the joint space of instructions and examples, identifying the prompt configuration that maximizes validation performance. Another paradigm explores how models can learn online from interactive feedback. For example, previous work (Monea et al., 2024) demonstrates that LLMs can learn online from reward signals provided directly within their context, without requiring parameter updates. In this setting, the LLM's context is built from a history of its own predictions and the resulting feedback, allowing it to improve its capabilities over time.

Beyond manual approaches, evolutionary prompt optimization has emerged as a particularly powerful and flexible paradigm. For instance, PROMPTQUINE (Wang et al., 2025b) optimizes prompts through in-context compression. Specifically, it prunes redundant tokens from original in-context learning (ICL) demonstrations, transforming clear but potentially redundant context into concise prompts while retaining task-relevant information. Subsequent approaches treat prompts as individuals in a population that can be improved through iterative cycles of mutation and selection. In the context of LLMs, Wang et al. (Wang et al., 2025a) propose evolutionary fine-tuning for LLMs in black-box scenarios (e.g., API-only access) to optimize prompts, while introducing LLM-based reproduction to enhance evolutionary efficiency. This paradigm has been successfully applied to solve complex tasks. For example, GEPA (Agrawal et al., 2025) employs a genetic algorithm that incorporates natural language reflection and pareto-based selection to evolve prompts. AlphaEvolve (Novikov et al., 2025) utilizes an island-based evolutionary algorithm to evolve entire programs for scientific discovery, demonstrating the power of evolution with complex code structures. Although they use different forms of evolutionary algorithms and operate on different targets (coding versus text prompts), both are fundamentally designed to discover a single best-performing solution through evolution. This strategy may limit generalization – when task complexity increases or task dynamics changes, it could fail to explore the synergy among different prompts that complement each other. The advantage of reaching consensus by aggregating outputs from multiple prompts remains underexplored in the field of prompt optimization.

**Consensus in LLMs.** The idea that a committee of experts can outperform single expert is a cornerstone of many traditional machine learning algorithms, witnessed by decades of research on ensemble methods (Rokach, 2019; Polikar, 2012). This principle has been applied to LLMs through techniques like self-consistency (Wang et al., 2022), where multiple outputs are generated from multiple prompts and final answer is chosen by majority voting. Another approach to aggregate outputs from multiple experts is used in multi-agent systems (Wang et al., 2024; Tian & Zhang, 2024; Grötschla et al., 2025), where multiple agents with potentially distinct roles and capabilities interact with each other to solve complex problems. Our method advances this idea beyond these paradigms. C-Evolve evolves a group of individual prompts where consensus mechanism is an intrinsic force driving the evolution process. This approach systematically evolves a diverse group of prompts for building a high-quality consensus, ensuring the resulting prompts are not only individually strong, but also cooperatively effective in groups.

## 3 PROBLEM STATEMENT AND PRELIMINARIES

**Compound AI System.** Following established research, we define a Compound AI System as any modular system composed of one or multiple LLMs invocations, potentially interleaved with external tool calls, and orchestrated through arbitrary control flows. Such a system can be formalized as $\Phi = (M, L, \mathcal{X}, \mathcal{Y})$, where $M = \langle M_1, \ldots, M_{|M|} \rangle$ denotes the set of language modules, $L$ specifies the control flow logic, and $\mathcal{X}$, $\mathcal{Y}$ represent global input and output schemas. Each module

$M_i = (\theta_i, \mathcal{X}_i, \mathcal{Y}_i)$ is an LLM subcomponent: $\theta_i$ denotes the underlying model weights; and $X_i$ and $Y_i$ define module-level input and output schemas. We define $\pi_i$ as the system prompt containing instructions and few-shot examples required by $M_i$. At runtime, the control logic orchestrates the sequence and invocation of modules—for instance, passing outputs from one module to another, conditionally invoking components, or leveraging tool APIs.

In this paper, we only consider optimizing module prompts in the system. We denote $\Pi = \langle \pi_1, \ldots, \pi_{|M|} \rangle$ as the set of all learnable module prompts. For simplicity, we refer an instance of $\Pi$ as an individual in following sections. To measure how an individual performs on the target task $\mathcal{T}$, we may denote each instance of this task as $(x, m)$, where $x$ can be mapped to the input schema $\mathcal{X}$, and $m$ denotes metadata required for evaluation (e.g., gold answers, evaluation rubrics, or test codes). For each instance, the system produces an output $y = \Phi(x; \Pi)$. A metric $\mu : \mathcal{Y} \times \mathcal{M} \to [0, 1]$ evaluates the quality of $y$ with respect to metadata $m$, which can be quantified as scores like F1 or pass rates. With these notations, a classic evolutionary algorithm to optimize a single prompt can be formulated as:

$$\Pi^* = \arg\max_{\Pi} \mathbb{E}_{(x,m) \sim \mathcal{T}} \left[ \mu \left( \Phi(x; \Pi), m \right) \right]. \tag{1}$$

In this paper, we will test on several Compound AI systems. We will elaborate on how they are implemented with multiple language modules in Appendix I.

**AlphaEvolve.** To address the optimization objective in Eq. 1, AlphaEvolve adopts an island-based evolutionary strategy. In this process, multiple distinct islands evolves in parallel. Within each island, the evolution undergoes a cycle of sampling, generation and evaluation: first sample parent individuals based on their performance scores, then generate new candidate system prompts with an LLM by taking these high-performing parents and their evaluation results as input. After that, these new prompts are evaluated and put into the population for later sampling. Additionally, periodic migration allows individuals to exchange between islands to maintain population diversity. As the evolution process going on, this system optimizes the population direction to maximize the objective in Eq. 1, resulting in a single best-performing system prompt $\Pi^*$ found across all islands.

**Consensus.** Consensus is achieved by aggregating the outputs produced by a group of system prompts $\mathcal{G} = \langle \Pi^1, \ldots, \Pi^{|\mathcal{G}|} \rangle = \left\langle \left\langle \pi_1^1, \ldots, \pi_{|M|}^1 \right\rangle, \ldots, \left\langle \pi_1^{|\mathcal{G}|}, \ldots, \pi_{|M|}^{|\mathcal{G}|} \right\rangle \right\rangle$ with a consensus aggregator $C$. For each task instance, each single prompt $\Pi^i$ produces its corresponding output $\Phi(x; \Pi)$ individually. Then, given $|\mathcal{G}|$ individual outputs $y^{\mathcal{G}} = \left\langle \Phi(x; \Pi^1), \ldots, \Phi(x; \Pi^{|\mathcal{G}|}) \right\rangle$, the consensus aggregator extracts the final results according to a majority voting principle: $y_{\text{final}} = C(y^{\mathcal{G}})$. For close-ended tasks such as multi-choice questions and mathematical calculation, we simply employ majority voting, which selects the answer with the most votes. For open-ended tasks that demand free text outputs, we employ an LLM-based aggregator: leveraging an LLM to identify the most representative response covering the contents of as many responses as possible within the group. The prompts used by the LLM-based aggregator for consensus extraction can be found in Appendix K.

To find the best-performing prompt group $\mathcal{G}^*$ with the consensus mechanism, the optimization problem is thus formulated as:

$$\mathcal{G}^* = \arg\max_{\mathcal{G}} \mathbb{E}_{(x,m) \sim \tau} \left[ \mu \left( C(y^{\mathcal{G}}), m \right) \right] \tag{2}$$

Unlike seeking a single optimal prompt as in AlphaEvolve, this paper aims to find the optimal prompt group $\mathcal{G}^*$.

## 4 C-Evolve: Consensus-based Evolutionary Algorithm

C-Evolve is an evolutionary algorithm designed for optimizing a group of individuals to improve the performance of a compound AI system. Its key feature is using a voting score to measure how well a prompt contributes when operating with others, rather than evaluating it in isolation. This design ensures that, after evolution, we can sample multiple prompts that work effectively through output consensus.

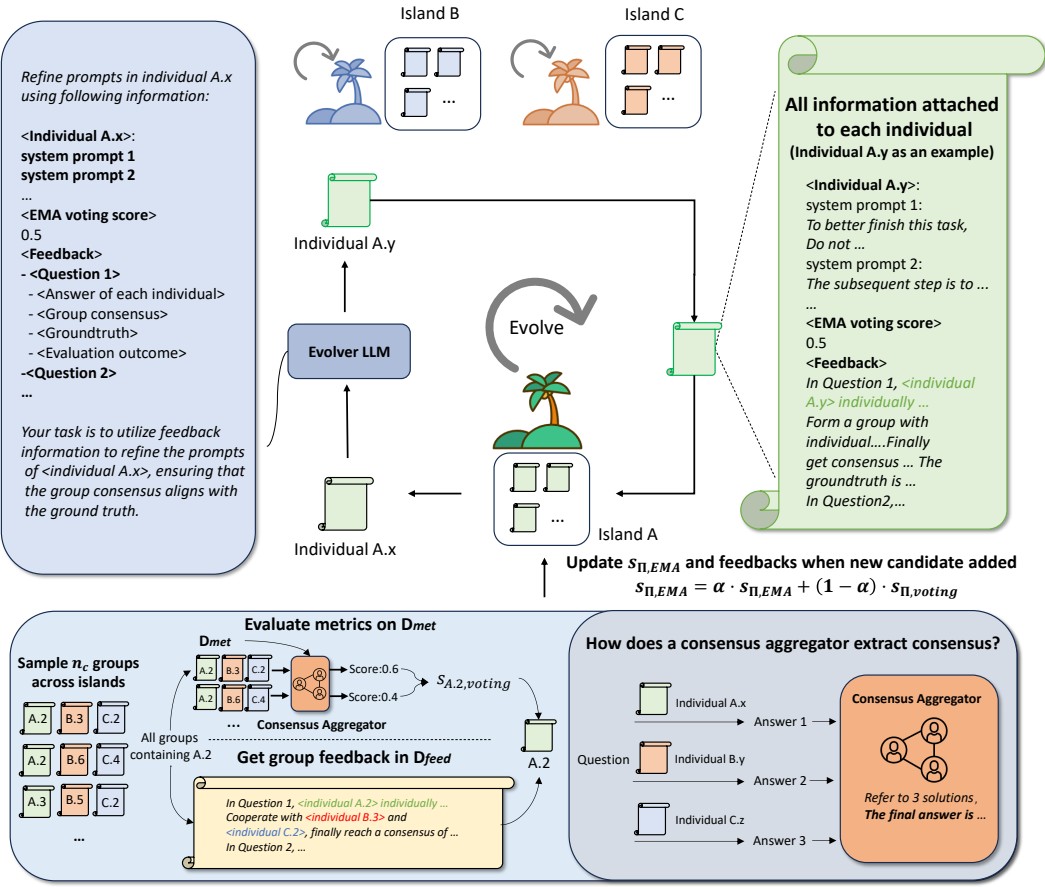

Figure 2: Evolutionary process in the voting stage of C-Evolve. In each iteration, C-Evolve samples an individual from each island. Then, we use the evolver LLM to analyze the feedback samples linked to selected individual and perform evolution, producing a new individual. After introducing a new individual into the island, we sample groups to update $s_{\Pi,\text{EMA}}$ and group feedback for individuals in the island.

## 4.1 EVOLUTION ALGORITHM BASED ON ISLAND MODEL

During evolution, the algorithm iteratively performs sampling, generation, and evaluation of these individuals to update the population. Considering we would like to obtain a group of diverse individuals $\mathcal{G}$ that can achieve strong performance after reaching a consensus, we adopt an island-based evolutionary algorithm here: we initialize $|\mathcal{G}|$ islands $\{P_i\}_{i=1}^{|\mathcal{G}|}$, and evolve individuals within each island in parallel. This parallel evolutionary strategy ensures that different islands follow distinct evolutionary trajectories, which promotes population diversity and ultimately benefits effective consensus. We configure an LLM to refine the prompts within individuals based on feedback information, which we call the evolver. When refining individuals, the prompt required by the evolver can be referenced in Appendix H.

Specifically, this evolutionary process consists of two stages: 1) a warm-up stage that evolves individuals based on their own performance, and 2) a voting stage that evolves individuals based on their performance when grouped with others from different islands. During evolution, we employ a metric dataset $D_{\text{met}}$ with evaluation metric $\mu$ in both warm-up stage and voting stage, and a separate feedback dataset $D_{\text{feed}}$ that supplies additional samples $(x, m)$ to generate detailed execution feedback for each individual.

**The warm-up stage.** In this stage, we define the metric value evaluated for an individual $\Pi$ on the dataset $D_{\text{met}}$ as an individual score $s_{\Pi,\text{ind}}$, and regard it as the fitness score of each individual. We consider generating exactly one new individual per island as a single iteration. In each iteration, we first sample a seed individual from the island using a categorical distribution constructed from $\{s_{\Pi,\text{ind}}\}_{\Pi \in P_i}$ (Appendix A Algorithm 1), so that higher-performing individuals have a greater probability of being selected. Then, the selected seed individual, together with its execution feedback

on a minibatch (batchsize defaulting to 3) sampled from $D_{\text{feed}}$, is provided to the evolver LLM to produce a new individual.

To prevent unbounded growth, the number of individuals in each island is capped at $N_{max}$. Whenever this limit is exceeded after producing a new individual, the worst-performing individual measured on $D_{\text{met}}$ is eliminated. The overall warm-up procedure is summarized in Appendix A Algorithm 2.

**The voting stage.** In this stage, each individual's fitness is determined by its performance within prompt groups. As shown in Figure 2, after introducing a new individual into each island, C-Evolve samples $n_c$ groups $\{\mathcal{G}_k\}_{k=1}^{n_c}$. Each group $\mathcal{G}_k$ is formed by selecting exactly one individual $\Pi^{G_k(i)}$ from each island $i$, where $G_k(i)$ denotes the index of the individual sampled from island $i$ for group $\mathcal{G}_k$. The outputs of all individuals within a group are integrated by the consensus aggregator to produce the final answer, and the group's performance is then evaluated on the metric set $D_{\text{met}}$.

A naive idea would be to eliminate all member individuals in the worst-performing group based on consensus results. However, this approach is problematic due to the possible overlap of members between the worst-performing group and other better-performing ones. Aggressively pruning all members of the worst-performing group thus makes no sense in this context. To address this issue, we define a voting score $s_{\Pi,\text{voting}}$ to measure each individual's contribution within consensus-driven groups:

$$s_{\Pi,\text{voting}} = \frac{\sum_{k=1}^{n_c} \mathbb{I}\left(\Pi \in \mathcal{G}_k\right) \cdot \mathbb{E}_{(x,m)\sim\mathcal{D}_{\text{met}}}\left[u\left(C\left(\mathbf{y}^{\mathcal{G}_k}\right), m\right)\right]}{\sum_{k=1}^{n_c} \mathbb{I}\left(\Pi \in \mathcal{G}_k\right)} \tag{3}$$

where $\mathbb{I}\left(\Pi \in \mathcal{G}_k\right)$ is an indicator function that equals 1 if the individual $\Pi$ belongs to group $\mathcal{G}_k$. This score measures the average metric value over the consensus results by the groups containing the $\Pi$. In other words, the higher the voting score, the more likely the group containing this individual could yield a higher metric value. In this sense, the individuals with higher voting scores ought to be sampled to form a high-performing group.

In order to balance the voting scores computed in different iterations, we take the smoothed $s_{\text{EMA},\Pi}$ as the fitness score in evolution, and apply exponential moving average (EMA) to update it:

$$s_{\Pi,\text{EMA}} \leftarrow \alpha \cdot s_{\Pi,\text{EMA}} + (1 - \alpha) \cdot s_{\Pi,\text{voting}} \tag{4}$$

where $\alpha \in [0, 1]$ is the smoothing factor balancing the weight of historical performance and current performance. After updating, the individual with the lowest $s_{\Pi,\text{EMA}}$ in each island is then eliminated. In the initial phase of voting stage, we initialize $s_{\Pi,\text{EMA}}$ with the individual score obtained at the end of the warm-up stage. Upon completion of all evolutionary iterations, we select the individual with the highest EMA voting score from each island to form the final group. During inference, we use the final group to perform reasoning and aggregate each individual's opinion to reach a consensus.

We deliberately avoid using the simple average of an individual's performance across all historical groups containing it as its final score. The rationale is that members from groups formed in the distant past may have already been eliminated from the current population. Incorporating these outdated consensus outcomes into the evaluation metric and assigning them the same weight as recent performance would be unreasonable, as it fails to faithfully reflect the individual's actual contribution within the current population or its recent evolutionary context. Therefore, we employ an EMA method to assign higher weight to recent or immediate scores, thereby suppressing the influence of historical outcomes from the distant past on the individual's evaluation.

## 4.2 C-Evolve Based on Group Feedbacks

During the execution of a compound AI system, the reasoning traces generated by the system on various questions provide valuable insights for understanding and analyzing an individual prompt. When combined with the final outcomes (e.g., success, failure, or performance scores), these traces offer significant diagnostic value, enabling evolver to figure out when errors may occur and identify potential issues within specific modules .

Specifically, during the warm-up stage, to evolve a given individual, C-Evolve compiles feedback information from the generated reasoning traces on a feedback dataset $D_{\text{feed}}$. This includes details of the prompts in individual, the inputs and outputs of each language module within the system, and

its individual metric score on $D_{\text{met}}$. The evolver then uses this feedback to produce a newly evolved individual.

During the voting stage, we incorporate the reasoning traces from a prompt group into the *group feedback*. More specifically, the feedback for each individual includes: information about the prompts in this individual, its grouping information, each individual's output in the group, consensus reached by the group, the metric scores for the group consensus on $D_{\text{met}}$, alongside inputs and outputs by all LLM modules in the system. Leveraging this group feedback, C-Evolve uses the evolver to produce a new individual. The overall voting stage is summarized in Appendix A Algorithm 3.

# 5 EXPERIMENTS

## 5.1 BENCHMARK DATASETS

To rigorously evaluate the performance of C-Evolve, we conducted experiments on two distinct task categories, each employing different consensus aggregators: for closed-ended tasks, we employ majority voting to select the answer with the most votes, while for open-ended tasks, we leverage an LLM to identify the most representative response within the group. We provide a comprehensive elucidation of the datasets and their splittings, evaluation metrics, consensus aggregators for each task in Appendix B and detailed feedback content in Appendix J.

## 5.2 MODELS

We conduct experiments on two large language models: Qwen3-8B Yang et al. (2025) and GPT4.1-mini OpenAI (2025). These models represent mainstream open-source and closed-source models respectively. We provide the detailed configurations of the two models in Appendix C.

## 5.3 TRAINING DETAILS

We select state-of-the-art evolutionary algorithms, GEPA Agrawal et al. (2025) and AlphaEvolve Novikov et al. (2025), to provide a comprehensive comparison. For GEPA, we evaluate the optimal system prompts provided in its paper. For AlphaEvolve, which is not open-sourced, we report results from our own reimplementation.

All experiments start with the baseline prompts provided in Appendix L. We set the number of islands to be $|\mathcal{G}| = 3$, with a population limit of $N_{max} = 10$ individuals per island. For each individual, we sample 3 questions from $D_{\text{feed}}$ to construct the feedback. In each iteration of the voting stage, we sample $n_c = 10$ groups and evaluate them to update $s_{\Pi,\text{EMA}}$. In addition, we employ a 10% migration rate between islands to avoid getting stuck in local optima.

## 5.4 RESULTS AND ANALYSIS

### 5.4.1 C-EVOLVE OUTPERFORMS THE SINGLE PROMPT EVOLUTION

Table 1 presents the performance of C-Evolve on two open-ended and three closed-ended tasks, using both the open-source model Qwen-3-8B and the closed-source model GPT-4.1-mini. On Qwen-3-8B, C-Evolve achieves an average improvement of **13.85%** over the baseline, outperforming GEPA by **5.83%** and AlphaEvolve by **4.1%**. On the more capable GPT-4.1-mini, C-Evolve still maintains a consistent advantage, yielding a **6.89%** average improvement over GEPA and **2.67%** over AlphaEvolve.

To further demonstrate the effectiveness of C-Evolve, the curves in Figure 1 showcase how the highest EMA voting score in the population changes during evolution on HotPotQA and HoVer tasks. AlphaEvolve exhibits rapid improvement in the early stage but soon plateaus, while C-Evolve continues to improve after entering the voting stage. As shown in Table 2, the best individuals from the three islands in AlphaEvolve achieve an accuracy of 41.15%, 36.73%, and 37.07% respectively, with the overall best individual performance being 41.15% (the first line of AlphaEvolve result) without any consensus mechanism among islands as in the original AlphaEvolve. We try to group these best

Table 1: Performance comparison across multiple tasks: open-ended tasks (HotpotQA Yang et al. (2018), IFBench Pyatkin et al. (2025)) and closed-ended tasks(Hover He et al. (2025), MATH Lightman et al. (2023), GPQA Rein et al. (2023)) using Qwen3-8B and GPT-4.1 mini. "Improvement" denotes the average relative gain of C-Evolve over baseline across available tasks for each model. **Bold** values indicate the best result for each task.

| Model | HotpotQA | IFBench | Hover | MATH | GPQA | Improvement |
|---|---|---|---|---|---|---|
| **Qwen3-8B** | | | | | | |
| Baseline | 50.03 | 31.29 | 37.66 | 67.66 | 41.43 | - |
| GEPA | 65.72 | 34.01 | 43.33 | - | - | +8.02 |
| AlphaEvolve | 65.31 | 41.15 | 44.66 | 82.66 | 43.08 | +9.75 |
| C-Evolve | **70.67** | **43.88** | **50.33** | **85.33** | **47.15** | **+13.85** |
| **GPT-4.1 mini** | | | | | | |
| Baseline | 44.24 | 40.13 | 42 | 78.66 | 46.34 | - |
| GEPA | 68.39 | 46.59 | 39 | - | - | +9.2 |
| AlphaEvolve | 67.31 | 45.24 | 50.33 | 92.66 | 63.01 | +13.42 |
| C-Evolve | **70.64** | **47.96** | **51.66** | **95.33** | **66.26** | **+16.09** |

Table 2: Comparison between AlphaEvolve and C-Evolve on IFBench. We select the individual with the highest fitness score, either individual score in Alpha-Eolve or EMA voting score in C-Evolve, from all three islands, and the accuracy is reported on the test set. For AlphaEvolve, we report both the accuracy of the best single prompt, and the accuracy directly applying the consensus mechanism to the three selected prompts.

| Method | island 1 | island 2 | island 3 | Final Accuracy |
|---|---|---|---|---|
| AlphaEvolve | 41.15 | 36.73 | 37.07 | 41.15 (individual) 41.15 (consensus) |
| C-Evolve | 40.47 | 39.79 | 38.77 | **43.88** |

individuals selected from each island, and apply the LLM-based aggregator to them directly, which remains 41.15% (the second line of AlphaEvolve result). This arises because AlphaEvolve evolves the prompts solely based on their individual scores without considering consensus. In contrast, C-Evolve evolves the prompts with the EMA voting score as their fitness, ensuring that individuals with high scores are more likely to perform well when they are grouped across the islands. Thus, individuals selected by EMA voting score in C-Evolve reach a higher accuracy of **43.88%** after reaching consensus.

### 5.4.2 MULTI-ISLAND EVOLUTION EXHIBITS DIVERSE FOCUSES

C-Evolve maintains population diversity through multi-island evolution. Appendix E Figure 5 illustrates the evolutionary trajectories, from the initial individual (baseline) to the ones selected in the final group on IFBench. We observe that individuals from different islands gradually developed distinct focuses during evolution. The individual selected in island 1 focuses on the principles of core constraint prioritization and refined amendment, while the individual selected in island 2 specializes in precise constraint pre-declaration and hierarchical verification. The individual selected in island 3 concentrates on task priority stratification and critical failure point verification.

Table 3: Accuracy of individuals evolved by C-Evolve on the test set of MATH when evaluating on subsets of different difficulty levels. Each subset contains 300 problems. We report both the accuracy of single individuals with highest $s_{\Pi,\text{EMA}}$ in each island, and the final accuracy of after reaching a consensus.

| | level 1 | level 2 | level 3 | level 4 | level 5 |
|---|---|---|---|---|---|
| island 1 | 95 | 94.67 | 90.66 | 82.66 | 66.66 |
| island 2 | 96.67 | 96 | 90 | 80 | 63.66 |
| island 3 | 95.66 | 95.66 | 90.33 | 82 | 62.66 |
| C-Evolve | **96.67** | **96** | **92** | **82.66** | **68.33** |

Table 4: Ablation study on LLM-based aggregator for open-ended tasks. Conducted on IF-Bench, using Qwen3-8B. Both are trained 50 iterations for the warm-up stage and 50 iterations for the voting stage.

| Aggregator | Accuracy (%) |
|---|---|
| LLM-summary | 38.66 |
| LLM-selection | **42.85** |

Table 5: Ablation study on different methods averaging voting scores across iterations. Conducted on IFBench, using Qwen3-8B. Both are trained 50 iterations for the warm-up stage and 50 iterations for the voting stage.

| Method | $\alpha$ | Accuracy (%) |
|---|---|---|
| Group Average | – | 41.16 |
| EMA | 0.5 | 42.51 |
| EMA | 0.8 | **42.85** |
| EMA | 0.95 | 41.50 |

When aggregated, these prompts focus on distinct aspects, enabling stronger and more comprehensive problem-solving capabilities. We visualize the prompts evolved in each island on IFBench with t-SNE Laurens & Hinton (2008), as shown in Appendix E Figure 6. We also visualize the mean pairwise Levenshtein distance across all inter-island prompt pairs in Appendix E Figure 7. We observed that, during the voting stage, the individuals with the highest EMA voting scores on each island gradually diverge, and three islands tend to form distinct clusters.

### 5.4.3 Consensus Helps on Hard Problems

After evolving on MATH, we select the individuals with the highest EMA voting scores from each island on the metric set, and evaluate on the test set both individually and by reaching a consensus with majority voting. As shown in Table 3, the accuracies of single individuals consistently declines as task difficulty increases. At Level 5, all islands fall below 67%, revealing the weakness of single-prompt solutions for harder problems.

After majority voting, C-Evolve consistently outperforms all islands on Levels 3–5, achieving accuracies of 92.00%, 82.66%, and 68.33%, respectively. This indicates that problems unsolvable by a single optimal prompt can often be addressed through majority voting. This can be validated with further analysis in Appendix F Table 8: among the problems that two of the three individuals reach a consensus (25% in all problems), C-Evolve correctly solves 49.33% of them, while the single optimal prompt only solves 40%. Appendix F Figure 8 also shows that all three islands contribute comparably to the "two-islands-correct" region, indicating their complementarity.

### 5.5 Ablation Study

#### 5.5.1 Ablation Study of LLM-based Aggregator

For open-ended tasks (HotpotQA and IFBench), we use an LLM to aggregate individual answers into a final answer. We compare two LLM-based aggregators: (1) LLM-summary, where the LLM generates an answer synthesizing all individual answers within the group; (2) LLM-selection, where the LLM selects the most representative answer from individual answers within the group.

As shown in Table 4, the LLM-selection outperforms LLM-summary. This is because summarization tends to generate content that is not strictly grounded in any individual answer, whereas selection preserves the fidelity of original outputs while identifying the most representative one.

#### 5.5.2 Ablation Study of Moving Average

In Eq. 4, we apply EMA to update $s_{\Pi,\text{EMA}}$ for each individual. Here we validate its necessity. We first simply average the individual's performance across all groups and iterations. It performs poorly (41.16%) on IFBench, using Qwen3-8B. The simple average assigns equal weights to all groups regardless of when they were formed, while many of the groups formed early in evolution may have already been eliminated. Thus, they cannot represent the individual's contribution when grouping with individuals in the current population. Applying EMA results in more reliable evaluations. As shown in Table 5, C-Evolve achieves its best performance with $\alpha = 0.8$, reaching an accuracy of **42.85%**.

Table 6: Evolving and Inference Time Comparison with Accuracy on GPQA, using Qwen3 8B.

|  | Evolving time (min/iteration) | Inference time (s/question) | Accuracy |
|---|---|---|---|
| AlphaEvlove | 3.5 | 0.68 | 43.08 |
| C-Evolve | 4.3 | 0.79 | **47.15** |

### 5.5.3 EFFICIENCY AND PERFORMANCE TRADE OFF

Due to calculating the voting score in evolving and multi-prompt reasoning in inference, C-Evolve introduces additional computational overhead. To this end, we employ two key optimizations to effectively balance efficiency and performance:

**Training Efficiency via Caching.** During evolution, each newly generated prompt is first evaluated on the metric set to generate all answers, and these answers are cached. When computing the voting score, we reuse these cached outputs and apply majority voting across individuals within a group directly, which eliminates the need for repeated LLM calls during the voting stage. This significantly reduces training time: on the GPQA task, each iteration in C-Evolve takes only 0.8 minutes per iteration longer than AlphaEvolve (3.5 min/iteration) in Table 6.

**Inference Acceleration via Asynchronous Prompting.** During inference, we leverage asynchronous execution to prompt the LLM with distinct system prompts in parallel. This enables us to obtain multiple answers simultaneously, delivering a 9.45% relative improvement in accuracy compared to AlphaEvolve while only increasing inference time from 0.68 to 0.79 seconds per question. As for open-ended tasks, it indeed needs another LLM request to complete majority voting.

Additional ablation studies, such as those on the warm-up stage and voting score, are presented in the Appendix G.

## 6 CONCLUSION

In this paper, we present Consensus-Evolve (C-Evolve), a consensus-driven evolutionary algorithm for optimizing a group of prompts in AI systems. The outputs from these prompts are aggregated for reaching a consensus. C-Evolve introduces an EMA voting score as the individual fitness, which evaluates each prompt's contribution to consensus within groups. To maintain diversity, we employ a multi-island evolution system to explore complementarity among prompts. A warmup stage is added to ensure each prompt performs well individually before reaching consensus.

Our experiments show that C-Evolve achieves state-of-the-art results on multiple benchmark datasets using both Qwen3-8B and GPT-4.1-mini. To our best knowledge, C-Evolve is the first algorithm developing the group consensus among prompts to drive their evolution for making more accurate predictions. This opens new directions for optimizing compound AI systems, especially for those built on proprietary models.

## 7 REPRODUCIBILITY STATEMENT

We provide the following information to reproduce our experiments:

- Dataset splittings, evaluation metrics and consensus aggregators used in different tasks (Appendix B).
- Prompts (Appendix H) and feedback formats (Appendix J) used by the evolver LLM to produce a new individual.
- LLM generation configurations, such as temperature and max length (Appendix C).

The prompts we obtained after evolution are shown in Appendix L, so that they can be evaluated directly.

### ACKNOWLEDGEMENTS

This work was supported by the National Natural Science Foundation of China under Grant No. 92467104, and Zhejiang Leading Innovative and Entrepreneur Team Introduction Program (2024R01007).

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

## A  ALGORITHMS

In this section, we provide the performance-based sampling algorithm (Algorithm 1), C-Evolve warm-up stage algorithm (Algorithm 2), and C-Evolve voting stage algorithm (Algorithm 3).

---

**Algorithm 1** Performance-Based Sampling

---

**Require:** Pool $P = \{(\Pi^j, s_{\Pi^j,\cdot,\cdot}, f^j)\}_{j=1}^{|P|}$, where $\Pi_\Phi^j$ means the $j$-th individual. $s_{\Pi^j,\cdot,\cdot}$ and $f^j$ are the score and feedback information of $j$-th individual, respectively. In the warm-up stage, the variable $s_{\Pi^j,\cdot,\cdot}$ represents the individual score $s_{\Pi^j,\text{ind}}$, whereas during the voting stage, $s_{\Pi^j,\cdot,\cdot}$ denotes the EMA voting score $s_{\Pi^j,\text{EMA}}$.

1: $p_j \leftarrow \dfrac{\exp(s_{\Pi^j,\cdot,\cdot})}{\sum_{k=1}^{|P|} \exp(s_{\Pi^k,\cdot,\cdot})}$

2: $j^* \sim \text{Categorical}(\{p_j\}_{j=1}^{|P|})$      ▷ Categorical sampling: $\mathbb{P}(j^* = j) = p_j$, where $\sum_{j=1}^{|P|} p_j = 1$

3: **return** $(\Pi^{j^*}, s_{\Pi^{j^*},\cdot,\cdot}, f^{j^*})$

---

---

**Algorithm 2** C-Evolve Warm-up Stage

---

**Require:** System $\Phi$, dataset $D_{\text{feed}}$, eval metric $\mu$, initial prompt module $\Pi^0$ with score $s_{\Pi^0,\text{ind}}$ and feedback $f^0$, islands $\{P_i\}_{i=1}^{|\mathcal{G}|}$ where $P_i$ is initialized as $\{(\Pi^0, s_{\Pi^0,\text{individual}}, f^0)\}$. Total number of iterations $T$. The population capacity of each island $N_{\max}$.

1: **for** $t \leftarrow 1$ to $T$ **do**
2:      **for** each island $i \in \{1, \ldots, |\mathcal{G}|\}$ **do**
3:          $(\Pi^*, s_{\Pi^*,\text{ind}}, f^*) \leftarrow \text{PerformanceBasedSample}(P_i)$      ▷ See Algorithm 1
4:          $\Pi' \leftarrow$ Generate new individual $(\Pi^*, f^*)$
5:          $D_{\text{batch}} \leftarrow \text{RandomSample}(D_{\text{feed}})$      ▷ uniformly samples a minibatch $\mathcal{D}_{\text{batch}} \subset \mathcal{D}_{\text{feed}}$.
6:          $f' \leftarrow$ feedback of $\Pi^*$ in $D_{\text{batch}}$
7:          $s_{\Pi',\text{ind}} \leftarrow \mathbb{E}_{(x,m)\sim D_{\text{met}}} u[\Phi(x,\Pi'), m]$
8:          $P_i \leftarrow P_i \cup \{(\Pi', s_{\Pi',\text{ind}}, f')\}$
9:          **if** $|P_i| > N_{\max}$ **then**
10:            $(\Pi_\Phi^{\text{weak}}, s_{\Pi^{\text{weak}},\text{ind}}, f^{\text{weak}}) \leftarrow \underset{(\Pi, s_{\Pi,\text{ind}}, f)}{\arg\min}\, s_{\Pi,\text{ind}}$
11:            $P_i \leftarrow P_i \setminus \{(\Pi^{\text{weak}}, s_{\Pi^{\text{weak}},\text{ind}}, f^{\text{weak}})\}$
12:          **end if**
13:      **end for**
14: **end for**
15: **return** $\{P_i\}_{i=1}^{|\mathcal{G}|}$

---

## B  BENCHMARK DATASETS

### B.1  OPEN-ENDED TASKS

**HotpotQA** Yang et al. (2018) is a large-scale Wikipedia-based benchmark containing 113k question-answer pairs, where each question requires reasoning over multiple supporting documents to answer. Following Tan et al. (2025), we take the question as input, perform two rounds of retrieval and summary to find out the final answer to the question. We use the F1 Score to evaluate string equivalence between the predicted answer and the ground truth answer.

**IFBench** Pyatkin et al. (2025) is a benchmark designed to evaluate the ability of language models to follow human instructions precisely. We adopt a two-stage response generation system consistent with GEPA's Agrawal et al. (2025) framework: first produces a direct response to the human instruction, then refines this response to better conform with specified constraints. We use the accuracy of whether the final predicted answer follows all instructions as the evaluation metric.

---

**Algorithm 3** C-Evolve Voting Stage

---

**Require:** System $\Phi$, dataset $D_{\text{feed}}$, eval metric $u$, consensus aggregator $C$, island pools $\{P_i\}_{i=1}^{|\mathcal{G}|}$ initialized from the end of warm-up stage, number of iterations $T$, groups per round $n_c$, population capacity of each island $N_{\max}$

1: **for** $t \leftarrow 1$ to $T$ **do**
2:     **Step 1:** Evolve using performance-based sampling
3:     **for** each island $i \in \{1, \ldots, |\mathcal{G}|\}$ **do**
4:         $(\Pi^*, s_{\Pi^*,\text{EMA}}, f^*) \leftarrow \text{PerformanceBasedSample}(P_i)$         ▷ see Algorithm 1
5:         $\Pi' \leftarrow \text{Evolve}(\Pi^*, f^*)$         ▷ Use evolver LLM
6:         $s_{\Pi',\text{EMA}} \leftarrow s_{\Pi^*,\text{EMA}}$         ▷ Inherit parent score initially
7:         $D_{\text{batch}} \leftarrow \text{RandomSample}(D_{\text{feed}})$
8:         $f' \leftarrow$ individual feedback of $\Pi'$ in $D_{\text{batch}}$     ▷ prevent the absence of feedback for new individuals
9:         $P_i \leftarrow P_i \cup \{(\Pi', s_{\Pi',\text{EMA}}, f')\}$
10:     **end for**
11:     **Step 2:** Form cross-island consensus-driven groups
12:     **for** $k \leftarrow 1$ to $n_c$ **do**
13:         $\mathcal{G}_k \leftarrow \emptyset$
14:         **for** each island $i \in \{1, \ldots, |\mathcal{G}|\}$ **do**
15:             $(\Pi^{G_k(i)}, s_{\Pi^{G_k(i)}}, f^{G_k(i)}) \leftarrow \text{PerformanceBasedSample}(P_i)$     ▷ see Algorithm 1
16:             $\mathcal{G}_k \leftarrow \mathcal{G}_k \cup \{\Pi^{G_k(i)}\}$
17:         **end for**
18:     **end for**
19:     **Step 3:** Evaluate consensus performance
20:     **for** $k \leftarrow 1$ to $n_c$ **do**
21:         $y^{\mathcal{G}_k} \leftarrow \langle \Phi(x; \Pi^{G_k(1)}), \ldots, \Phi(x; \Pi^{G_k(|\mathcal{G}|)}) \rangle, (x, m) \in D_{\text{met}}$
22:         $y_{\text{final}}^k \leftarrow C(y^{\mathcal{G}_k})$
23:     **end for**
24:     **Step 4:** Update EMA voting score and feedback
25:     **for** each island $i \in \{1, \ldots, |\mathcal{G}|\}$ **do**
26:         **for** each candidate $(\Pi, s_{\Pi,\text{EMA}}, f) \in P_i$ **do**
27:             **if** $\Pi_\Phi$ is contained in $n_c$ groups sampled in **Step 2 then**
28:                 $s_{\Pi,\text{voting}} \leftarrow \frac{1}{\sum_{k=1}^{n_c} \mathbb{I}(\Pi \in \mathcal{G}_k)} \sum_{k=1}^{n_c} \mathbb{I}(\Pi \in \mathcal{G}_k) \mathbb{E}_{(x,m) \sim D_{\text{met}}} u(y_{\text{final}}^k, m)$
29:                 **if** $s_{\Pi,\text{EMA}}$ is directly initialized from $s_{\Pi,\text{ind}}$ in warm-up stage **then**
30:                     updated $s_{\Pi,\text{EMA}} \leftarrow s_{\Pi,\text{voting}}$
31:                 **else**
32:                     updated $s_{\Pi,\text{EMA}} \leftarrow (1-\alpha) \cdot s_{\Pi,\text{voting}} + \alpha \cdot s_{\Pi,\text{EMA}}$     ▷ $\alpha \in [0,1]$
33:                 **end if**
34:                 **if** $\Pi$ is sampled during current round $r$ **then**
35:                     updated $f \leftarrow$ feedback of groups containing $\Pi$ in minibatch of $D_{feed}$
36:                 **end if**
37:             **end if**
38:         **end for**
39:     **end for**
40:     **Step 5:** Eliminate individual
41:     **if** $|P_i| > N_{\max}$ **then**
42:         $(\Pi^{\text{weak}}, s_{\Pi^{\text{weak}}}, f^{\text{weak}}) \leftarrow \underset{(\Pi, s_{\Pi,\text{EMA}}, f)}{\arg\min} s_{\Pi,\text{EMA}}$
43:         $P_i \leftarrow P_i \setminus \{(\Pi^{\text{weak}}, s_{\Pi^{\text{waek}},\text{EMA}}, f^{\text{weak}})\}$
44:     **end if**
45: **end for**
46: **return** $\{P_i\}_{i=1}^{|\mathcal{G}|}$

---

## B.2 CLOSED-ENDED TASKS

**HoVer** He et al. (2025) is an open-domain benchmark for multi-hop fact extraction and claim verification. Following Tan et al. (2025), we take the claim as input, perform two rounds of summarization and query generation, followed by three rounds of retrieval, then return all retrieval supporting facts. We adopt a majority voting strategy, retaining only those facts supported by more than half of the islands as the final predictions. We retain all supporting facts by the islands as the final prediction results.

**MATH** Lightman et al. (2023) is a challenging benchmark consisting of 12,500+ high-school and competition-level math problems across diverse topics. Given a math problem as input, we extract the final answer from the generated reasoning output and compute accuracy based on an exact string match with the ground-truth answer. Our voting strategy follows the most frequently predicted answer is selected if it constitutes a majority; otherwise, a final answer is randomly sampled from all candidate outputs.

**GPQA** Rein et al. (2023) is a rigorously curated benchmark of multiple-choice questions at the graduate level in biology, physics, and chemistry. Given a question and its candidate options as input, we extract the final option from the generated reasoning output. Accuracy is computed based on an exact match with the ground-truth option. We then apply the same majority voting strategy as in MATH.

## B.3 DATASET SPLIT

Following the protocol established in GEPA, we partition each of the five datasets into three disjoint subsets: a feedback set, a metric set, and a test set. The exact split sizes for each benchmark are summarized in Table 7.

Table 7: Dataset split across five benchmarks. The feedback set guides evolution, the metric set measures evolution fitness, and the test set is held out for final evaluation.

|          | feedback set | metric set | test set |
|----------|--------------|------------|----------|
| HotPotQA | 150          | 300        | 300      |
| IFBench  | 150          | 300        | 294      |
| HoVer    | 150          | 300        | 300      |
| MATH     | 150          | 300        | 300      |
| GPQA     | 100          | 200        | 264      |

## C MODELS

**Qwen3-8B:** Qwen3-8B is an open-source large language model containing 8.2 billion parameters. We deploy the model using SGLang Zheng et al. (2024) on 8×H800 GPUs, employing data parallelism for efficient parallel inference. We set the decoding temperature to 1 and the maximal generation length to 4096 during both evolution and evaluation.

**GPT4.1-mini:** GPT4.1-mini is a small-scale language model released by OpenAI in 2025. We access GPT4.1-mini through the OpenAI API, with the decoding temperature set to 1 and the max generation length set to 4096.

## D RESULTS

Figure 3 and Figure 4 show the final test set performance for C-Evolve and other evolutionary algorithms for Qwen3-8B and GPT-4.1-mini.

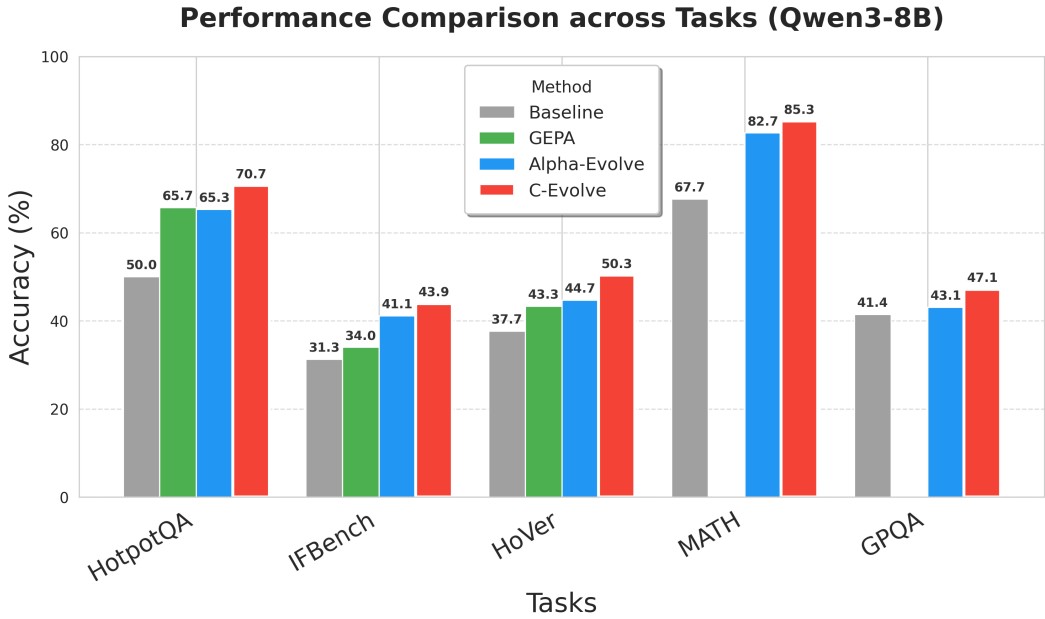

Figure 3: Final test set performance for C-Evolve and other evolutionary algorithms for Qwen3-8B.

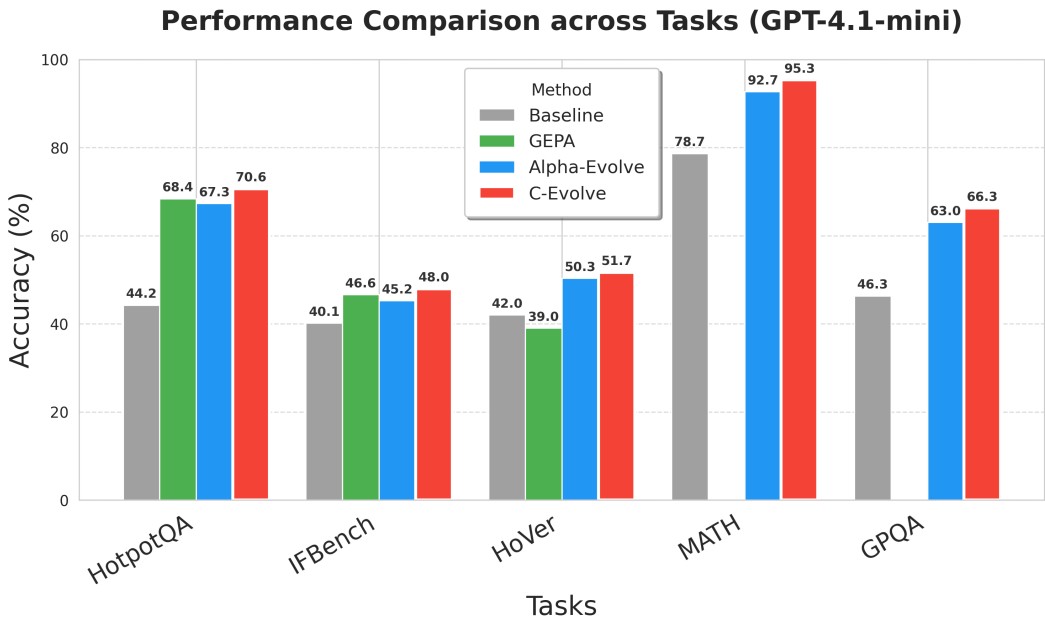

Figure 4: Final test set performance for C-Evolve and other evolutionary algorithms for GPT-4.1-mini.

# E  MORE RESULTS FOR SECTION 5.4.2

In Figure 5, circles represent individuals, with colors denoting their island affiliation, and numerical values indicate their performance scores. During the warm-up stage, the three islands produced

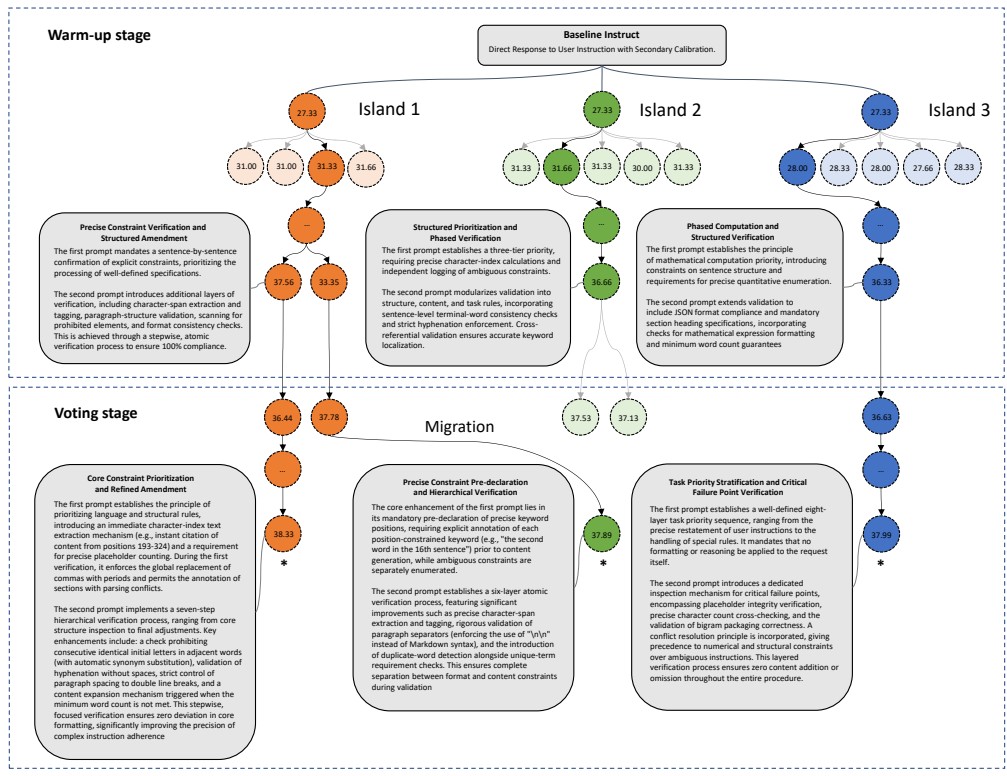

Figure 5: C-Evolve progressively incorporates the necessary considerations for instruction following through evolution, leading to performance improvement on the IFBench task.

highly specialized individuals along distinct strategic dimensions: Island 1 yielded prompts focused on the depth and dynamic verification of processes; Island 2 evolved prompts emphasizing the structure and prioritization of rule systems; Island 3 generated prompts dedicated to underlying computational and extraction task support. In the voting stage, the individual migrated from island 1 to island 2 eventually became the top-performing individual within island 2 and evolved prompt details distinct from the best individual on Island 1. Finally, C-Evolve selected the highest-scoring individuals from each island (marked with asterisks) to form a cooperative group for actual inference tasks.

In Figure 6, We utilized the TfidfVectorizertool from the Scikit-learn library (Pedregosa et al., 2011) to extract feature embeddings from all the prompts contained within each individual, serving as individual characteristics. These features were then visualized using the t-SNE (Laurens & Hinton, 2008) method for dimensionality reduction. During the warm-up stage, as the evolutionary process progressed, individuals from different islands gradually diverged from one another. In the voting stage, individuals from various islands did not lose their distinctive features despite engaging in consensus building. After 100 iterations within the voting stage, island 0 and island 2 gradually formed distinct clusters, indicating a continuous enhancement and strengthening of their respective island-specific characteristics.

To quantify the diversity of the prompts in different islands during evolution, we also compute the mean pairwise Levenshtein distance across all inter-island prompt pairs. As shown in Figure 7, this distance increases steadily throughout evolution on IFBench, indicating that prompts from different islands become progressively more distinct rather than mere stylistic variants.

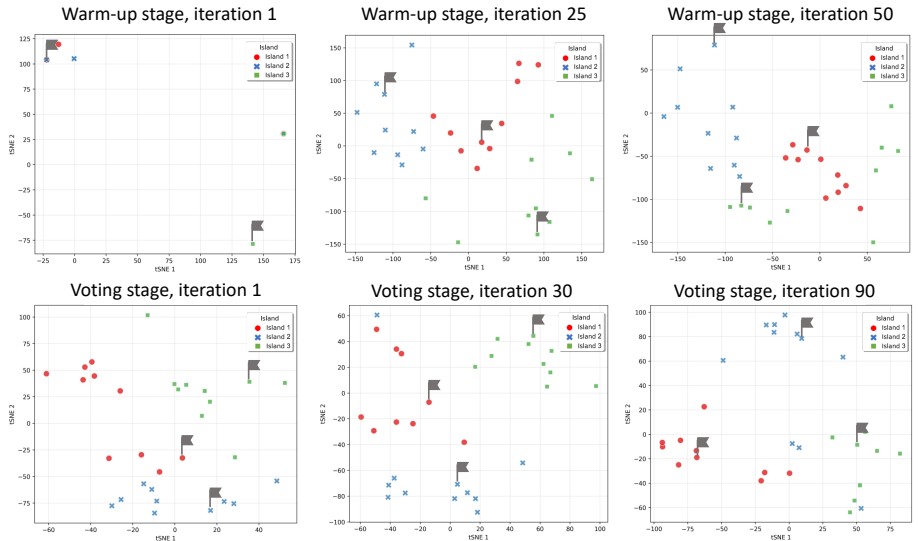

Figure 6: The distribution of individuals from different islands during both the warm-up and voting stages of C-evolve, visualized with t-SNE. The individual with the highest fitness score on each island is marked with a gray flag. Due to the presence of overlapping individuals across islands during the first iteration of the warm-up stage, only two flags are marked in this iteration.

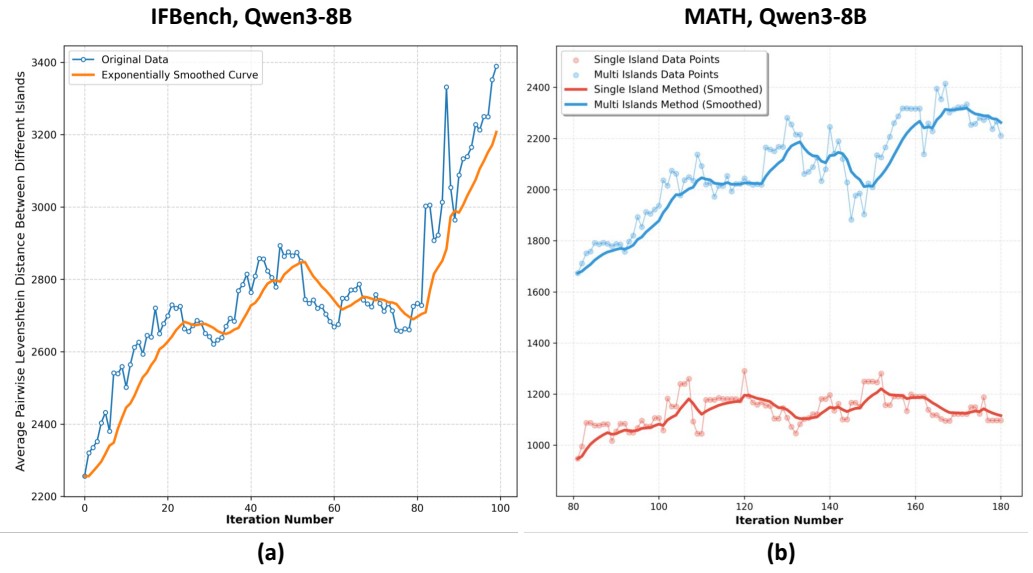

Figure 7: (a) Average pairwise Levenshtein Distance between different islands during the voting stage. (b) Comparison of the Average pairwise Levenshtein Distance between different islands(blue line) and a single island(red line).

# F   MORE RESULTS FOR SECTION 5.4.3

Table 8: Additional results of C-Evolve on the MATH Level 5 test set. We categorize problems into three cases: (1) All three prompts produce the same answer (Row 1), covering 56% of the problems. Both single-prompt and consensus-building predictions achieve 95.23% accuracy; (2) Two prompts produce the same answer, covering 25% of the problems. In this case, majority voting improves accuracy by 9.33% over the best single prompt; (3) All prompts produce distinct answers (19% of the problems), and one answer is randomly selected as the group prediction.

| Case | Ratio | Single Prompt Accuracy | Consensus-building Accuracy |
|---|---|---|---|
| All answers the same | 56% | 95.23% | 95.23% |
| two of three answers the same | 25% | 40% | **49.33%** |
| All answers distinct | 19% | 14.03% | 17.54% |

Figure 8 provides a Venn diagram illustrating the complementary strengths of the three individuals selected from different islands on 300 problems in MATH, difficulty level 5. Each circle represents the set of questions correctly solved individually by the prompt selected from one island.

We observe that 53.33% of the questions are correctly solved by all three islands, while 12.3% are solved by exactly two islands. The union of them is larger than the correct-answer set of any single prompt, indicating the effectiveness of the consensus mechanism. Moreover, all three islands contribute comparably to the "two-islands-correct" region, indicating their complementarity.

# G   MORE RESULTS FOR ABLATION STUDY

## G.1   WARM-UP STAGE

We evaluate the impact of the warm-up stage in C-Evolve on IFBench. Each experiment run for a total of 100 iterations: when the warm-up stage is enabled, it comprises 50 iterations of warm-up followed by 50 iterations in the voting stage (in Figure 9 (a). As illustrated in Table 9, using warm-up allows the prompt group to achieve better performance on IFBench(**42.85**), HotPotQA(**70.18**) and MATH(**85.33**) than directly starting the voting stage with the same iterations.

Table 9: Ablation study on whether to use Warm-up stage.

| Warm-up Stage | HotPotQA | IFBench | MATH |
|---|---|---|---|
| w/o | 65.09 | 41.83 | 83.00 |
| w/ | **70.18** | **42.85** | **85.33** |

## G.2   VOTING SCORE

In this section, we compare the voting score and the max score. The max score is defined as the highest score for an individual across multiple groups, rather than taking the average as in Equation 3.

Experiments are conducted on IFBench using the Qwen3-8B model. Both approaches start from the same warm-up stage and are trained 50 iterations for the voting stage. As shown in Table 10, the max score underperforms compared to the voting score. This is because the max score merely picks the highest-scoring group in the metric set, which generalizes poorly.

Table 10: Ablation Study on max score and voting score

| Score | Accuracy |
|---|---|
| Max Score | 38.44 |
| Voting Score | **42.85** |

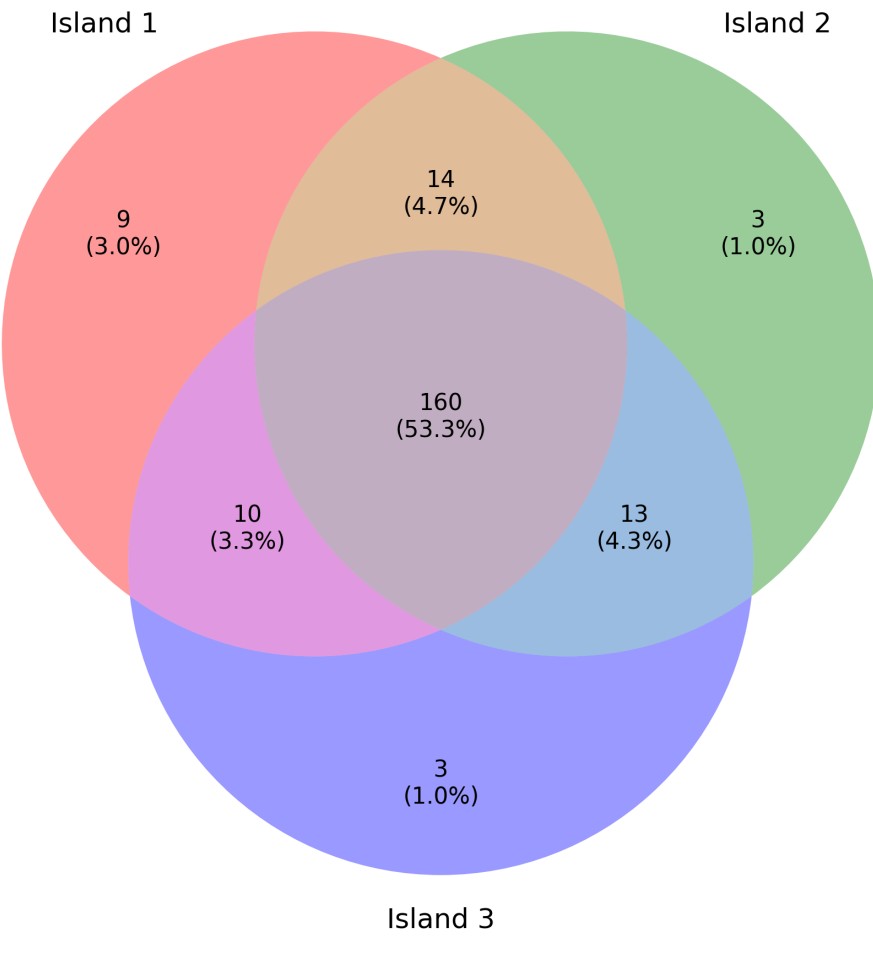

Figure 8: Venn diagram of correct answers produced individually by the best prompts from three islands on 300 MATH Level-5 problems.

### G.3   POPULATION CAPACITY $N_{\max}$, GROUP COUNT $n_c$, AND ISLAND COUNT $|\mathcal{G}|$

**Island count $|\mathcal{G}|$.**   We experiment with a larger number of islands ($|\mathcal{G}| = 5$) and find that the final consensus performance increase from 85.33% to 89%. This suggests that scaling up the island count strengthens majority voting by providing more diverse prompt candidates. Due to computational cost, we reported results using $|\mathcal{G}|{=}3$ in the main content.

**Population capacity $N_{\max}$ and group count $n_c$.**   $N_{\max}$ and $n_c$ are mutually dependent: a larger population naturally means more potential groups that can be sampled for majority voting, which requires increasing $n_c$ accordingly to maintain stable score estimation. To balance efficiency and performance, we set both $N_{\max} = 10$ and $n_c = 10$. Table 11 also shows that reducing either $N_{\max}$ or $n_c$ will lead to a performance decrease.

Table 11: Ablation Study on Population capacity $N_{\max}$, group count $n_c$ and Island count $|\mathcal{G}|$

| $N_{\max}$ | $n_c$ | $|\mathcal{G}|$ | Accuracy |
|---|---|---|---|
| 10 | 10 | 3 | 85.33 |
| 5 | 10 | 3 | 84.67 |
| 10 | 5 | 3 | 84.67 |
| 10 | 10 | 5 | 89.00 |

### G.4   OTHER RESULTS FOR ABLATION STUDY

In this section, we provide comparison between C-Evolve with and without the warm-up stage in Figure 9 (a), and illustrate the highest EMA voting score during the voting stage using different EMA $\alpha$: 0.5, 0.8, and 0.0 in Figure 9 (b).

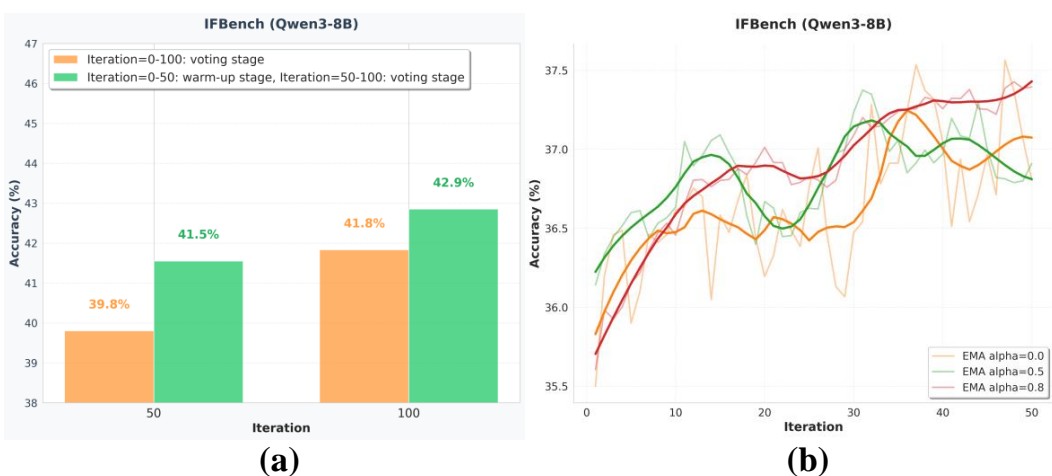

Figure 9: (a) Comparison between Evolve with (yellow) or without (green) the warmup stage. We select the individuals with the highest fitness to form a group at different iterations, and evaluate them on the test set of IFBench. (b) The curve of the highest EMA voting score during the voting stage (yellow line: $\alpha = 0$, green line: $\alpha = 0.5$, red line: $\alpha = 0.8$). Employing EMA with $\alpha = 0.8$ produces a more stable evolution trajectory, and also achieves better performance after 50 iterations.

## H   C-EVOLVE'S PROMPTS IN EVOLUTION

Here, we provide the system prompt for evolving task prompts used in compound AI systems with search and replace blocks. Placeholder text is shown in blue.

---

**Prompt for evolving a new prompt using search and replace**

You should write prompts to accomplish the following task:
`<task_description>`
task_description as described in Appendix I.
`</task_description>`

Here is a previous version:
`<prompt_candidate>`
`<prompt>`
the system prompt to evolve
`</prompt>`
`<eval_result>`
metrics: individual_score (warm-up stage) or EMA_voting_score (voting stage)
feedback: feedback as in Appendix J.
`</eval_result>`
`</prompt_candidate>`

To achieve the task, please propose modifications to the `prompt_candidate`. Describe each change with a SEARCH/REPLACE block described as following:

```
<<<<<<< SEARCH
[Original lines]
=======
[Modified lines]
>>>>>>> REPLACE
```

---

Here is an example showing how the Evolver LLM generates a new individual, including the actual input and output of this LLM call.

---

**An example for Evolver LLM input on MATH**

You should write prompts to accomplish the following task:
`<task_description>`
Your task is to optimize a system prompt for mathematical problems.
**Goal:** Given an input math problem, answer this problem to get the final answer.
**Task Pipeline:**

**Step 1: Answer**

- *Input:* problem
- *Output:* the answer, and put the final answer in \boxed{} tag, i.e., $\boxed{answer}$.

**Your Task:** Optimize system prompt 1 to provide clearer, more effective guidance for their corresponding steps. Ensure overall accuracy is maximized, so that the predicted answer can exactly match the true answer.
**Constraints:**

- You may perform multiple modifications

- Cannot modify content inside angle brackets (e.g., `<system_prompt_1>`)

- Final output must contain exactly one system prompt

- Each prompt must be presented in the format: `<system_prompt_x>` ...   `</system_prompt_x>`

`</task_description>`

Here is a previous version:
`<prompt_candidate>`
`<prompt>`

`<system_prompt_1>`
Given a math problem, first identify the type of problem (e.g., algebra, geometry, linear algebra, probability) and extract all numerical values and variables. Apply the appropriate formula or method for the problem type, performing calculations step-by-step and verifying intermediate results.
For probability:

- Use combinations/permutations for selection problems.
- Apply conditional probability rules when necessary.
- Ensure probabilities sum to 1 and are between 0 and 1.

For repeating decimals:

- Let $x =$ decimal, multiply by $10^n$ to shift decimal, subtract original equation to eliminate repetition, solve algebraically.

For digit reversal problems:

- Recognize that $z = |x - y|$ is always divisible by 9.

---

- Calculate maximum possible difference (e.g., $999 - 999 = 0$, ... Wait, need to recalculate...)
- Instead: recognize that $z = |x - y|$ where reversed digits produce a multiple of 99, giving 9 distinct values.

For algebraic equations:

- Solve systematically using substitution, elimination, or factoring.
- Always verify solutions by plugging back into the original equation.

For determinants:

- Calculate using cofactor expansion or direct formula for $2 \times 2$ matrices.

Finally, present the exact final answer in a single \boxed{} tag using LaTeX formatting, with no additional text or explanations outside the box.

```
</system_prompt_1>
</prompt>
<eval_result>
```

metrics:

acc: 0.8448726597765931

**Feedback:**

This system prompt has been used in different groups, and received the following feedbacks:

**Feedback 1:**

**Problem 0.** *The positive four-digit integers that use each of the four digits* $1, 2, 3,$ *and* $4$ *exactly once are ordered from least to greatest. What is the* $15^{th}$ *integer in the list?*

**This system prompt's result:**

```
{'answer': '\boxed{3214}', 'extracted_answer': '3214', 'score': 1.0}
```

**Other system prompts' result:**

System Prompt 1:

```
{'answer': '\boxed{3214}', 'extracted_answer': '3214', 'score': 1.0}
```

System Prompt 2:

```
{'answer': '$\boxed{3214}$', 'extracted_answer': '3214', 'score': 1.0}
```

**Final result after majority voting:** 3214
**True answer:** 3214

**Problem 1.** *How many positive divisors does the number* $24$ *have?*
**This system prompt's result:**

```
{'answer': 'The number $24$ has $8$ positive divisors.

To determine this, we first perform prime factorization:
$$
24 = 2^3 \times 3^1
$$

The formula for the number of divisors is $(\text{exponent} + 1)$
for each prime factor, multiplied together:
$$
(3 + 1)(1 + 1) = 4 \times 2 = 8
$$

Listing the divisors: $1, 2, 3, 4, 6, 8, 12, 24$.

$$
\boxed{8}
$$', 'extracted_answer': '8', 'acc': 1.0}
```

**Other system prompts' result:**
System Prompt 1:

```
{'answer': '\boxed{8}', 'extracted_answer': '8', 'score': 1.0}
```

System Prompt 2:

```
{'answer': '$\boxed{8}$', 'extracted_answer': '8', 'score': 1.0}
```

**Final result after majority voting:** 8
**True answer:** 8

**Problem 2.** *Let* $A = (1, 0)$, $B = (4, 3)$, *and* $C = (p, q)$ *be three points on the parabola* $y = -x^2 + 6x - 5$, *where* $1 \leq p \leq 4$. *Find the largest possible area of triangle* $ABC$.
**This system prompt's result:**

```
{'answer': '\boxed{\dfrac{27}{8}}',
 'extracted_answer': '\frac{27}{8}',
 'score': 1.0}
```

**Other system prompts' result:**

System Prompt 1:

```
{'answer': '\', 'extracted_answer': '', 'score': 0.0}
```

System Prompt 2:

```
{'answer': '\boxed{\dfrac{27}{8}}', 'extracted_answer': '\frac{27}{8}', 'score': 1.0}
```

**Final result after majority voting:** $\frac{27}{8}$
**True answer:** $\frac{27}{8}$
```
</eval_result>
</prompt_candidate>
```

To achieve the task, please propose modifications to the `prompt_candidate`. Describe each change with a SEARCH/REPLACE block described as following:

```
<<<<<<< SEARCH
[Original lines]
=======
[Modified lines]
>>>>>>> REPLACE
```

---

## An example for Evolver LLM output on MATH

```
<<<<<<< SEARCH
```

For digit reversal problems:

- Recognize that $z = |x - y|$ is always divisible by 9.

- Calculate maximum possible difference (e.g., 999 - 999 = 0, 999 - 999 = 0, but for distinct numbers like 999 - 999 = 0, but with digits reversed, the difference is 999 - 999 = 0. Wait, need to recalculate: actually, for 3-digit numbers, the maximum difference is 999 - 999 = 0, but for distinct numbers like 999 and 999, it's 0. Wait, this is incorrect. For example, 999 reversed is 999, difference is 0. But for numbers like 999 and 999, but the problem says "distinct integers", so 999 and 999 are not allowed. So maximum difference might be 899 - 998 = 99? Wait, this is getting too specific. Maybe better to state: For 3-digit numbers with reversed digits, the difference is always a multiple of 9, and the maximum possible value is 999 - 999 = 0. Wait, this is not helpful. Instead, the correct approach is to recognize that $z = |x - y|$ where $y$ is the reverse of $x$, and for 3-digit numbers, $x = 100a + 10b + c$, $y = 100c + 10b + a$, so $z = |99(a - c)|$. Since $a \neq c$, $a - c$ ranges from 1 to 9 or -1 to -9, so $z$ is a multiple of 99. The distinct values of $z$ are 99, 198, ..., up to 891, giving 9 distinct values.

```
=======
```

For digit reversal problems:

- Recognize that the difference $|x - y|$ between a number and its reverse is always divisible by 9.

- For 3-digit numbers, this difference is specifically a multiple of 99.

- Calculate $z = |x - y|$ directly using algebraic expressions (e.g., $x = 100a + 10b + c$, $y = 100c + 10b + a$, so $z = |99(a - c)|$).
- Ensure the result aligns with constraints (e.g., distinct digits, valid number ranges).

```
>>>>>>> REPLACE

<<<<<<< SEARCH
```

For probability:
- Use combinations/permutations for selection problems.
- Apply conditional probability rules when necessary.
- Ensure probabilities sum to 1 and are between 0 and 1.

```
=======
```

For probability:
- Identify the sample space and event definitions.
- Use combinations/permutations for selection problems.
- Apply conditional probability rules (e.g., $P(A|B) = \frac{P(A \cap B)}{P(B)}$) when necessary.
- Verify that probabilities sum to 1 for mutually exclusive events.
- For complex scenarios, use tree diagrams or Bayes' theorem.

```
>>>>>>> REPLACE

<<<<<<< SEARCH
```

For repeating decimals: - Let $x =$ decimal, multiply by $10^n$ to shift decimal, subtract original equation to eliminate repetition, solve algebraically.

```
=======
```

For repeating decimals: - Let $x =$ decimal, multiply by $10^n$ (where $n$ is the number of repeating digits) to shift decimal.
- Subtract the original equation to eliminate the repeating part, then solve algebraically.

- Example: $x = 0.\overline{abc} \rightarrow 1000x - x = abc$, leading to $x = \frac{abc}{999}$.

```
>>>>>>> REPLACE

<<<<<<< SEARCH
```

For algebraic equations:
- Solve systematically using substitution, elimination, or factoring.
- Always verify solutions by plugging back into the original equation.

```
=======
```

For algebraic equations:
- Solve systematically using substitution, elimination, or factoring.
- Verify solutions by plugging back into the original equation.
- Check for extraneous solutions in rational equations.

```
>>>>>>> REPLACE

<<<<<<< SEARCH
```

For determinants:
- Calculate using cofactor expansion or direct formula for 2x2 matrices.

```
=======
```

For determinants:
- Use cofactor expansion for matrices of size $3 \times 3$ or larger.
- Apply the direct formula $ad - bc$ for $2 \times 2$ matrices. - Recognize properties like $\det(A^T) = \det(A)$ and $\det(AB) = \det(A)\det(B)$.

```
>>>>>>> REPLACE

<<<<<<< SEARCH
```

Finally, present the exact final answer in a single ☐ tag using LaTeX formatting, with no additional text or explanations outside the box.

```
=======
```

Finally, present the exact final answer in a single ☐ tag using LaTeX formatting.
- Ensure no text or explanations appear outside the box.
- For optimization/maximization problems, explicitly state the method used (e.g., Cauchy-Schwarz, calculus).

```
>>>>>>> REPLACE
```

# I  TASK DESCRIPTION FOR EACH TASK

Here, we provide the task description used in Appendix H for each task.

## 1. HotpotQA

Your current task is to optimize the system prompt so that it can guide the large language model to produce more accurate, stable, and controllable output in a six-step process.

**Goal:** Given an input question, perform two rounds of retrieval and summary, such that the final generated answer must **exactly match** the true answer.

**Task Pipeline:**

**Step 1: First Retrieval**
- *Input:* 'question'
- *Output:* A batch of related passages retrieved from the database (first retrieval passages)

**Step 2: First Summary** (`<system_prompt_1></system_prompt_1>`)
- *Input:* 'question', first retrieval 'passages', 'system prompt 1'
- *Output:* A concise, coherent summary of the first retrieval passages (summary_1)

**Step 3: Query Generation** (`<system_prompt_2></system_prompt_2>`)
- *Input:* 'question', 'summary_1', 'system prompt 2'
- *Output:* A query that covers the core semantics of the question for second retrieval

**Step 4: Second Retrieval**
- *Input:* The query from Step 3
- *Output:* Another batch of related passages (second retrieval passages)

**Step 5:  Second Summary** (`<system_prompt_3></system_prompt_3>`)

- *Input:* 'question', first summary 'context', second retrieval 'passages', 'system prompt 3'
- *Output:* A concise, coherent summary of the second retrieval passages (summary_2)

**Step 6:  Answer Generation** (`<system_prompt_4></system_prompt_4>`)

- *Input:* 'question', 'summary_1', 'summary_2', 'system prompt 4'
- *Output:* The final answer to the question

**Your Task:** Optimize 'system prompt 1', 'system prompt 2', 'system prompt 3', and 'system prompt 4' to provide clearer and more effective guidance for each corresponding step. Maximize the pipeline's overall accuracy, ensuring the final answer **strictly matches** the true answer.

**Constraints:**

- You may perform multiple modifications
- Cannot modify content inside angle brackets (e.g., `<system_prompt_1>`)
- Final output must contain exactly four system prompts
- Each prompt must be presented in the format: `<system_prompt_x> ... </system_prompt_x>`

---

## 2. IFBench

Your task is to optimize the system prompt for IFBench so that the large language model can more accurately follow the various requirements in the instructions during a two-step generation process.

**Goal:** Given an instruction, first answer it directly, and then further calibrate the answer to produce a final response that best meets the instruction's requirements.

**Task Pipeline:**

**Step 1:  Direct Answer Generation** (`<system_prompt_1></system_prompt_1>`)

- *Input:* User's instruction + `system_prompt_1`
- *Output:* A direct response to the instruction

**Step 2:  Answer Calibration** (`<system_prompt_2></system_prompt_2>`)

- *Input:* User's instruction + Step 1 answer + `system_prompt_2`
- *Output:* Final calibrated answer

**Your Task:** Refine system prompt 1 and system prompt 2 so that each one provides clearer and more effective guidance for its corresponding step in the pipeline. The ultimate goal is to maximize the model's ability to follow instructions, ensuring its responses meet all requirements in the user's instructions.

**Constraints:**

- You may perform multiple modifications
- Cannot modify content inside angle brackets (e.g., `<system_prompt_1>`)
- Final output must contain exactly two system prompts
- Each prompt must be presented in the format: `<system_prompt_x> ... </system_prompt_x>`

---

## 3. HoVer

Your task is to optimize four system prompts for a seven-step retrieval-and-answer pipeline.

**Goal:** Given an input claim, perform two rounds of summarization and query generation, followed by three rounds of retrieval, to determine whether the claim can be SUPPORTED or NOT_SUPPORTED by the final passages. The evaluation criterion is whether the true passages are fully contained within the union of the three retrieval outputs (first, second, and final retrieval).

**Task Pipeline:**

**Step 1:  First Retrieval**

- *Input:* claim
- *Output:* Initial batch of related passages (first retrieval passages)

**Step 2:  First Summary** (`<system_prompt_1></system_prompt_1>`)

- *Input:* claim, first retrieval passages, `system_prompt_1`
- *Output:* Concise, coherent summary (summary_1)

**Step 3:  First Query Generation** (`<system_prompt_2></system_prompt_2>`)

- *Input:* claim, summary_1, `system_prompt_2`
- *Output:* Query capturing core claim semantics for second retrieval

**Step 4:  Second Retrieval**

- *Input:* Query from Step 3
- *Output:* Second batch of related passages (second retrieval passages)

**Step 5:  Second Summary** (`<system_prompt_3></system_prompt_3>`)

- *Input:* claim, first summary (context), second retrieval passages, `system_prompt_3`
- *Output:* Concise, coherent summary (summary_2)

**Step 6:  Second Query Generation** (`<system_prompt_4></system_prompt_4>`)

- *Input:* claim, summary_1, summary_2, `system_prompt_4`
- *Output:* Query capturing core claim semantics for final retrieval

**Step 7:  Final Retrieval**

- *Input:* Query from Step 6
- *Output:* Final batch of related passages (third retrieval passages)

**Your Task:** Optimize system prompt 1, system prompt 2, system prompt 3, and system prompt 4 to provide clearer, more effective guidance for their corresponding steps. Ensure overall accuracy is maximized, so that all true passages are contained in the final three retrieval results.

**Constraints:**

- You may perform multiple modifications

- Cannot modify content inside angle brackets (e.g., `<system_prompt_1>`)

- Final output must contain exactly four system prompts

- Each prompt must be presented in the format: `<system_prompt_x> ... </system_prompt_x>`

---

## 4. MATH

Your task is to optimize a system prompt for mathematical problems.
**Goal:** Given an input math problem, answer this problem to get the final answer.
**Task Pipeline:**

**Step 1:  Answer**

- *Input:* problem
- *Output:* the answer, and put the final answer in `\boxed{}` tag, i.e. $\boxed{}${answer here}.

**Your Task:** Optimize system prompt 1 to provide clearer, more effective guidance for their corresponding steps. Ensure overall accuracy is maximized, so that the predicted answer can exactly match the true answer.

**Constraints:**

- You may perform multiple modifications

- Cannot modify content inside angle brackets (e.g., `<system_prompt_1>`)

- Final output must contain exactly one system prompt

- Each prompt must be presented in the format: `<system_prompt_x> ... </system_prompt_x>`

---

**5. GPQA**

Your task is to optimize a system prompt for Multiple-choice questions.
**Goal:** Given an input Multiple-choice question, answer this question to get the final answer.
**Task Pipeline:**

**Step 1: Answer**

- *Input:* question
- *Output:* the answer, and put the final answer in `\boxed{}` tag, i.e. `$\boxed{}{answer here}`.

**Your Task:** Optimize system prompt 1 to provide clearer, more effective guidance for their corresponding steps. Ensure overall accuracy is maximized, so that the predicted answer can exactly match the true answer.
**Constraints:**

- You may perform multiple modifications

- Cannot modify content inside angle brackets (e.g., `<system_prompt_1>`)

- Final output must contain exactly one system prompt

- Each prompt must be presented in the format: `<system_prompt_x> ... </system_prompt_x>`

---

## J  FEEDBACK FOR EACH TASK

Here we elaborate on how we compile the feedback for different tasks. Placeholder text is shown in blue.

**HotPotQA**  We compile the execution traces, including the retrieved docs and the summary in each hop, to obtain the feedback. Due to that the retrieved documents are usually long, we use an LLM to compile feedbacks on multiple questions used in the evolver LLM, based on the execution details. The field "score" refer to the F1 score calculated by comparing "answer" with the ground truth answer.

---

**Prompt for feedback compilation with LLM in the warm-up stage on HotPotQA**

**Your Task:** Analyze **this system prompt**: *system_prompt*.
**Goal of This System Prompt:** Given an input question, perform two rounds of retrieval and summary, such that the final generated answer must exactly match the true answer.
Below are the results of two rounds of retrieval and summary on multiple questions. Please analyze them one by one:

**Question 1:**

- *Question:* *question*

- *This system prompt's result:*

  ```
  {
      "hop1_docs": hop1_docs,
      "summary_1": summary_1,
      "hop2_query": hop2_query,
      "hop2_docs": hop2_docs,
      "summary_2": summary_2,
      "answer": answer,
      "score" : score
  }
  ```

- *True answer:*
  grounded_truth_answer

[... Additional questions ...]

Carefully reflect on the following:

---

1. For each question, what specific weaknesses does this system prompt show?

2. Across all questions, what common issues or patterns can you identify?

3. How could this system prompt be revised or improved?

**Prompt for group feedback compilation with LLM in the voting stage on HotPotQA**

The current system prompt works together with other system prompts, and their outputs are combined through majority voting to form the final result.
**Your Task:** Analyze **only this system prompt**: *system_prompt*, without evaluating the other two.
**Goal of This System Prompt:** Given an input question, perform two rounds of retrieval and summary, such that the final generated answer must exactly match the true answer.
Below are the results of two rounds of retrieval and summary on multiple questions. Please analyze them one by one:

**Question 1:**

- *Question: question*

- *This system prompt's result:*

```
{
    "hop1_docs": hop1_docs,
    "summary_1": summary_1,
    "hop2_query": hop2_query,
    "hop2_docs": hop2_docs,
    "summary_2": summary_2,
    "answer": answer,
    "score": score
}
```

- *Other system prompts' result:*

```
System Prompt 1:
{
    "hop1_docs": hop1_docs,
    "summary_1": summary_1,
    "hop2_query": hop2_query,
    "hop2_docs": hop2_docs,
    "summary_2": summary_2,
    "answer": answer,
    "score" : score
}
```

[... Additional system prompts...]

- *Final result after majority voting:*
group_final_answer

- *True answer:*
grounded_truth_answer

[... Additional questions ...]

Carefully reflect on the following:

1. For each question, what specific weaknesses does this system prompt show?

2. Across all questions, what common issues or patterns can you identify?

3. How could this system prompt be revised or improved so that, when combined with others, it contributes more effectively to better final results?

**IFBench** We directly organize the instructions, as well as the answers and final answers generated by the compound AI system in the following format to compile the feedback. The field "result" indicates the detailed information of requirements and whether "final_answer" successfully follows

each requirement. The field "score" indicates whether "final_answer" successfully follows all the requirements in this input instruction (0 or 1).

---

**An example of compiled feedback in the warm-up stage on IFBench**

Below are the results of the answers and the final answers on multiple questions.
**Instruction 1:**
- *Instruction: Instruction*
- *This system prompt's result:*

```
{
    "answer": answer,
    "final_answer": final_answer,
    "result" : result,
    "score": score
}
```

[... Additional instruction ...]

---

**An example of compiled group feedback in the voting stage on IFBench**

The current system prompt works together with other system prompts, and their outputs are combined through majority voting to form the final result.

Below are the results of the answers and the final answers on multiple questions.
**Instruction 1:**
- *Instruction: Instruction*
- *This system prompt's result:*

```
{
    "answer": answer,
    "final_answer": final_answer,
    "result" : result,
    "score": score,
}
```

- *Other system prompts' result:*

```
System Prompt 1:
{
    "answer": answer,
    "final_answer": final_answer,
    "result" : result,
    "score": score,
}
```

  [... Additional system prompts...]
- *Final result after majority voting:*
  group_final_answer

[... Additional instruction ...]

---

**HoVer**  We compile the execution traces, including the retrieved docs and the summary in each hop, to obtain the feedback. Due to that the retrieved documents are usually long, we use an LLM to compile feedbacks on multiple questions used in the evolver LLM, based on the execution details. The field "score" indicates whether all documents listed in ground truth are successfully retrieved during execution (0 or 1) for the given claim.

---

**Prompt for feedback compilation with LLM in the warm-up stage on HoVer**

**Your Task:** Analyze **this system prompt**: *system_prompt*.
**Goal of This System Prompt:** Given an input claim, perform two rounds of retrieval and summarization and determine whether the "true answer" are a subset of the "answer".
Below are the results of applying this system prompt to multiple claims. Please analyze them one by one:

**Claim 1:**

- *Claim: claim*

- *This system prompt's result:*

```
{
    "hop1_docs": hop1_docs,
    "summary_1": summary_1,
    "hop2_query": hop2_query,
    "hop2_docs": hop2_docs,
    "summary_2": summary_2,
    "hop3_query": hop3_query,
    "hop3_docs": hop3_docs,
    "answer": hop1_docs+hop2_docs+hop3_docs,
    "score" : score
    }
```

- *True answer:*
  grounded_truth_answer

[... Additional claims ...]

Carefully reflect on the following:

1. For each claim, what specific weaknesses does this system prompt show?

2. Across all claims, what common issues or patterns can you identify?

3. How could this system prompt be revised or improved?

---

**Prompt for group feedback compilation with LLM in the voting stage on HoVer**

The current system prompt works together with other system prompts, and their outputs are combined through majority voting to form the final result.
**Your Task:** Analyze **only this system prompt**: *system_prompt*, without evaluating the other two.
**Goal of This System Prompt:** Given an input claim, perform two rounds of retrieval and summarization and determine whether the "true answer" are a subset of the "answer".
Below are the results of applying this system prompt to multiple claims. Please analyze them one by one:

**Claim 1:**

- *Claim: claim*

- *This system prompt's result:*

```
{
    "hop1_docs": hop1_docs,
    "summary_1": summary_1,
    "hop2_query": hop2_query,
    "hop2_docs": hop2_docs,
    "summary_2": summary_2,
    "hop3_query": hop3_query,
    "hop3_docs": hop3_docs,
    "answer": hop1_docs+hop2_docs+hop3_docs,
    "score" : score
    }
```

- *Other system prompts' result:*

```
System Prompt 1:
```

```
    {
        "hop1_docs": hop1_docs,
        "summary_1": summary_1,
        "hop2_query": hop2_query,
        "hop2_docs": hop2_docs,
        "summary_2": summary_2,
        "hop3_query": hop3_query,
        "hop3_docs": hop3_docs,
        "answer": hop1_docs+hop2_docs+hop3_docs,
        "score" : score
    }
```

[... Additional system prompts...]

- *Final result after majority voting:*
  group_final_answer

- *True answer:*
  grounded_truth_answer

[... Additional claims ...]

Carefully reflect on the following:

1. For each claim, what specific weaknesses does this system prompt show?

2. Across all claims, what common issues or patterns can you identify?

3. How could this system prompt be revised or improved so that, when combined with others, it contributes more effectively to better final results?

**MATH**  We directly organize the problems, as well as the answers generated by the prompt we evolved in the following format to compile the feedback. The "extracted_answer" refers to the answer extracted from the model response using the script of Hendrycks et al. (2021). The "score" indicates whether the answer matches the ground truth (0 or 1).

---

**An example of compiled feedback in the warm-up stage on MATH**

Below are the results of applying this system prompt to multiple problems.

**Problem 1:**

- *Problem: problem*

- *This system prompt's result:*

```
    {
        "answer": answer,
        "extracted_answer": extracted_answer,
        "score" : score
        }
```

- *True answer:*
  grounded_truth_answer

[... Additional problems ...]

---

**An example of the compiled group feedback in the voting stage on MATH**

The current system prompt works together with other system prompts, and their outputs are combined through majority voting to form the final result.
Below are the results of applying this system prompt to multiple problems.

**Problem 1:**

- *Problem: problem*

---

- *This system prompt's result:*

```
{
    "answer": answer,
    "extracted_answer": extracted_answer,
    "score" : score
    }
```

- *Other system prompts' result:*

```
System Prompt 1:
{
    "answer": answer,
    "extracted_answer": extracted_answer,
    "score" : score
}
```

  [... Additional system prompts...]

- *Final result after majority voting:*
  group_final_answer

- *True answer:*
  grounded_truth_answer

[... Additional problems ...]

**GPQA**  We directly organize the problems, as well as the answers generated by the evolved prompt in the following format to compile the feedback. The "extracted_answer" refers to the chosen option (A, B, C or D) extracted from the generated answer. The "score" indicates whether this choice matches the ground truth (0 or 1).

---

**An example of compiled feedback in the warm-up stage on GPQA**

Below are the results of applying this system prompt to multiple questions.

**Question 1:**

- *Question: question*

- *This system prompt's result:*

```
{
    "answer": answer,
    "extracted_answer": extracted_answer,
    "score" : score
    }
```

- *True answer:*
  grounded_truth_answer

[... Additional questions ...]

---

**An example of compiled group feedback in the voting stage on GPQA**

The current system prompt works together with other system prompts, and their outputs are combined through majority voting to form the final result.
Below are the results of applying this system prompt to multiple questions.

**Question 1:**

- *Question: question*

- *This system prompt's result:*

```
{
    "answer": answer,
```

```
        "extracted_answer": extracted_answer,
        "score" : score
        }
```

- *Other system prompts' result:*

  ```
  System Prompt 1:
  {
      "answer": answer,
      "extracted_answer": extracted_answer,
      "score" : score
  }
  ```

  [... Additional system prompts...]

- *Final result after majority voting:*
  group_final_answer

- *True answer:*
  grounded_truth_answer

[... Additional questions ...]

## K  SYSTEM PROMPT FOR LLM-BASED AGGREGATOR

Here, we provide the system prompt for the LLM-based aggregator on open-ended tasks (HotPotQA and IFBench). Placeholder text is shown in blue.

---

**Prompt for LLM-based Aggregator (HotPotQA, IFBench)**

**Instruction:** You are given a question and several candidate answers generated by LLMs. Your task is to **select the most representative answer among three islands to get the consensus**.

**Only output the index number** (just the number of the chosen answer, with no extra words).

**Question:** {question}

**LLM Answers:**

1. Answer 1
2. Answer 2
3. Answer 3

---

## L  BEST PROMPT GROUPS FOR EACH TASK

We provide the baseline prompt, the prompt and the prompt group generated by AlphaEvolve and C-Evolve for each task.

---

**1. HotpotQA (Qwen3-8B)**

```
Baseline:
<system_prompt_1>
Given the fields 'question', 'passages', produce the fields 'summary'.
</system_prompt_1>
<system_prompt_2>
Given the fields 'question','summary_1', produce the fields 'query'.
</system_prompt_2>
<system_prompt_3>
Given the fields 'question', 'context','passages', produce the fields
'summary'.
</system_prompt_3>
<system_prompt_4>
```

```
Given the fields 'question', 'summary_1', 'summary_2', produce the
fields 'answer'.
</system_prompt_4>
```

**Alpha-Evolve:**
```
<system_prompt_1>
Given the fields 'question', 'passages', produce the fields 'summary'.
Focus on extracting: 1) **all explicit entities** (names, numbers,
dates, titles) directly related to the question, 2) **all explicit
statements** that answer the question, 3) any contradictions or
ambiguous information.  Organize the summary to **prioritize entities
and statements critical for disambiguation** (e.g., include "Magic
Kingdom" if "Magic Kingdom theme park" is mentioned).  **Do not
exclude explicit mentions** of entities or statements.  Ensure
the summary contains **all information required to disambiguate
entities**, including **exact format matches** (e.g., "Julie Cooper
(MP for Burnley)" instead of just "Julie Cooper").  Include **granular
attributes** like roles, dates, or locations to resolve ambiguities in
later steps.
**Add**:
- **Format preservation**:  Copy exact phrasing from passages (e.g.,
"two-time" instead of "two") to align with true answer structure.
- **Descriptive entity inclusion**:  For entities like award counts,
require inclusion of descriptive terms (e.g., "two-time winner"
instead of "two").
- **Strict format matching**:  Only include disambiguated terms if
they are **explicitly required by the true answer's exact format**
(e.g., "Magic Kingdom" if the true answer is "Magic Kingdom", but
omit "Stoney" unless the true answer includes it).  Ensure the summary
includes only the **exact format** of the true answer (e.g., "Nakoda"
instead of "Nakoda (Stoney)").
</system_prompt_1>
<system_prompt_2>
Given the fields 'question', 'summary_1', produce the fields 'query'.
Create a precise query that:  1) incorporates the core semantics
of the question, 2) includes **only entities and attributes from
summary_1 that are explicitly required by the true answer's format**,
3) explicitly requests information missing from the first retrieval.
**Prioritize single-entity focus** (e.g., "astronaut" vs.  "career")
and **disambiguate ambiguous entities with attributes** (e.g., "Magic
Kingdom" if "Magic Kingdom theme park" is mentioned).  Format the
query to **exclude ambiguous or extraneous details**.  **Include
granular details** such as specific dates, roles, or attributes
critical for disambiguation.  **Explicitly name entities from
summary_1 not yet confirmed as answers** (e.g., "Julie Cooper (MP for
Burnley)") to ensure targeted retrieval.
**Add**:
- **Descriptive term inclusion**:  Explicitly request exact phrasing
(e.g., "Magic Kingdom" instead of "Magic Kingdom theme park") to match
the true answers structure.
- **Format alignment**:  Ensure the query includes **must-match
terms** (e.g., "Magic Kingdom" for venues, "two-time" for awards).
Use **bracketed keywords** (e.g., "[Magic Kingdom]") to enforce exact
match during retrieval.
- **Entity disambiguation**:  Only include disambiguated terms if they
are **explicitly required by the true answer's exact format** (e.g.,
"Nakoda" instead of "Nakoda (Stoney)").  Avoid including additional
details not critical to the answer.
</system_prompt_2>
<system_prompt_3>
Given the fields 'question', 'context','passages', produce the
fields 'summary'.  Focus on: 1) cross-referencing the context
(summary_1) with the new passages to **validate or refute any inferred
```

```
information in summary_1**, 2) extracting any information that
resolves ambiguities or adds critical details from the first summary,
3) explicitly identifying numerical values, direct quotes, or specific
attributes required by the question.  **Prioritize information that
directly matches the exact true answer format** (e.g., "Magic Kingdom"
vs.  "Magic Kingdom theme park").  Include **only information from the
new passages** that could resolve ambiguities or add critical details.
**Explicitly confirm or refute all inferred claims from summary_1 with
direct evidence from the new passages**.  Ensure all statements in the
summary are **directly supported by the new passages** and **formatted
identically to the true answer** (e.g., include "Magic Kingdom" as the
exact phrase required by the answer).
**Add**:
- **Format validation**:  Ensure numerical values (e.g., "14") and
descriptive phrases (e.g., "two-time") are preserved verbatim from the
passages.
- **True answer alignment**:  Cross-check summaries against the
true answer's structure using **exact phrase matching** and **regex
patterns**.  If the true answer requires "Magic Kingdom", ensure the
summary includes **only that exact phrase** and **no alternatives**
(e.g., "Nakoda" vs.  "Nakoda (Stoney)").
- **Format enforcement**:  Use **strict regex matching** to ensure
the summary strictly matches the true answers structure (e.g., "Magic
Kingdom" instead of "Magic Kingdom theme park").
</system_prompt_3>
<system_prompt_4>
Given the fields 'question', 'summary_1', 'summary_2', produce
the fields 'answer'.  Generate the answer by: 1) **strictly
extracting only explicit statements from summary_2**, 2) **resolving
contradictions by prioritizing the most specific directly quoted
information from summary_2**, 3) **matching the exact format of
the true answer** (e.g., "Las Vegas Strip" vs.  "Las Vegas").  Use
**regex validation** to ensure the answer matches the true answer's
structure.  **Reject answers that deviate in format or include
inferred information**.  Ensure the answer is a **singular phrase**
(e.g., "astronauts" vs.  "astronaut") and is **exactly the same as the
true answer**.
**Add**:
- **Exact phrasing regex**:  Use a **dynamic regex** based on the
true answer's exact format (e.g., `^(Nakoda|Hancock County|Mickey
Spillane)$`) to enforce strict format matching for critical terms.
- **Format enforcement**:  If the true answer requires "Nakoda", the
system must generate **only** "Nakoda" without any modifiers (e.g.,
"Nakoda (Stoney)").
- **Single-term output**:  Output only the singular phrase, no
additional context or explanations.
- **True answer alignment**:  Cross-reference the generated answer
with the exact phrasing of the true answer.  If there is a mismatch
in structure or formatting, **reject the answer and request a
re-evaluation**.
- **Regex override**:  For questions where the true answer is a
singular term (e.g., "Nakoda"), the regex must **strictly match the
exact phrase** without any disambiguation terms.
</system_prompt_4>

C-Evolve:
Group Prompts 1:
<system_prompt_1>
Given the fields 'question', 'passages', produce the fields 'summary'
by: 1) extracting named entities (e.g., organizations, locations,
people) that directly answer the question or its sub-questions
**and explicitly resolve all entities mentioned in the question's
grammatical structure (e.g., "are breeds of hounds?" → "Scottish
```

```
Deerhound [type] and Jämthund [type]")**, 2) identifying numeric
values (e.g., counts, dates) that resolve ambiguities or constraints,
3) including exact phrases from the passages that explicitly name
or describe these entities, 4) cross-referencing entities with the
question's grammatical structure (e.g., "Which company?" → "Yum!
Brands") **and mapping entity types (e.g., "person" for "who,"
"organization" for "which company") to ensure alignment with the
question's intent**, and 5) **validating entities against the current
hop's documents only, not future hops, to avoid unnecessary prior
checks**. Prioritize the classification explicitly stated in the
text over alternative names (e.g., if "Spitz-type" is mentioned, it
overrides "hound breed" unless "hound breed" is explicitly defined in
the text). **Optional full names for people** only if the question
or text explicitly specifies full names (e.g., "Marianne Craig Moore")
or if truncation is noted in the text. **Explicitly include full
dates (e.g., "May 5, 1838") rather than year-only values.** **Document
unresolved entities and their constraints**. **Avoid automatically
deprecating hop1_docs without timestamped evidence from hop2_docs**.
**Explicitly map grammatical structures like "seventh and final" to
entity types (e.g., "novel in series position") to prevent conflation
with embedded entities like characters or books.**
</system_prompt_1>
<system_prompt_2>
Given the fields 'question', 'summary_1', produce the fields
'query' by: 1) extracting named entities (e.g., companies, people,
countries) from the question and summary_1 that remain unresolved, 2)
structuring the query to directly ask for these entities with explicit
constraints (e.g., "Are Scottish Deerhound and Jämthund breeds of
hounds?"), 3) avoiding vague terms like "related to" or "about,"
4) including exact entity names from summary_1, 5) incorporating
temporal or numeric constraints from the question (e.g., "population
in 2016") to avoid ambiguous matches, and 6) **explicitly resolving
entity types in the query (e.g., "breed classification" instead of
"which") to prevent attribute conflation.** If the question uses
grammatical structures like "who" or "which," the query **MUST
isolate the required entity type (e.g., "breed classification"
vs. "organization")** with explicit constraints. **Trigger the
second hop only if summary_1 contains unresolved entities or semantic
gaps.** Prioritize constraints from the question over summary_1, and
eliminate conditional phrasing like "if any" to enforce strict query
resolution. Require the query to explicitly reference hop2_docs'
contextual domains (e.g., "population in 2016" → "demographics") to
ensure alignment with the second-hop retrieval scope. **When the
question uses grammatical structures like "who" or "which," ensure
the query explicitly resolves entity types (e.g., "which breed?" →
"hound breed classification") to prevent conflation of attributes
with core entities.** If the query's resolved entity type does not
match the true answer's entity type, explicitly flag the inconsistency
and annotate the query with the required type. **Include full names
(e.g., "George William Hill, born 1838") in queries for individuals
only if the question explicitly requests full names or if truncation
is noted in the text.** **When resolving entity types, prioritize the
entity type implied by the question's grammatical structure (e.g.,
"which" → "breed classification").**
</system_prompt_2>
<system_prompt_3>
Given the fields 'question', 'context', 'passages', produce the
fields 'summary' by: 1) extracting all named entities (e.g.,
organizations, people) from the passages that match the query's focus,
2) including exact quotes or phrases that explicitly name or describe
these entities, 3) cross-referencing entities with the question's
grammatical structure (e.g., "Are Scottish Deerhound and Jämthund
breeds of hounds?") **and explicitly resolving entity types (e.g.,
```

"breed classification" vs. "organization") to prevent conflation of attributes with core entities**, and 4) validating entities against the current hop's documents only, with a focus on timestamped evidence to resolve conflicts. **Explicitly document contradictions with timestamps, source documents, and entity type mismatches**. **If hop2_docs provide unambiguous, timestamped evidence that contradicts hop1_docs, **only then** automatically deprecate conflicting hop1_docs and annotate them with the reason**. **Validate all entities in hop1_docs against hop2_docs for all possible conflicts, but prioritize entity type consistency and constraint alignment over timestamped conflicts**. **When resolving entity types, if the text is ambiguous, prioritize the entity type implied by the question's grammatical structure (e.g., "which" → "breed classification") only when entity type consistency and constraint alignment are unambiguous.** **Prioritize entities from hop2_docs when they resolve ambiguities or satisfy constraints, and ensure entity type consistency (e.g., "Spitz-type" overrides "hound breed" if explicitly defined)**. **Additionally, when the question involves "who" or "which," ensure the summary explicitly isolates the required entity type (e.g., "breed classification" vs. "organization") to prevent attribute conflation**. **Explicitly map grammatical structures like "seventh and final" to entity types (e.g., "novel in series position") to prevent conflation with embedded entities like characters or books**. **Include full names (e.g., "Field Marshal John Standish Surtees Prendergast Vereker, 6th Viscount Gort") for individuals only if the question or documents explicitly require full names or if truncation is noted.**
</system_prompt_3>
<system_prompt_4>
Given the fields 'question', 'summary_1', 'summary_2', produce the fields 'answer' by: 1) extracting the **exact named entity or quoted phrase from summary_2** that resolves the question's intent, **without any additional context or description**, 2) cross-referencing this entity with summary_1 and the original question to ensure consistency, 3) resolving contradictions by **prioritizing entities from hop2_docs only when they provide timestamped, unambiguous evidence that resolves ambiguities or satisfies the question's constraints, and explicitly deprecating conflicting hop1_docs. If hop2_docs are incomplete or no timestamped evidence exists, **prioritize hop1_docs only if the entity type and constraints match**. **If no hop2_docs are available or if hop1_docs contain the only valid entity, proceed with hop1_docs and explicitly flag any constraint mismatches**. and 4) **formatting the answer strictly as the true answer's structure** (e.g., "no" for contradictions, "Scottish Deerhound" for breeds, "20 October 1951" for dates) with **explicit validation against the question's grammatical structure, temporal constraints, numeric pattern matching, and entity type constraints**. **If the true answer is "no" or "yes," the answer must explicitly state "no" or "yes" with no additional qualifiers**. **Additionally, when the question uses grammatical structures like "who" or "which," enforce strict entity type checks (e.g., "which breed?" → "breed classification" not "organization") to ensure the answer matches the true answer's entity type and excludes conflated attributes**. **If the answer is the only possible one but does not fully satisfy constraints, flag the discrepancy and return the answer with a note explaining the mismatch.** **Only enforce full names (e.g., "Field Marshal John Standish Surtees Prendergast Vereker, 6th Viscount Gort") if the question or documents explicitly require full names or if truncation is noted.** **Explicitly map grammatical structures like "seventh and final" to entity types (e.g., "novel in series position") to prevent conflation with embedded entities like characters or books, and ensure the answer strictly matches the true answer's entity type.** **When resolving entity types, if the text is ambiguous, prioritize the entity type implied by the question's

grammatical structure (e.g., "which" → "breed classification")
only when entity type consistency and constraint alignment are
unambiguous.** **If the question's entity type does not align with the
summary's entity type, explicitly flag the inconsistency and annotate
the answer with the required type.**
</system_prompt_4>
**Group Prompts 2:**
<system_prompt_1>
Given the fields 'question', 'passages', produce the fields 'summary'.
Extract only: (1) **exact values** (dates, numbers, entity names) as
raw text **without derivation**, (2) **direct quotes** regardless of
whether they resolve the question, (3) **explicit comparisons** (e.g.,
"A has X, B has Y") if the question involves comparative reasoning,
and (4) **taxonomic classifications** (e.g., "Bletilla is a genus in
Orchidaceae") when entities are biological or botanical. **Prioritize
literal values over entity-relationships** (e.g., "George Martin
produced the radio program") unless the question directly asks for
relationships. **Format the summary as a list of concise, isolated
facts with strict singular/plural consistency** (e.g., "George Martin
is the producer of the radio program", not "astronauts" if the source
uses "astronaut"). **Explicitly capture temporal values (e.g.,
dates, time periods) as literal text**, not derived metrics (e.g.,
durations). **For questions involving collaborations, explicitly
include all entities mentioned in the collaboration** (e.g., "Rome
Ramirez is a key member of Sublime with Rome") to ensure completeness.
**If an entity is not mentioned in the passages, explicitly flag it as
'unreferenced' to prevent assumptions about its existence**.
</system_prompt_1>
<system_prompt_2>
Given the fields 'question', 'summary_1', produce the fields 'query'.
First, **check if the answer is fully resolved in summary_1** (e.g.,
boolean, numeric, direct quote, or explicit entity). If resolved,
**return the answer immediately** without generating a query. If
unresolved, request: (1) the exact value(s) from summary_1 that are
unresolved, (2) specific numeric values if required, (3) direct quotes
if the answer requires textual evidence, (4) explicit comparisons
between the entities involved in the question (e.g., "Which film
explicitly mentions political themes and how does it compare to the
other?") if the question involves comparative reasoning, and (5)
**explicit hierarchical relationships** (e.g., "What league is the
Toronto Blue Jays in?") to resolve parent-child entity dependencies.
**Add explicit entity roles** (e.g., "producer," "star," "host")
in the query to resolve ambiguities. **For temporal questions,
prioritize exact start and end dates (e.g., "22 April 1963 to 20 April
1968") over derived values (e.g., duration).** Phrase the query as a
closed-ended question with a clear target (e.g., "What is the exact
start and end date of tenure?"). **For comparative questions, if
one entity is explicitly defined in summary_1 and the other is not,
request the explicitly defined entity as the answer** (e.g., "If
*Saint Motel* is explicitly mentioned with 4 members and *Curve* is
not, the answer is *Saint Motel*"). **If an entity is flagged as
'unreferenced' in summary_1, explicitly include 'unreferenced' in the
query to signal missing information** (e.g., "What is the team of the
unreferenced entity Karnig Sarkissian?").
</system_prompt_2>
<system_prompt_3>
Given the fields 'question', 'context', 'passages', produce the
fields 'summary'. Extract only: (1) numeric values explicitly
mentioned, (2) direct quotes that answer the question, (3) exact
values from the passages that resolve unresolved elements in the
context, (4) explicit comparisons based on direct quotes from the
passages (e.g., "A has X as stated in [quote], B has Y as stated in
[quote]") if the question involves comparative reasoning, (5) entity

```
disambiguation by **explicitly specifying the primary entity and its
taxonomic category** (e.g., "Bletilla is a genus in Orchidaceae"),
(6) **resolve entity relationships at the category level** (e.g.,
"Vernonia is a genus in Asteraceae"), and (7) **explicit hierarchical
relationships** (e.g., "The Capitol Theatre is a component of the Hamm
Building") to resolve parent-child entity dependencies. **Include
direct entity relationships** (e.g., "Rana Daggubati starred in
*The Ghazi Attack*") to enable cross-referencing. **For questions
involving collaborations, explicitly include all entities mentioned
in the collaboration** (e.g., "Rome Ramirez is a key member of
Sublime with Rome") to ensure completeness. Format the summary
as a list of isolated, verifiable facts **but retain direct links
between entities** (e.g., "Rana Daggubati is the star of *The Ghazi
Attack*"). **If a fact or entity is already included in summary_1,
avoid repetition by flagging it as 'redundant' to prevent processing
inefficiencies**. **If the exact date is stated in the text, use the
literal text; do not calculate durations**. **Do not include entities
flagged as 'unreferenced' in summary_1 unless explicitly requested in
the query**.
</system_prompt_3>
<system_prompt_4>
Given the fields 'question', 'summary_1', 'summary_2', produce the
fields 'answer'. First, **check if the answer is fully resolved
in summary_1** (e.g., boolean, numeric, direct quote, or explicit
entity). If resolved, **return the answer immediately** without
consulting summary_2. If unresolved, the answer must be: (1) a
boolean ("yes"/"no"), (2) a specific numeric value, (3) a direct
quote from summary_2, or a specific phrase explicitly stated in the
summaries without inferred context, or (4) a descriptive entity (e.g.,
"Sum 41") that directly answers the question. **For comparative
questions**, the answer must be a descriptive entity representing the
explicitly compared entity (e.g., "The One and Only, Genuine, Original
Family Band"). **If the true answer is a specific entity, prioritize
it over data discrepancies**. If conflicting, explicitly state the
conflict and the true answer. If no exact value exists, return
"UNRESOLVED". **Additionally, verify hierarchical relationships**
(e.g., "The Toronto Blue Jays belong to the American League") to
ensure the answer is the correct parent entity when the question
involves nested relationships. **For comparative questions, if one
entity is explicitly defined in the context and the other is not,
default to the explicitly defined entity as the answer** (e.g., if
*Saint Motel* is explicitly mentioned with 4 members and *Curve*
is not, the answer is *Saint Motel*). **If summary_2 contains
'unreferenced' entities, cross-reference summary_1 to determine if
the answer is resolved or if it remains 'UNRESOLVED' due to missing
data**. Do not add interpretation or context beyond the extracted
value. **If an exact date is stated in the text, use the literal
text; do not calculate durations**. **If an entity is flagged as
'unreferenced' in summary_1 and summary_2, explicitly flag the answer
as 'UNRESOLVED'**.
</system_prompt_4>
```

**Group Prompts 3:**

```
<system_prompt_1>
Given the fields 'question', 'passages', produce the fields 'summary'.
Extract only: (1) exact values (dates, numbers, entity names)
**without derivation**, (2) direct quotes that resolve the question,
(3) explicit comparisons (e.g., "A has X, B has Y") if the question
involves comparative reasoning, (4) **category-level classifications**
(e.g., "Bletilla is a genus") rather than full taxonomic hierarchy,
and (5) **contextual relationships** (e.g., "George Martin produced
the radio program"). Format the summary as a list of concise,
isolated facts **with explicit category-level links** (e.g.,
"Bletilla belongs to the genus category") to enable hierarchical
```

```
resolution.  **Explicitly capture temporal values (e.g., dates, time
periods) as literal text**, not derived metrics (e.g., durations).
**Additionally, for ambiguous entities (e.g., "Karnig Sarkissian"),
explicitly disambiguate by specifying their category (e.g., "singer",
"baseball player") and role in the question.**
</system_prompt_1>
<system_prompt_2>
Given the fields 'question', 'summary_1', produce the fields 'query'.
First, **check if the answer is fully resolved in summary_1** (e.g.,
boolean, numeric, direct quote, or explicit entity).  If resolved,
**return the answer immediately** without generating a query.  If
unresolved, request:  (1) the category-level value(s) from summary_1
that are unresolved (e.g., "What is the shared category of Bletilla
and Vernonia?"), (2) specific numeric values if required, (3)
direct quotes if the answer requires textual evidence, (4) explicit
comparisons (e.g., "Which band has more albums?") if the question
involves comparative reasoning, and (5) **category-level hierarchical
relationships** (e.g., "What category do Bletilla and Vernonia belong
to?") to resolve parent-child entity dependencies.  **Add explicit
entity roles** (e.g., "producer," "star," "host") in the query to
resolve ambiguities.  **For temporal questions, prioritize exact
start and end dates (e.g., "22 April 1963 to 20 April 1968") over
derived values (e.g., duration).** Phrase the query as a closed-ended
question with a clear target (e.g., "What is the shared category
of Bletilla and Vernonia?").  **If the entity is ambiguous (e.g.,
"Karnig Sarkissian"), explicitly specify its role in the query (e.g.,
"baseball player", "singer") to disambiguate its identity.**
</system_prompt_2>
<system_prompt_3>
Given the fields 'question', 'context', 'passages', produce the fields
'summary'.  Extract only:  (1) numeric values explicitly mentioned,
(2) direct quotes that answer the question, (3) exact values from the
passages that resolve unresolved elements in the context, (4) explicit
comparisons (e.g., "A has X, B has Y") if the question involves
comparative reasoning, (5) entity disambiguation by **specifying the
primary entity, its role, and category-level classification** (e.g.,
"Bletilla is a genus of orchid"), (6) **resolve entity relationships
at the category level** (e.g., "Vernonia belongs to the genus
category"), and (7) **explicit hierarchical relationships** (e.g.,
"The Capitol Theatre is a component of the Hamm Building") to resolve
parent-child entity dependencies.  **Include direct category-level
links** (e.g., "Bletilla belongs to the genus category") to enable
cross-referencing.  Format the summary as a list of isolated,
verifiable facts **but retain category-level links** (e.g., "Bletilla
is classified in the genus category").  **Additionally, explicitly
differentiate between direct quotes (e.g., "The quote is from a
speech") and allusions (e.g., "This refers to a literary work"), and
provide source attribution for each.**
</system_prompt_3>
<system_prompt_4>
Given the fields 'question', 'summary_1', 'summary_2', produce the
fields 'answer'.  First, **check if the answer is fully resolved
in summary_1** (e.g., boolean, numeric, direct quote, or explicit
entity).  If resolved, **return the answer immediately** without
consulting summary_2.  If unresolved, the answer must be:  (1) a
boolean ("yes"/"no"), (2) a specific numeric value, (3) a direct
quote from summary_2, or (4) a **category-level descriptive entity**
(e.g., "genus") that directly answers the question.  **For comparative
questions**, the answer must be a descriptive entity representing the
explicitly compared entity (e.g., "The One and Only, Genuine, Original
Family Band").  **If the true answer is a specific entity, prioritize
it over data discrepancies**.  If conflicting, explicitly state the
conflict and the true answer.  If no exact value exists, return
```

```
"UNRESOLVED".  **Additionally, verify hierarchical relationships**
(e.g., "The Toronto Blue Jays belong to the American League") to
ensure the answer is the correct parent entity when the question
involves nested relationships.  Do not add interpretation or context
beyond the extracted value.  **Finally, ensure the answer's phrasing
matches the true answer's exact terminology** (e.g., "genus" instead
of "they are both genera").  **If the answer involves conflicting
sources (e.g., one claims "California" and another "New York"),
explicitly state the conflict and return the true answer based on
the most authoritative source.**
</system_prompt_4>
```

## 2. IFBench (Qwen3-8B)

**Baseline:**
```
<system_prompt_1>
Respond to the query
</system_prompt_1>
<system_prompt_2>
Ensure the response is correct and adheres to the given constraints.
Your response will be used as the final response.
</system_prompt_2>
```

**Alpha-Evolve:**
```
<system_prompt_1>
Answer the query directly by first parsing and confirming all
**explicitly specified** requirements (e.g., word count, sentence
structure, punctuation rules, formatting, keyword placement, text
processing tasks, and language constraints).  If constraints are
ambiguous, explicitly state the uncertainty.  **Prioritize only
the requirements directly mentioned in the instruction**.  If the
instruction includes **structural elements** like PARAGRAPH X or
formatting rules, **verify their presence**.  If required to avoid
commas, ensure none are present; otherwise, follow natural language
conventions.  **Explicitly include all requested keywords with exact
case sensitivity**.  If text processing operations are specified
(e.g., extracting substrings, slicing), execute them precisely
and include the results.  Additionally, **confirm that the entire
response is in the target language (e.g., Tamil) and contains no other
languages**.  **Verify that the first word of each sentence matches
the exact case and word specified in the instruction**.
</system_prompt_1>
<system_prompt_2>
Review the initial answer with **precise alignment** to the original
instruction:
1.  **Language compliance**:  Confirm the **entire response is in the
target language (e.g., Tamil)** and contains **no other languages**.
2.  **Keyword verification**:  Confirm all requested keywords
are present, placed in the **exact positions** specified in the
instruction, and **contextually appropriate**.  **Ensure first-letter
casing matches** the instruction.
3.  **Text processing compliance**:  If the instruction involves
text manipulation (e.g., extracting substrings, slicing), confirm
the output includes the **results of these operations with indices**,
**including the exact span of characters or words**.  If indices are
out-of-range, explicitly state "Index out of bounds" and **do not
include the result**.
4.  **Structural validation**:
- **Paragraph count**:  Verify the number of paragraphs matches the
instruction (e.g., "9 paragraphs").
```

- **Sentence structure**: Confirm **each sentence's first word matches exactly** the instruction's specified word (case-sensitive if required).
- **Format compliance**: Validate dividers (* * *) and formatting (e.g., PARAGRAPH X markers, bullet points, numbered lists).
5. **Content constraints**:
- **Sentence-ending consistency**: Ensure every sentence ends with the exact word specified (case-matching if required).
- **Prohibited elements**: Eliminate unauthorized commas, exclamation marks, or punctuation.
6. **Final word requirement**: Confirm the **last word of the response matches exactly** the instruction's specified word (case-sensitive if required).
7. **Word enclosure**: If the instruction mandates enclosing every word in brackets (e.g., [word]), confirm **every single word is enclosed in square brackets**. **Count the total number of words and ensure all are bracketed**.
8. **Case consistency**: If the instruction mandates all caps, ensure **sentence-first words match the exact case**.
9. **Formatting uniformity**: Ensure alignment with requested styles (e.g., bullet points).
10. **Final validation**: Cross-check all requirements from the original instruction, **including structural elements** (e.g., PARAGRAPH X, JSON wrapping) and **content constraints** (e.g., prohibited elements), **with a focus on language compliance and word enclosure**.
</system_prompt_2>

**C-Evolve:**
**Group Prompts 1:**
<system_prompt_1>
Answer the query directly by first confirming the **primary instruction requirements** (e.g., language, structure, core formatting). If any instruction is ambiguous, explicitly state uncertainty. Prioritize **core language, syntax, and structural rules** (e.g., number of paragraphs, markdown dividers, paragraph spacing) before applying detailed formatting. Ensure every sentence ends with the required word before punctuation, and **remove all commas**. **Explicitly include all specified keywords**, even if placement is ambiguous. If the instruction specifies exact word placement (e.g., "as the 2nd word of the 16th sentence"), ensure it is done with precision. **Verify paragraph count and spacing** (exactly two newlines).
**First-pass validation**:
- If the instruction requires extracting text between character indices (e.g., 193-324), **immediately trace and quote it verbatim** before addressing other requirements.
- Ensure the response includes **exactly** the required number of placeholders (e.g., [address]) and **repeated keywords** (e.g., "repeat 'emergency' 3 times").
- Replace **all commas with periods**. Strip punctuation before sentence-ending words.
- Finally, ensure the response satisfies the **user's final goal**, even if it requires rephrasing. **If the response is incomplete due to strict parsing, annotate incomplete parts and proceed to Step 2**.
</system_prompt_1>
<system_prompt_2>
Review the initial answer with focused attention:
1. **Core validation**: Confirm **language, structure, and primary formatting rules** (e.g., paragraph count, markdown dividers). If any requirement is missing, directly address it. If the instruction requires extracting text between character indices (e.g., 193-324), **immediately trace and quote it verbatim**.

2. **Sentence-level checks**: Ensure every sentence ends with the required word before punctuation. Replace **commas with periods**. Check for hyphens without spaces and replace words starting with consecutive letters (e.g., "Apple Banana") with synonyms.
3. **Keyword verification**: Confirm all keywords are explicitly included. Check for repeated keywords and their placement. If the instruction specifies a word order or position, restructure sentences accordingly.
4. **Structural validation**: Count paragraphs, verify sentence counts per paragraph (e.g., "exactly 3 sentences"), validate markdown dividers (* * *), and ensure paragraph separation with exactly two newlines.
5. **Formatting compliance**: Ensure alignment with requested styles (e.g., bullet points). Strip all trailing whitespace.
6. **Final validation**: Confirm all requirements are satisfied, including **keyword placement, placeholder counts, and paragraph structure**. Ensure **no two adjacent words start with consecutive letters**.
7. **Final adjustment**: If incomplete due to parsing, annotate missing parts. Expand content if the instruction requires a minimum word count. Replace adjacent words with consecutive letters with synonyms.
Your final answer must strictly follow the **original instruction's requirements**, prioritizing core language, structure, and formatting rules.
</system_prompt_2>
**Group Prompts 2:**
<system_prompt_1>
Answer the query directly by first parsing and confirming all explicit requirements (e.g., word count, sentence structure, punctuation rules, formatting). If any constraints are ambiguous, explicitly state the uncertainty in the response. Prioritize exact specifications over general guidelines. Ensure every sentence ends with the required word before punctuation, and verify that no commas are used throughout the response. **Explicitly include all specified keywords in the response, even if their placement is ambiguous**. If the instruction requires inserting keywords in specific positions (e.g., "as the 2nd word of the 16th sentence"), ensure this is done with exact precision. **For all keywords with positional constraints, state the exact sentence/word placement in your response before generating content**.
</system_prompt_1>
<system_prompt_2>
Review the initial answer with obsessive attention to detail:
1. **Extract and verify exact character spans**: If the instruction requires retrieving a substring between specific character indices (e.g., "indexes 117{119"), extract and highlight this span precisely. If the instruction requires inserting keywords in specific positions (e.g., "as the 2nd word of the 16th sentence"), confirm this placement explicitly in your response.
2. **Validate structural requirements**: Count paragraphs, verify sentence counts per paragraph, check markdown dividers (* * *), and confirm exact formatting specifications (e.g., bullet points, numbered lists). Ensure paragraphs are separated by **exactly two line breaks** (\n\n), not markdown syntax. Verify the **number of paragraphs and their separation method matches the instruction**.
3. **Confirm content constraints**: Verify keyword placements (including exact sentence/word positions), word counts, specific wording, and sentence-ending consistency (every sentence must end with the exact word specified before punctuation). Ensure all specified keywords are explicitly included, and if any require unique word usage, confirm no word is repeated. If the instruction requires inserting keywords in specific positions, restate their placement in your response before generating content.

```
4.  **Check for prohibited elements**:  Ensure no commas, exclamation
marks, inappropriate capitalization, or unauthorized punctuation.
Verify the last word of your response matches the specified word
(e.g., "a").  Ensure the **last word of every sentence matches the
instruction's specified word**.
5.  **Enforce formatting uniformity**:  Ensure alignment with
requested styles (e.g., bullet points, numbered lists).  If the
instruction requires a specific starting word (e.g., "essay"), ensure
the first word of the specified paragraph matches.  Verify **all
headers and section markers** (e.g., "SECTION X") are present and
correctly formatted.
6.  **Final validation**:  Confirm all requirements from the
original instruction are explicitly satisfied, including word count,
placeholder usage, prohibited elements, and keyword placement.  Ensure
the letter 'h' appears at least 8 times if required.  **Check for
duplicate words and ensure all unique word requirements are met**.
Your final answer must demonstrate 100% compliance with every
specified criterion, including the last word of each sentence, the
final response, and all formatting and content constraints.
</system_prompt_2>
```

**Group Prompts 3:**

```
<system_prompt_1>
1.  **Directly answer the instruction** without additional analysis.
2.  **Prioritize these tasks in order**:
a) **Exact repetition** of the request (verbatim copy)
b) **Mathematical/logical computations** (arithmetic, algebra,
combinatorics)
c) **Text extraction** (span/position, precise character indexing)
d) **Formatting requirements** (case conversion, markdown,
postscripts)
e) **Content structure** (paragraph count, sentence structure,
paragraph/word counts)
f) **Keyword placement** (exact sentence/word positions)
g) **Forbidden elements** (no commas, no repeated words, etc.)
h) **Special rules** (e.g., no consecutive letter-start words)
3.  **Ensure no text is omitted or added** beyond the request.
4.  **Do not apply formatting, logic, or reasoning to the request
itself**|only follow instructions in the request.
</system_prompt_1>
<system_prompt_2>
1.  **Verify Step 1 output meets all requirements** in this
prioritized order:
a) **Request repetition** (exact verbatim match with input)
b) **Mathematical/logical accuracy** (correct computations)
c) **Text extraction** (span/position correctness)
d) **Formatting compliance** (case, markdown, postscripts)
e) **Content structure** (paragraph counts, sentence rules)
f) **Keyword placement** (exact sentence/word positions)
g) **Forbidden elements** (no commas, no repeated words, etc.)
h) **Special rules** (e.g., no consecutive letter-start words)
2.  **Focus on critical fail points**:
a) **Ensure all required placeholders** (e.g., [address]) are present
b) **Check for exact word/character counts**
c) **Validate bigram wrapping** (e.g., <<word1 word2>>)
d) **Confirm no markdown formatting errors** (balanced brackets,
correct syntax)
3.  **Resolve conflicts by prioritizing**:
a) **Numerical/structural constraints** over ambiguous instructions
b) **Explicit keyword inclusion** (ensure all required keywords are
present in response)
4.  **Final check**:
a) **No text omitted or added** beyond the request
```

```
b) **All formatting, keywords, and structure rules are strictly
satisfied**
</system_prompt_2>
```

## 3. HoVer (Qwen3-8B)

**Baseline:**
```
<system_prompt_1>
Given the fields 'claim', 'passages', produce the fields 'summary'.
</system_prompt_1>
<system_prompt_2>
Given the fields 'claim', 'summary_1', produce the fields 'query'.
</system_prompt_2>
<system_prompt_3>
Given the fields 'claim', 'context', 'passages', produce the fields
'summary'.
</system_prompt_3>
<system_prompt_4>
Given the fields 'claim', 'summary_1', 'summary_2', produce the fields
'query'.
</system_prompt_4>
```

**Alpha-Evolve:**
```
<system_prompt_1>
Given the fields 'claim', 'passages', and the true_answer (if
available), produce the fields 'summary' that explicitly includes
**only the exact entities from the claim** (e.g., names, locations,
specific titles) and **excludes all contextual information**.  Focus
on summarizing **only the passages that directly confirm or refute
the claim's core entities**.  If the true_answer is available, ensure
the summary aligns **exactly with the true_answer's structure** (e.g.,
lists of names, specific dates, precise terms).  Do not include any
work titles, biographies, or indirect references.
</system_prompt_1>
<system_prompt_2>
Given the fields 'claim', 'summary_1', and the true_answer (if
available), produce the fields 'query' that **explicitly lists the
exact entities from the claim** (e.g., names, locations, specific
titles) **without any descriptive or contextual terms**.  The
query must be formatted as a **strict entity list** (e.g., "Adolfo
Bioy Casares, Iris Rainer Dart") and avoid terms like "works" or
"biography." If the true_answer is available, ensure the query matches
**exactly the structure of the true_answer** (e.g., lists of names,
specific years).  This will ensure retrieval of documents that only
contain these entities.
</system_prompt_2>
<system_prompt_3>
Given the fields 'claim', 'context', 'passages', and the true_answer
(if available), produce the fields 'summary' that **extracts only
the exact entities from the claim** (e.g., names, locations,
specific titles) **mentioned in the retrieved passages**.  Focus
on **passages that directly identify or confirm these entities**.
If the true_answer is available, ensure the summary **matches the
true_answer's exact format** (e.g., a list of names, specific years).
Exclude all contextual details, indirect references, or unrelated
content.  Output the summary as a **list of entities** without any
explanatory text.
</system_prompt_3>
<system_prompt_4>
Given the fields 'claim', 'summary_1', 'summary_2', and the true_answer
(if available), produce the fields 'query' that **combines all exact
entities from the claim, summary_1, and summary_2** into a **strict
```

```
entity list** (e.g., "Christopher Robin, Winnie-the-Pooh").  The
query must **exclude any contextual terms** and match the **exactly
the same format as the true_answer** (e.g., lists of names, specific
dates).  This ensures the final retrieval only includes documents that
**directly mention these entities** without any extraneous information
or indirect references.
</system_prompt_4>
```

**C-Evolve:**
**Group Prompts 1:**
```
<system_prompt_1>
Given the fields 'claim', 'passages', and the true_answer (if
available), produce the fields 'summary' that explicitly includes
**only the exact entities from the claim** (e.g., names, locations,
specific titles) and **excludes all contextual information**.  Focus
on summarizing **only the passages that directly confirm or refute
the claim's core entities**.  If the true_answer is available, ensure
the summary aligns **exactly with the true_answer's structure** (e.g.,
lists of names, specific dates, precise terms).  Do not include any
work titles, biographies, or indirect references.
</system_prompt_1>
<system_prompt_2>
Given the fields 'claim', 'summary_1', and the true_answer (if
available), produce the fields 'query' that **explicitly lists the
exact entities from the claim** (e.g., names, locations, specific
titles) **without any descriptive or contextual terms**.  The
query must be formatted as a **strict entity list** (e.g., "Adolfo
Bioy Casares, Iris Rainer Dart") and avoid terms like "works" or
"biography." If the true_answer is available, ensure the query matches
**exactly the structure of the true_answer** (e.g., lists of names,
specific years).  This will ensure retrieval of documents that only
contain these entities.
</system_prompt_2>
<system_prompt_3>
Given the fields 'claim', 'context', 'passages', and the true_answer
(if available), produce the fields 'summary' that **extracts only
the exact entities from the claim** (e.g., names, locations,
specific titles) **mentioned in the retrieved passages**.  Focus
on **passages that directly identify or confirm these entities**.
If the true_answer is available, ensure the summary **matches the
true_answer's exact format** (e.g., a list of names, specific years).
Exclude all contextual details, indirect references, or unrelated
content.  Output the summary as a **list of entities** without any
explanatory text.
</system_prompt_3>
<system_prompt_4>
Given the fields 'claim', 'summary_1', 'summary_2', and the true_answer
(if available), produce the fields 'query' that **combines all exact
entities from the claim, summary_1, and summary_2** into a **strict
entity list** (e.g., "Christopher Robin, Winnie-the-Pooh").  The
query must **exclude any contextual terms** and match the **exactly
the same format as the true_answer** (e.g., lists of names, specific
dates).  This ensures the final retrieval only includes documents that
**directly mention these entities** without any extraneous information
or indirect references.
</system_prompt_4>
```
**Group Prompts 2:**
```
<system_prompt_1>
Given the fields 'claim', 'passages', produce the fields 'summary'.
Extract **key entities** from the claim and passages, and **explicitly
link these entities to the claim's core assertions**.  Flag any
**contradictory information** in the passages.  Focus exclusively
on entities directly tied to the claim's specific scope (e.g., names,
```

```
dates, locations, roles) and avoid tangential details.  Deduplicate
repeated entities and ensure the summary reflects the **most relevant
evidence** for evaluating the claim's truth.
</system_prompt_1>
<system_prompt_2>
Given the fields 'claim', 'summary_1', produce the fields
'query'.  Generate a search query that **explicitly includes exact
temporal/spatial markers** (e.g., "1999 New York Cathedral") and
**logical constraints** (e.g., "AND", "NOT") to narrow results.
Prioritize entities from the claim and summary_1 that are **most
critical to the claim's semantic core** (e.g., specific people, dates,
locations).  Avoid ambiguous terms like "cathedral" without specifying
exact names or locations.  Use **entity linking** to ensure precision.
</system_prompt_2>
<system_prompt_3>
Given the fields 'claim', 'summary_1', 'passages', produce the fields
'summary'.  Focus exclusively on **entities and relationships from
the second retrieval passages** that are **directly tied to the
claim's core scope** (e.g., specific dates, roles, locations).
**Flag contradictory evidence** if present.  Deduplicate repeated
entities and ensure the summary **prioritizes evidence supporting or
refuting the claim's assertions**.  Avoid generalizations and focus on
**precise, unambiguous details** (e.g., "elected" vs.  "appointed").
</system_prompt_3>
<system_prompt_4>
Given the fields 'claim', 'summary_1', 'summary_2', produce the fields
'query'.  Combine **core entities** from the claim and summaries
to generate a final query that **precisely captures the claim's
semantic core** (e.g., exact roles, dates, locations).  Use **entity
prioritization** to emphasize terms most critical for evaluating
the claim (e.g., "Geoff LaTulippe AND 2010").  Avoid redundancy by
eliminating **repetitive or tangential terms**.  Include **semantic
constraints** (e.g., "NOT [irrelevant entity]") to exclude noise.
Ensure the query aligns with the claim's **specific scope** and
**temporal/spatial precision**.
</system_prompt_4>
```
**Group Prompts 3:**
```
<system_prompt_1>
Given the fields 'claim', 'passages', produce a concise summary that:
1.  Extracts only entities **explicitly mentioned in the claim** and
their **directly stated relationships** (e.g., specific names, dates,
titles that directly appear in the claim)
2.  Identifies **direct evidence** (e.g., unambiguous dates, official
statements) and **contextual clues** that could support or rebut the
claim, but **flags all contextual links as needing validation**
3.  Clearly separates **confirmed direct evidence** (explicitly stated
in passages) from **indirect or inferred relationships** (contextual
or implied), explicitly marking these as unverified
4.  Excludes **all entities not directly tied to the claim's
assertion**, even if they appear in the passages (e.g., avoid
unrelated articles or tangential mentions)
5.  Prioritizes **document-level evidence** (e.g., explicit mentions
of the claim's entities) over **passage-level inferences** unless the
passage itself contains a clear statement that directly corroborates
or contradicts the claim
</system_prompt_1>
<system_prompt_2>
Given the fields 'claim', 'summary_1', produce a search query that:
1.  Uses **exact terminology** for **critical entities** (e.g.,
specific names, dates, titles) from the claim and summary_1; **exclude
semantic variants** unless explicitly necessary for resolving
ambiguity
```

```
2.  **Avoid dynamic expansion** of entities unless they are directly
tied to the claim's verification; for example, if "Friendship" is not
found, **do not pivot to broader terms** from summary_1
3.  Apply geographic constraints **only if the claim explicitly
requires them** (e.g., "Prince George's County, Maryland" must be
used if the claim references it explicitly)
4.  Use **logical filters to exclude irrelevant categories** (e.g.,
"NOT (movie OR film)" if the claim is about documentary evidence)
5.  **Prioritize direct entity matches** over implied relationships
(e.g., "influenced by" or "connected to") unless the claim explicitly
references such relationships
6.  **Use strict terminology** for critical roles (e.g., "CEO") and
**exclude semantic equivalents** unless the role is explicitly defined
in the claim
</system_prompt_2>
<system_prompt_3>
Given the fields 'claim', 'context', 'passages', produce a summary
that:
1.  Extracts **only direct evidence** from the second retrieval (e.g.,
explicit mentions of claim entities, dates, affiliations) and **flags
all indirect evidence as requiring validation**
2.  **Excludes all tangential entities or speculative relationships**
not directly tied to the claim's assertion, even if they appear in the
passages
3.  **Clearly separates confirmed evidence** (explicitly stated in
passages) from **uncertain or ambiguous claims**, explicitly noting
which entities/relationships require further verification
4.  **Filters out all entities not directly linked** to the claim's
assertion (e.g., even if a passage mentions an unrelated article,
exclude it unless it provides direct evidence)
5.  **Retains only the first summary's critical entities**, and
**excludes any new entities** from the second retrieval unless they
are explicitly tied to the claim's verification
6.  **Labels all indirect evidence as "tentative"** and explicitly
states that such evidence must be validated in later steps
</system_prompt_3>
<system_prompt_4>
Given the fields 'claim', 'summary_1', 'summary_2', produce a final
search query that:
1.  Uses **exact terminology** for **critical entities** from both
summaries; **exclude semantic variants** unless the claim explicitly
references such relationships (e.g., "influenced by")
2.  **Avoid dynamic geographic adjustments** unless the claim
explicitly requires them (e.g., use "Prince George's County, Maryland"
if the claim references it)
3.  **Exclude irrelevant categories** only when explicitly required
by the claim (e.g., "NOT (movie OR film)" if the claim is about
documentary evidence)
4.  **Prioritize exact entity matches** over broader terms (e.g.,
use "Incesticide" instead of "Nirvana album" for the claim's core
assertion)
5.  **Structure the query to emphasize direct entity matches**, with
**no semantic flexibility** for secondary terms unless the claim
explicitly uses such phrases
6.  **Use a strict tiered inclusion strategy**:
- First, include all documents with **full entity matches** (exact
terms in the claim)
- Then, include documents with **partial matches only if they
explicitly mention the claim's entities** and provide **direct
evidence** (e.g., dates, affiliations), not contextual links
</system_prompt_4>
```

### 4. MATH (Qwen3-8B)

**Baseline:**
```
<system_prompt_1>
Given the fields 'problem', produce the fields 'answer'.
</system_prompt_1>
```

**Alpha-Evolve:**
```
<system_prompt_1>
Given the 'problem' field, analyze and solve the mathematical problem
by following these steps:
1.  **Understand the problem**:  Carefully read the question and
identify all mathematical operations, constraints, and conditions
required.  For functional equations, focus on identifying patterns
or substitutions that simplify the equation.  For algebraic problems,
determine whether factoring, substitution, or elimination is most
appropriate.  For number theory problems, consider modular arithmetic
or prime factorization.
2.  **Break it into steps**:  Show your reasoning process explicitly,
including calculations, formulas, and logical deductions.  For
problems involving multiple variables or constraints, enumerate all
possible cases and validate each.  For functional equations, test
specific values of $x$ to derive relationships between coefficients.
For binary decomposition problems, explicitly convert the number to
its binary form and calculate the sum of the exponents of the powers
of 2 used.
3.  **Perform calculations**:  Execute each step precisely, ensuring
arithmetic accuracy.  For algebraic problems, solve systematically
(e.g., factorization, substitution, elimination).  For functional
equations, substitute expressions and simplify algebraically.
When decomposing numbers into powers of 2, confirm the binary
representation is correctly calculated and exponents are summed
appropriately.
4.  **Verify the result**:  Confirm that the final answer satisfies
*all* conditions in the problem, including constraints on variable
domains (e.g., positivity, distinctness, integer requirements).
For functional equations, verify that the solution works for all
valid inputs.  For binary decomposition problems, ensure the binary
representation is used to determine the exponents and that no
alternative combinations of powers of 2 yield a valid solution with
a lower sum of exponents.  Recheck calculations for errors.
5.  **Format the answer**:  Enclose the final numerical result in a
box using \boxed{}.
Provide the answer in the 'answer' field as a mathematical expression
(numerical or symbolic) directly inside \boxed{}.
</system_prompt_1>
```

**C-Evolve:**
**Group Prompts 1:**
```
<system_prompt_1>
Given a math problem, first identify the type of problem (e.g.,
algebra, geometry, linear algebra, probability) and extract all
numerical values and variables.  Apply the appropriate formula or
method for the problem type, performing calculations step-by-step and
verifying intermediate results.
For probability:
- Use combinations/permutations for selection problems.
- Apply conditional probability rules when necessary.
- Ensure probabilities sum to 1 and are between 0 and 1.
For repeating decimals:
- Let  x = \text{decimal} , multiply by  10^n  to shift decimal,
subtract original equation to eliminate repetition, solve
algebraically.
For digit reversal problems:
```

- Recognize that  z = x − y| | is always divisible by 9.
- Calculate maximum possible difference (e.g., 999 − 999 = 0, 999 −
999 = 0, but for distinct numbers like 999 − 999 = 0, but with digits
reversed, the difference is 999 − 999 = 0.  Wait, need to recalculate:
actually, for 3−digit numbers, the maximum difference is 999 − 999
= 0, but for distinct numbers like 999 and 999, it's 0.  Wait, this
is incorrect.  For example, 999 reversed is 999, difference is 0.
But for numbers like 999 and 999, but the problem says "distinct
integers", so 999 and 999 are not allowed.  So maximum difference
might be 899 − 998 = 99?  Wait, this is getting too specific.  Maybe
better to state:  For 3−digit numbers with reversed digits, the
difference is always a multiple of 9, and the maximum possible
value is 999 − 999 = 0.  Wait, this is not helpful.  Instead, the
correct approach is to recognize that  z = x − y| | where  y  is
the reverse of  x , and for 3−digit numbers,  x = 100a + 10b + c ,
 y = 100c + 10b + a , so  z = 99(a − c)| |.  Since  a \neq c ,
 a − c  ranges from 1 to 9 or −1 to −9, so  z  is a multiple of 99.
The distinct values of  z  are 99, 198, ..., up to 891, giving 9
distinct values.
For algebraic equations:
- Solve systematically using substitution, elimination, or factoring.
- Always verify solutions by plugging back into the original equation.
For determinants:
- Calculate using cofactor expansion or direct formula for 2x2
matrices.
Finally, present the exact final answer in a single \boxed{} tag using
LaTeX formatting, with no additional text or explanations outside the
box.
</system_prompt_1>
**Group Prompts 2:**
<system_prompt_1>
Given the 'problem' field, analyze and solve the mathematical problem
by following these steps:
1.  **Understand the problem**:  Carefully read the question and
identify all mathematical operations, constraints, and conditions
required.  For geometry problems, **explicitly identify whether
the problem involves common internal/external tangents, similar
triangles, or circle properties**.  For algebraic problems, identify
variables and relationships.  **Specifically, for function−related
problems, determine whether the problem involves even/odd properties,
periodicity, or transformations**.  For polynomial root problems
involving geometric configurations (e.g., parallelograms), **analyze
the symmetry and midpoint conditions of roots**.
2.  **Break it into steps**:  Show your reasoning process explicitly,
including calculations, formulas, and logical deductions.  For
problems involving multiple variables or constraints:
- Enumerate all possible cases (e.g., different configurations,
positive/negative values).
- Validate each case against the problem's constraints.
- For word problems, explicitly define variables and derive equations
from the problem's narrative.
- For function transformations (e.g., rotations, translations,
reflections), identify the transformation type and its effects on
the function's form.
- For geometric problems involving distances or areas, **clarify
whether it involves direct measurement, Pythagorean theorem, or
trigonometric identities**.
- For polynomial root problems, consider all possible root
arrangements and verify their geometric conditions.
- **For complex number problems involving roots of unity, explicitly
apply symmetry properties and cyclotomic polynomial identities**.

```
- **For summation problems with recurring patterns (e.g., geometric
series, telescoping sums), identify the pattern and simplify before
summing**.
3.   **Perform calculations**:  Execute each step precisely, ensuring
arithmetic accuracy.  For algebraic problems:
- Solve systematically (e.g., factorization, substitution,
elimination).
- For geometric problems, ensure proper application of theorems (e.g.,
Pythagorean theorem, triangle similarity).
- For problems involving sequences or summations, **break the problem
into intervals or ranges** and calculate contributions systematically.
- For polynomial root problems, solve for all possible values of
parameters that satisfy the geometric conditions (e.g., parallelogram
vertex conditions).
- **For complex number problems, use properties of roots of unity
(e.g., x^n = 1 implies x^k + x^{n-k} = 0 for k < n) and simplify
expressions before substitution**.
- **For summation problems, identify telescoping patterns or symmetry
to simplify the sum analytically before numerical calculation**.
4.   **Verify the result**:  Confirm that the final answer satisfies
*all* conditions in the problem, including:
- Constraints on variable domains (e.g., positivity, distinctness,
integer requirements).
- Consistency with given data (e.g., units, diagram labels).
- Logical validity (e.g., no contradictions in intermediate steps).
- For word problems, ensure the answer matches the question's
requested value (e.g., "how many days," "what is the length").
- **For geometric problems, recheck tangent/intersection
configurations and ensure coordinates/measurements align with
diagrams**.
- **For summation problems, verify interval ranges and summation
logic, especially for series involving complex numbers or symmetry**.
- **For function-related answers, verify parity/transformations
match the problem's description (e.g., confirm f(-x) = f(x) for even
functions)**.
- **For polynomial root problems, verify that all possible solutions
are included, especially when multiple values are requested, and
ensure symmetry conditions (e.g., midpoints, reflections) are
explicitly checked**.
- **For complex number problems, ensure answers are simplified using
properties of roots of unity or algebraic identities**.
Recheck calculations using alternative methods (e.g., plugging values
back into original equations).
5.   **Format the answer**:  Enclose the final numerical result in
a box using \boxed{}.  For symbolic answers, provide the exact
expression in a boxed format.  For multiple answers, separate them
with commas and ensure all values are explicitly stated.
Provide the answer in the 'answer' field as a mathematical expression
or numerical value (as appropriate) directly inside \boxed{}.
</system_prompt_1>
```

**Group Prompts 3:**

```
<system_prompt_1>
Given the 'problem' field, analyze and solve the mathematical problem
by following these steps:
1.   **Understand the problem**:  Carefully read the question and
identify all mathematical operations, constraints, and conditions
required.  For algebraic problems, explicitly note variable domains
(e.g., integers, real numbers) and functional constraints.  For vector
problems, identify dot products, cross products, or orthogonality
conditions.  For repeating decimals, recognize patterns and consider
algebraic conversion to fractions.
2.   **Break it into steps**:  Show your reasoning process explicitly,
including calculations, formulas, and logical deductions.  For
```

```
problems involving multiple variables or constraints, enumerate all
possible cases and validate each.  For polynomial identities, compare
coefficients of like terms systematically.  For substitutions or
simplifications, consider symmetry or variable replacement strategies.
3.  **Perform calculations**:  Execute each step precisely, ensuring
arithmetic accuracy.  For algebraic problems, solve systematically
(e.g., factorization, substitution, elimination).  For vector
problems, calculate dot products or magnitudes as needed.  For
repeating decimals, set up algebraic equations to express them as
fractions and solve.  For functional problems, verify constraints for
all domain values (e.g., check the equation holds for all x in the
domain).
4.  **Verify the result**:  Confirm that the final answer satisfies
*all* conditions in the problem, including constraints on variable
domains (e.g., positivity, distinctness, integer requirements).  For
vector problems, ensure orthogonality conditions are met by verifying
dot products equal zero.  For functional problems, recheck that the
functional equation holds for all values in the domain.  Recheck
calculations for errors.
5.  **Format the answer**:  Enclose the final numerical result in a
box using \boxed{}.
Provide the answer in the 'answer' field as a mathematical expression
(numerical or symbolic) directly inside \boxed{}.
</system_prompt_1>
```

## 5. GPQA (Qwen3-8B)

**Baseline:**
```
<system_prompt_1>
Given the fields 'question', produce the fields 'answer'.
</system_prompt_1>
```

**Alpha-Evolve:**
```
<system_prompt_1>
Given a multiple-choice question, follow these steps to determine the
correct answer:
1.  **Parse question rigorously**:  Identify the core objective, key
terms, and domain-specific context (e.g., quantum mechanics, genetics,
biochemistry, or thermodynamics).  Extract numerical values, units,
and conceptual constraints (e.g., "refractory tumor," "molecular
weight," or "spin states").
2.  **Evaluate options methodically**:  For each option, assess
scientific validity using domain-specific principles (e.g., quantum
states in qubit space, hyperfine transitions in hydrogen atoms,
or chromosomal mechanisms in meiosis).  Verify alignment with
standard references (e.g., Bloch sphere coordinates, known transition
frequencies, or genetic marker functions).
3.  **Eliminate contradictions**:  Systematically discard options that
violate numerical constraints (e.g., mismatched units or impossible
probabilities), conceptual inconsistencies (e.g., incorrect molecular
interactions), or empirical data (e.g., IR absorption ranges or
reaction mechanisms).
4.  **Prioritize context-specific accuracy**:  Select the option that
satisfies all question conditions while aligning with domain-specific
knowledge (e.g., quantum state vectors, hyperfine splitting formulas,
or epigenetic mechanisms).
5.  **Validate with precision**:  Ensure the answer adheres to
established scientific interpretations, avoids common misconceptions
(e.g., confounding spin values or meiotic errors), and incorporates
critical reference data (e.g., quantum state geometry, nuclear spin
statistics, or chromatin modification patterns).
</system_prompt_1>
```

```
C-Evolve:
Group Prompts 1:
<system_prompt_1>
Given a multiple-choice question, follow these steps to determine the
correct answer:
1.  **Parse the question meticulously**:  Identify the core objective,
key terms, and context (e.g., reaction mechanisms, physical
principles, or biological pathways).  Focus on domain-specific
terminology (e.g., "cyclohexane rings," "electron-positron
annihilation," or "electric field symmetry").
2.  **Analyze all options systematically**:
- For chemical questions, verify reaction mechanisms, stoichiometry,
and product stability (e.g., ozonolysis patterns, acid-catalyzed
dehydration).
- For physics/math questions, check unit consistency, formula
application, and numerical validity (e.g., energy thresholds, electric
field equations).
- For biological questions, confirm pathway fidelity, genetic marker
interactions, or enzymatic specificity (e.g., URA3/LEU2 functions).
3.  **Eliminate contradictory options**:
- Discard choices that violate conservation laws (e.g.,
energy/momentum), thermodynamic principles, or known chemical
behaviors (e.g., ozonolysis products).
- Rule out options inconsistent with provided data (e.g., CMB photon
energies, molecular symmetry groups).
4.  **Select contextually optimal answers**:
- For organic chemistry, prioritize stereochemical accuracy (e.g.,
absolute configurations in drug synthesis).
- For physics, ensure solutions match boundary conditions (e.g.,
electric field inside/outside a shell).
- For biology, validate alignment with empirical data (e.g., drug
discovery workflows, genetic marker interactions).
5.  **Validate against domain-specific standards**:
- Confirm answers match authoritative references (e.g., PDB formats
for protein structures, standard IR spectroscopy ranges).
- Avoid common misconceptions (e.g., confusing "d2h" with "c2h"
symmetry groups).
- For ambiguous questions, explicitly state assumptions and justify
alignment with scientific principles.
</system_prompt_1>
Group Prompts 2:
<system_prompt_1>
Given a multiple-choice question, follow these steps to determine the
correct answer:
1.  **Understand the question**:  Identify the core concept being
tested (e.g., reaction mechanisms, NMR spectroscopy, quantum
mechanics, or molecular orbital theory).  Focus on the specific
context, such as:
- Organic chemistry:  functional group transformations, aromaticity,
or hydrogen bonding.
- NMR spectroscopy:  splitting patterns, coupling constants, or
shielding effects.
- Quantum mechanics:  spin interactions, energy transitions, or
resonance phenomena.
- **Numerical calculations**:  pH, concentration, or equilibrium
problems (e.g., weak acids/bases, buffer calculations).
2.  **Analyze options**:  For each choice, evaluate its validity by:
a) Applying domain-specific knowledge:
- For organic chemistry:  track electron flow, reaction conditions
(e.g., t-BuOK, cupric acetate), and aromaticity rules (e.g., Huckel's
Rule).
- For NMR: determine splitting patterns based on neighboring
hydrogens, coupling constants, and shielding effects.
```

```
– For quantum mechanics: calculate energy levels, spin transitions,
or magnetic interactions.
– For numerical calculations: apply relevant formulas (e.g.,
 \text{pH} = -\log[H^+] , equilibrium expressions).
b) Verify logical consistency with the question's premises (e.g.,
final product formation, reaction pathways, or experimental data).
c) Cross-check against established principles (e.g., Diels-Alder
rules, NMR splitting rules, quantum mechanical spin statistics).
3. **Eliminate distractors**: Rule out options that:
a) Conflict with the question's context (e.g., incorrect reaction
intermediates or unverified side products).
b) Propose mechanisms incompatible with known chemistry (e.g.,
non-aqueous conditions for hydrolysis or invalid resonance
structures).
c) Include irrelevant details (e.g., excess hydrogenation steps in a
one-step reaction or mismatched aromaticity counts).
d) Contain numerical errors (e.g., incorrect pH calculations,
mismatched concentrations).
4. **Select the best answer**: Choose the option that:
– Most directly addresses the question's requirements (e.g., final
product structure, reaction mechanism, or spectral pattern).
– Aligns with both theoretical rules and experimental evidence (e.g.,
Huckel's Rule for aromaticity, NMR splitting for shielding).
– For numerical problems, ensures accurate calculations (e.g., proper
use of  K_a , stoichiometry, or logarithmic relationships).
5. **Verify**: Ensure the answer:
a) Satisfies stoichiometric or structural constraints (e.g., valence,
stereochemistry, or aromaticity).
b) Matches known experimental data (e.g., NMR spectra, reaction
yields, or quantum mechanical predictions).
c) Avoids overgeneralization (e.g., assuming a non-aromatic structure
when Huckel's Rule applies).
d) Accurately solves numerical problems (e.g., correct pH or
concentration values based on given  K_a  and initial concentrations).
e) Explicitly checks for **formula application errors** (e.g., Nernst
equation, Henderson-Hasselbalch, or stoichiometric ratios).
f) Ensures **unit consistency** and **correct significant figures** in
numerical answers.
</system_prompt_1>
```

**Group Prompts 3:**

```
<system_prompt_1>
Given a multiple-choice question, follow these steps to determine the
correct answer:
1. **Identify the domain-specific principles**: Determine whether
the question relates to physics (e.g., redshift, electromagnetic
spectra), chemistry (e.g., molecular structure, reaction mechanisms),
or other fields (e.g., molecular biology, biochemistry), and recall
the **core principles and factual knowledge** specific to that domain.
2. **Analyze each option methodically**:
a. For **physics/astrophysics** questions: Apply formulas like
redshift  z = \frac{\lambda_{observed}}{\lambda_{rest}} - 1 ,
energy equations, or electromagnetic wave properties. **Verify unit
consistency**, **validate formula applicability to the question's
context**, and **cross-check with experimental results or theoretical
models**. Prioritize options that align with **established empirical
data**.
b. For **chemistry** questions: Use reaction mechanisms (e.g.,
metathesis, imine formation), stoichiometric ratios, catalytic
behavior (e.g., TsOH vs. HCl in cyclohexanone-piperidine reactions),
and **common spectroscopic data** (e.g., FTIR: C=O ~    1700 cm^{-1}, C=C
~    1600{1650 cm^{-1}).  **Ensure stoichiometric and structural validity**,
and **prioritize options with known chemical shifts or reaction
patterns**.
```

```
c.  For **molecular biology/biochemistry** questions:  Apply knowledge
of gene regulation (e.g., Tet promoter functionality in fungal species
like *Saccharomyces cerevisiae*), protein engineering (e.g., CAR
T-cell design), and **factual biological systems** (e.g., known
mechanisms of antigen transport through the Golgi).  **Reject options
that contradict established biological or biochemical facts**.
3.  **Compare validity across options**:  Rank options based on:
- **Direct application of domain-specific formulas or facts** (e.g.,
C=O peak position, Tet system presence in specific species)
- **Alignment with experimental observations or theoretical
predictions** (e.g., known gene expression outcomes in model
organisms)
- **Absence of contradictions in the question's conditions** (e.g.,
logical consistency with Tet promoter requirements)
- **Elimination of options with unsupported assumptions or unmet
prerequisites** (e.g., unexplained protein localization, missing
cofactors)
4.  **Select the most robust answer**:  Choose the option supported
by **highest empirical evidence**, **mathematical/chemical accuracy**,
**logical consistency with the question's context**, and **explicit
exclusion of invalid options**.
5.  Final answer must be in the format \boxed{answer}, using the exact
letter (A/B/C/D) corresponding to the correct choice.
</system_prompt_1>
```

### 6. HotpotQA (GPT-4.1-mini)

**Baseline:**
```
<system_prompt_1>
Given the fields 'question', 'passages', produce the fields 'summary'.
</system_prompt_1>
<system_prompt_2>
Given the fields 'question','summary_1', produce the fields 'query'.
</system_prompt_2>
<system_prompt_3>
Given the fields 'question', 'context','passages', produce the fields
'summary'.
</system_prompt_3>
<system_prompt_4>
Given the fields 'question', 'summary_1', 'summary_2', produce the
fields 'answer'.
</system_prompt_4>
```

**Alpha-Evolve:**
```
<system_prompt_1>
Given the fields 'question' and 'passages', produce a concise,
strictly factual, and structured summary focused on key facts that
directly answer the question.  Exclude all peripheral, anecdotal,
legal, or irrelevant details.  Clarify entity membership precisely
when relevant (e.g., distinguish original band members from
collaborators or guests) and include timeline details only if they
directly affect core facts.  When strong evidence supports a probable
partial inference resolving a unclear fact, you may cautiously include
it, clearly framed as "probable" or "likely" to guide retrieval while
avoiding unsupported assumptions.
Organize the summary in these labeled sections:
1.  Key facts:  List confirmed and strongly supported facts directly
answering the question, including any safe probable inferences clearly
noted.
2.  Missing information:  Bullet point concrete factual gaps or
unknowns requiring follow-up retrieval, avoiding vague or broad items.
Omit this section if no missing info remains.
```

```
3. Ambiguities: Specify any contradictory, uncertain, or unresolved
facts found using precise language. Omit this section if none exist.
Avoid repeating the question text or providing opinions. Write
clearly, precisely, and unambiguously to support accurate downstream
steps. This summary is a focused knowledge snapshot for guiding the
next retrieval.
</system_prompt_1>
<system_prompt_2>
Given the fields 'question' and 'summary_1', generate a concise
retrieval query that precisely targets only the explicitly listed
missing information and unresolved ambiguities from 'summary_1'.
- Exclude all facts already known or confirmed.
- Avoid repeating wording verbatim from the original question or
previous summaries to reduce redundancy in retrieval.
- Formulate the query as a minimal, direct natural language question
or focused precise phrase designed to retrieve essential complementary
or clarifying facts that fully resolve the identified gaps.
- Use clear, neutral, and unambiguous wording.
- Avoid verbosity, elaboration, background details, or irrelevant
content.
- If no missing information or ambiguities remain, respond exactly
with: "No further query needed."
The query must maximize precision and relevance to enable effective,
focused second retrieval aimed at thorough knowledge closure.
</system_prompt_2>
<system_prompt_3>
Given the fields 'question', 'context' (the first summary), and
second retrieval 'passages', produce a concise, strictly factual,
and integrated summary synthesizing all evidence from both retrieval
rounds.
- Use 'context' as the baseline.
- Augment it only with new, directly relevant, and complementary facts
found in the second retrieval 'passages'.
- Exclude any speculation, assumptions, or imprecise language.
- Explicitly discard or ignore any irrelevant, off-topic, or
low-quality passages or facts; do not include them in the summary.
- If large parts of second retrieval are irrelevant, note this briefly
as retrieval noise.
Organize the summary into these labeled sections:
1. Updated key facts: Integrate, reconcile, and update confirmed
facts from both retrievals, including safely framed probable
inferences supported by concrete evidence.
2. Resolved missing information: List each previously missing
information item now conclusively answered. Omit this section if
none remain.
3. Persisting ambiguities or contradictions: Specify any remaining
uncertainties, contradictions, or unresolved conflicts clearly and
unambiguously. Omit this section if none persist.
4. [Optional] Irrelevant or rejected information: Briefly note
if significant portions of the second retrieval were irrelevant,
unreliable, or discarded, to indicate retrieval noise.
Avoid repetition, unrelated content, or vague phrasing. The goal
is to fully refine and narrow knowledge gaps, enabling confident,
precise, and unambiguous final answer generation.
</system_prompt_3>
<system_prompt_4>
Given the fields 'question', 'summary_1', and 'summary_2', generate
the final answer that exactly matches the true canonical answer in
both precise content and formatting.
- Output only the final answer text with no explanations, commentary,
citations, or additional context.
```

– If the collected information is insufficient, ambiguous, or
contradictory preventing a confident and exact determination, respond
exactly with:
Answer not found.
Strictly follow these formatting rules by question type:
– Yes/no questions:  output exactly "yes" or "no" (lowercase, no
punctuation).
– Date questions:  output only the exact year, full date, or date
format as appropriate, without any extra words or formatting.
– Entity or name questions:  output the full canonical proper name,
including all middle names, suffixes, official variants, and precise
spellings exactly as expected and verified.  Do not abbreviate or omit
any parts.
– List or multiple-entity answers:  include all required entities
completely and explicitly, formatted in their full canonical
forms (e.g., all official collaborators or members), separated by
appropriate standard delimiters, and in canonical order if known.
– All other question types:  output a concise, precise, and minimal
answer with no filler words, qualifiers, elaboration, or punctuation
beyond what is canonical.
Ensure absolute accuracy, exact canonical form, completeness, and
precise formatting to maximize exact match correctness.  Normalize and
sanitize final answers to conform exactly with canonical expected
forms, avoiding minor mismatches caused by extraneous details,
spelling variants, or punctuation.
</system_prompt_4>

**C-Evolve:**
**Group Prompts 1:**
<system_prompt_1>
Given the fields 'question' and 'passages', produce a concise,
strictly factual, and unambiguous summary synthesizing all key
information directly relevant to answering the question exactly.
Explicitly highlight and retain every important qualifier|such
as geographic locations, temporal references, precise numeric
specifics, and other contextual details|only if they are critical
to an exact final answer match.  For every numeric or quantifiable
fact, standardize the expression to the most precise form supported
by the passages; if only approximations or ranges are present,
clearly flag these as ambiguities and specify the precise numeric
detail needed for exact matching.  Carefully identify, enumerate, and
explicitly flag every missing, ambiguous, contradictory, uncertain, or
insufficient piece of information|including entity or name confusions,
multiple plausible interpretations, partial or inconclusive evidence,
and temporal or ownership uncertainties.  For each flagged issue,
clearly specify precisely what further information or clarification
is required to fully resolve it.  Always include a structured
section titled 'Missing or Ambiguous Information' that itemizes all
identified gaps, uncertainties, or conflicting points.  Avoid any
redundancy by including only new, pertinent evidence that meaningfully
advances understanding and reduces ambiguity.  Do not include any
speculation, assumptions, or unsupported inferences; however, if
indirect inferences are made from evidence, explicitly mark them
with confidence levels (e.g., [INFERENCE, HIGH CONFIDENCE]). When
stating facts that could contribute to the final answer, phrase them
as concisely and minimally as possible, omitting generic or redundant
terms that do not affect the exactness of the answer (e.g., omit words
like \company" if \American fast food" suffices).  This summary must
serve as a clear, evidence-grounded, and comprehensive foundation for
generating a narrowly focused and gap-targeting second retrieval query
that addresses every flagged uncertainty or ambiguity with maximal
precision.
</system_prompt_1>

```
<system_prompt_2>
Given the fields 'question' and 'summary_1', generate one or
more precise, narrowly tailored second retrieval queries that
explicitly and comprehensively target each specific gap, ambiguity,
contradiction, unresolved point, or entity disambiguation need
identified and enumerated in the 'Missing or Ambiguous Information'
section of 'summary_1'. Do not include or reiterate any information
already confirmed or clearly resolved by 'summary_1'. Each query must
be focused exclusively on retrieving complementary, clarifying, or
disambiguating evidence that directly resolves a flagged uncertainty
or missing qualifier in order to enable a final answer that exactly
matches the ground truth. Avoid general, broad, or unrelated topics
that could introduce noise or irrelevant results. Ensure that each
query is as minimal as possible while fully covering the targeted
ambiguity or information gap. Organize queries to maximize efficiency
and effectiveness by minimizing redundancy or overlap. Phrase queries
using terminology and key terms aligned closely with the original
question's semantics to maintain retrieval precision and relevance.
If multiple ambiguities exist, create separate, distinct queries for
each and prioritize them to enable staged or parallel retrievals.
</system_prompt_2>
<system_prompt_3>
Given the fields 'question', 'context' (the first summary), and
'passages' from the second retrieval, produce a concise, logically
coherent, and comprehensive summary integrating all available
evidence. Rigorously compare, cross-validate, and reconcile facts
and qualifiers from both the first summary and second retrieval
passages, explicitly addressing and resolving any contradictions,
ambiguities, or discrepancies by assessing internal consistency,
confidence levels, source reliability, and directness of evidence.
If indirect inferences are made, flag them clearly with appropriate
confidence annotations (e.g., [INFERENCE, MODERATE CONFIDENCE]).
Include every key qualifier and disambiguating detail essential
to produce an exact and fully specified final answer. If any
conflicting information remains unresolved after thorough analysis,
clearly and explicitly enumerate the residual uncertainties in a
dedicated 'Residual Uncertainties' section, and if feasible, specify
what further information or data would be required to definitively
resolve them. Do not repeat or restate conclusively confirmed
facts without integration or refinement; instead, synthesize them
concisely into an integrated fact set. When integrating facts,
phrase all summaries as concisely as possible, avoiding generic or
superfluous terms that do not affect final answer precision. This
second summary must constitute a definitive, meticulously checked,
and confidence-calibrated factual basis that directly enables final
answer generation strictly matching the true answer without deviation.
Confirm that all previously flagged gaps are addressed or explicitly
declared unresolved.
</system_prompt_3>
<system_prompt_4>
Given the fields 'question', 'summary_1', and 'summary_2', produce
the final answer text that exactly matches the expected true answer
word for word with no deviations whatsoever. The answer must be
strictly concise, unambiguous, and dry without any explanation,
filler, opinion, speculation, or extra sentences. Do not paraphrase,
round, generalize, or elaborate under any circumstance; output only
the exact text that yields an exact match to the ground truth label.
Include all critical qualifiers only if and strictly as they appear
in the ground truth answer|no additions or omissions. Exclude any
additional context, background information, or details beyond the true
answer content. Before outputting, rigorously verify that the answer
completely and exactly resolves the question with respect to wording,
format, punctuation, and all critical qualifiers as needed for an
```

```
exact evaluation match.  Follow case formatting conventions exactly
as expected for the true answer (e.g., lowercase 'no' if the answer
is 'no').  If any uncertainty, ambiguity, or insufficient evidence
persists preventing an exact match, respond with the exact phrase:
"Insufficient data to provide the exact answer." Do not output
anything else.  Only produce the final answer after all verifications;
do not hallucinate or produce answers from incomplete or contradictory
summaries.
</system_prompt_4>
Group Prompts 2:
<system_prompt_1>
Given the fields 'question' and 'passages', produce a concise,
strictly factual, and structured summary focused on key facts that
directly answer the question.  Exclude all peripheral, anecdotal,
legal, or irrelevant details.  Clarify entity membership precisely
when relevant (e.g., distinguish original band members from
collaborators or guests) and include timeline details only if they
directly affect core facts.  When strong evidence supports a probable
partial inference resolving a unclear fact, you may cautiously include
it, clearly framed as "probable" or "likely" to guide retrieval while
avoiding unsupported assumptions.
Organize the summary in these labeled sections:
1.  Key facts:  List confirmed and strongly supported facts directly
answering the question, including any safe probable inferences clearly
noted.
2.  Missing information:  Bullet point concrete factual gaps or
unknowns requiring follow-up retrieval, avoiding vague or broad items.
Omit this section if no missing info remains.
3.  Ambiguities:  Specify any contradictory, uncertain, or unresolved
facts found using precise language.  Omit this section if none exist.
Avoid repeating the question text or providing opinions.  Write
clearly, precisely, and unambiguously to support accurate downstream
steps.  This summary is a focused knowledge snapshot for guiding the
next retrieval.
</system_prompt_1>
<system_prompt_2>
Given the fields 'question' and 'summary_1', produce a concise,
precise retrieval query that exclusively targets the explicitly listed
missing information, unresolved ambiguities, or hypotheses flagged in
'summary_1' which are required to fully answer the question.
- Do not include any facts already confirmed or known.
- Avoid duplicating exact phrasing from the question or prior
summaries to reduce redundancy.
- Formulate the query as a minimal, direct natural language question
or a focused, precise search phrase.
- Use clear, neutral, and unambiguous language focused solely on
resolving identified knowledge gaps or testing hypotheses.
- Do not add background, elaboration, or unrelated details.
- If 'summary_1' contains no missing info, ambiguities, or hypotheses
that prevent answering, respond exactly with:  "No further query
needed."
This query should maximize retrieval precision and relevance to close
knowledge gaps effectively in the second retrieval.
</system_prompt_2>
<system_prompt_3>
Given the fields 'question', 'context' (the first summary), and second
retrieval 'passages', produce a concise, strictly factual summary
that integrates evidence from both retrieval rounds to provide the
most complete and reconciled knowledge snapshot for final answer
generation.
- Use 'context' as the foundation, expanding only with newly
discovered, directly relevant, and well-supported facts from the
second retrieval.
```

```
- Cross-check, reconcile, and resolve conflicts or inconsistencies
across retrievals, explicitly stating resolutions and noting
confidence.
- Where relevant, integrate partial or probabilistic inferences
supported by evidence, clearly flagged with confidence levels.
- Do not include speculation beyond what can be cautiously inferred
nor vague language.
- Exclude irrelevant, low-quality, or off-topic passages or facts
unless briefly summarizing their exclusion as retrieval noise.
- If a significant portion of the second retrieval is irrelevant or
discarded, note this succinctly as retrieval noise.
Organize into these sections with clear bullet points:
1.  Updated key facts:  Fully integrated and reconciled confirmed
facts and strongly supported probable inferences from both retrievals,
clearly marked by confidence.
2.  Resolved missing information:  List previously missing or
hypothesized details now conclusively answered or disproven.  Omit
if none.
3.  Persisting ambiguities, hypotheses, or contradictions:  Clearly
list any remaining unresolved uncertainties, conflicts, or unconfirmed
hypotheses.  Omit if none.
4.  [Optional] Irrelevant or rejected information:  Brief note if
substantial content from second retrieval was discarded as retrieval
noise.
Avoid repetition, vague phrasing, or unrelated material.
This summary must aim to fully close knowledge gaps and ambiguities to
support exact and confident final answer generation.
</system_prompt_3>
<system_prompt_4>
Given the fields 'question', 'summary_1', and 'summary_2', generate
the final answer that exactly matches the true canonical answer in
both precise content and formatting.
- Output only the final answer text with no explanations, commentary,
citations, or additional context.
- If the collected information is insufficient, ambiguous, unresolved,
or contradictory preventing a confident and exact determination,
respond exactly with:
Answer not found.
Strictly adhere to these formatting rules by question type:
- Yes/no questions:  output exactly "yes" or "no" (lowercase, no
punctuation).
- Date questions:  output only the exact year, full date, or precise
date format, with no extra words or punctuation.
- Entity or name questions:  output the full canonical proper name,
including all middle names, suffixes, honorifics, official variants,
and exact spellings exactly as expected.  Do not abbreviate or omit
any parts.
- List or multiple-entity answers:  include all required entities
explicitly, using their full canonical forms, separated by standard
delimiters, and in canonical order if known.
- Numeric answers (counts, amounts):  spell out numbers (e.g., "two"
instead of "2") in standard canonical form unless the canonical answer
format explicitly requires digits.
- All other question types:  output a concise, precise, minimal answer
with no filler, qualifiers, elaboration, or extraneous punctuation
beyond what is canonical.
Ensure absolute accuracy, canonical form, completeness, and precise
formatting to maximize exact match scoring.
Apply normalization and sanitization rigorously to avoid minor
mismatches caused by extraneous details, spelling variants,
punctuation, or formatting differences.
</system_prompt_4>
Group Prompts 3:
```

```
<system_prompt_1>
You are given the fields 'question' and 'passages' (a batch of
retrieved documents). Generate a concise, coherent summary that
captures all key facts directly relevant to answering the question
precisely and unambiguously. Present the direct candidate answer
or finding upfront in a clear, minimal sentence or phrase exactly
matching the expected answer type and canonical format (e.g., a
date in canonical form, a full name, or a simple "yes" or "no").
If the question expects a yes/no answer, respond with "yes" or "no"
directly unless ambiguity exists. If multiple valid candidates
exist and the question suggests plurality, include all clearly
distinguished; if singular, select the most authoritative or canonical
answer with justification. Explicitly state any information missing
from the passages that prevents a definitive answer, using clear,
unambiguous language such as "No information about X is provided."
Identify and explicitly describe any ambiguities, contradictions,
or conflicting evidence found within the passages, summarizing
their nature succinctly without hedging. When the question involves
temporal, ownership, authorship, or attribution nuances, clarify the
relevant time frames or roles, prioritizing authoritative sources, and
distinguishing founder versus current ownership or individual versus
group authorship as appropriate. For questions involving comparative
or multi-part conditions, explicitly state whether sufficient evidence
exists to conclude or if information is insufficient, summarizing
asymmetries. Avoid including extraneous or background details; focus
only on information essential to answering the question. Do not
speculate or introduce external knowledge beyond the passage content.
Verify and present named entities, organizations, or official terms in
their current, official, or canonical forms; if uncertain, note this
as missing information to be retrieved. This summary will support
downstream query refinement and answer accuracy, so provide clear,
decisive facts with precise, minimal, and non-redundant wording.
</system_prompt_1>
<system_prompt_2>
You are given the fields 'question' and 'summary_1' (the initial
summary of retrieved passages). Analyze the summarized information
along with any explicitly identified missing details, ambiguities, or
contradictions in summary_1. Generate a precise, focused retrieval
query that directly targets the minimal missing information or
resolves the noted uncertainties. Ensure the query accurately
captures the core semantic intent of the original question,
emphasizing the retrieval of new, complementary, and authoritative
evidence specifically addressing gaps, temporal or attributional
nuances, or conflicting points. Avoid vague or overly broad query
terms; prioritize clarity and specificity to maximize downstream
retrieval relevance and improve answer completeness and exact
correctness.
</system_prompt_2>
<system_prompt_3>
You are given the fields 'question', 'context' (the first summary),
and 'passages' (a new batch of retrieved documents). Integrate and
synthesize all information from the first summary and the new passages
into a concise, coherent summary that explicitly resolves or clearly
articulates previously identified ambiguities, contradictions, or
missing information. Provide clear, decisive conclusions supported
by the combined evidence, ensuring all named entities and relevant
data are presented in their full canonical and official forms.
Where conflicts in evidence persist, clearly describe the nature
and implications of these conflicts without hedging, prioritizing
the most authoritative info. Include any necessary reasoning,
temporal context, or comparisons relevant to precisely answering
the question. Avoid redundancy and exclude information not strictly
necessary to support the final answer. This summary should maximize
```

```
completeness, clarity, and precision to effectively guide the final
answer generation.
</system_prompt_3>
<system_prompt_4>
You are given the fields 'question', 'summary_1', and 'summary_2'
(the two summaries from retrieval hops). Produce the final answer
as a single, concise text string that exactly matches the true
expected answer in canonical and official form|no explanations,
clarifications, or additional context. Prioritize exact match,
clarity, and unambiguity, ensuring the answer is definitive, directly
supported by the provided summaries, and appropriately formatted to
the expected answer type (e.g., full names, dates in canonical format,
yes/no). Do not add any information beyond what is supported by the
summaries or infer beyond the provided evidence. If the summaries do
not contain sufficient evidence to produce a confident answer, respond
exactly with "Insufficient information."
</system_prompt_4>
```

### 7. IFBench (GPT-4.1-mini)

**Baseline:**
```
<system_prompt_1>
Respond to the query
</system_prompt_1>
<system_prompt_2>
Ensure the response is correct and adheres to the given constraints.
Your response will be used as the final response.
</system_prompt_2>
```

**Alpha-Evolve:**
```
<system_prompt_1>
You are given a user instruction. Provide a **direct, concise,
and precise** answer that **strictly** fulfills **every explicitly
stated** requirement in the instruction, including content, language,
formatting, style, structure, numbering, punctuation, and length
constraints. Match these **exactly and without deviation**. Do
**not** include explanations, rationales, clarifications, or any
information beyond what is explicitly requested. Address **all parts
and subparts** of the instruction fully, avoiding assumptions or
interpretations beyond the literal text. Your output must be **only**
the answer that exactly matches the user's request; do **not** add any
commentary, prefatory remarks, or extra content.
</system_prompt_1>
<system_prompt_2>
Given the user instruction and the answer generated in Step 1, perform
a thorough and critical review of the answer for any mistakes,
omissions, contradictions, or nonconformities with **all explicit
and implicit requirements** in the instruction. Correct all such
issues to ensure **full and exact compliance** with every instruction
detail, including content accuracy, formatting, style, length,
vocabulary restrictions, repetition rules, language, and required
structure. Enhance the answer's clarity, coherence, completeness, and
adherence without adding explanations, rationales, or meta-commentary.
Output **only** the final, polished response that best fulfills
every aspect of the user's instruction, including any subtle or
implied requirements, guaranteeing maximal alignment with the original
instruction.
</system_prompt_2>
```

**C-Evolve:**
**Group Prompts 1:**
```
<system_prompt_1>
```

```
When given the instruction, produce a direct and exact answer that
fully addresses every explicit requirement without omitting any part.
Follow all specified formatting rules, language or style constraints,
mandatory inclusions or exclusions of words or phrases, and precise
output structure exactly as stated.  Do not include explanations,
reasoning, or additional commentary|your output should be strictly
the answer as required.  If multiple parts or complex directives are
present, answer each completely but concisely, adhering tightly to the
user's exact wording, formatting, and style instructions.  Prioritize
literal and explicit compliance over paraphrasing, elaboration, or
inferred interpretation, ensuring no deviations from the instruction's
demands.
</system_prompt_1>
<system_prompt_2>
Thoroughly examine the Step 1 answer alongside the original
instruction to identify any errors, omissions, ambiguities, or
failures to meet the instruction's requirements.  Correct all such
issues by enhancing the response to fully and precisely satisfy every
explicit requirement, constraint, formatting rule, and specified
style.  Refine and polish the answer for clarity, coherence,
completeness, and natural fluency, while strictly adhering to all
exact wording mandates, required phrases, forbidden words, and
formatting instructions.  The final output must be a perfectly
calibrated, comprehensive, and ready-to-deliver response that complies
flawlessly with the user's instruction, leaving no room for ambiguity
or error.
</system_prompt_2>
```

**Group Prompts 2:**

```
<system_prompt_1>
Carefully and thoroughly read the user's instruction.  Identify
and understand ALL requirements, conditions, constraints, including
formatting, style, language, punctuation, keywords, length limits, or
any other explicit elements.  Produce a direct, precise, and faithful
answer that strictly adheres to every instruction detail, without
omissions, additions, or deviations.  Do NOT include explanations,
reasoning, or commentary unless the instruction explicitly requests
them.  Ensure the response fully respects requested output structure
and style.  Your output must solely and exactly fulfill the
instruction as given.
</system_prompt_1>
<system_prompt_2>
Carefully re-examine the answer generated in Step 1 together with the
original user instruction.  Detect and correct any errors, omissions,
inconsistencies in content, format, style, language, punctuation,
or failure to meet explicit instruction constraints such as required
keywords, forbidden words, exact counts of sentences or paragraphs, or
specified structure.  Refine the response for clarity, completeness,
coherence, and overall quality.  The final answer must be a polished,
comprehensive, and perfectly compliant version meeting ALL instruction
requirements without changing the intended meaning.  Confirm that
formatting, style, and all explicit demands from the user instruction
and Step 1 output are strictly fulfilled.
</system_prompt_2>
```

**Group Prompts 3:**

```
<system_prompt_1>
You are given a user instruction.  Your task is to generate a precise,
direct, and thoroughly complete response that fully satisfies
every explicit and implicit requirement stated in the instruction.
This includes all aspects such as factual accuracy, computational
correctness, required language, formatting conventions, stylistic
tone, organizational structure, numbering schemes, punctuation rules,
response length, mandatory keyword usage, section and paragraph
counts, and any other details.  Your output must be solely the direct
```

```
answer requested|do not include explanations, rationale, commentary,
or any extraneous information.  Rigorously adhere to all formatting,
style, and structural requirements specified in the instruction unless
they are explicitly reserved for Step 2 calibration.  Ensure your
response is clear, logically coherent, unambiguous, concise, and
exhaustively addresses every part of the instruction without omissions
or assumptions.
</system_prompt_1>
<system_prompt_2>
Given the user instruction and the Step 1 answer, carefully analyze
the response to identify and correct all errors, missing components,
formatting or style issues, length constraints, redundant or repeated
wording, violations of language or keyword constraints, mismatches
in section or paragraph count, or any other failures to fully meet
explicit and implicit instruction requirements.  Improve, refine,
and polish the Step 1 answer to produce a final response that is
error-free, fully compliant, maximally clear, coherent, precise,
and optimized to satisfy or exceed all user requirements.  Provide
only the finalized, calibrated answer; do not include explanations,
justifications, commentary, or any content beyond what the final
instruction demands.
</system_prompt_2>
```

## 8. HoVer (GPT-4.1-mini)

**Baseline:**
```
<system_prompt_1>
Given the fields 'claim', 'passages', produce the fields 'summary'.
</system_prompt_1>
<system_prompt_2>
Given the fields 'claim', 'summary_1', produce the fields 'query'.
</system_prompt_2>
<system_prompt_3>
Given the fields 'claim', 'context', 'passages', produce the fields
'summary'.
</system_prompt_3>
<system_prompt_4>
Given the fields 'claim', 'summary_1', 'summary_2', produce the fields
'query'.
</system_prompt_4>
```

**Alpha-Evolve:**
```
<system_prompt_1>
You are given the fields 'claim' and 'passages' retrieved in the first
retrieval step.
Your task is to produce the field 'summary' as a concise (max
3 sentences), coherent synthesis that balances strict factual
accuracy and proactive evidence bridging to support effective claim
verification.
- Explicitly identify and disambiguate all key entities, attributes,
and relationships mentioned or implied in the claim, especially when
multiple similarly named candidates appear.  Rank these candidates by
relevance or likelihood using passage evidence.
- When essential entities or relationships referenced by the claim are
missing from the passages, infer or hypothesize plausible identities
or links strictly based on provided evidence without unsupported
speculation, to proactively fill critical information gaps needed
for verification.
- Clarify or normalize ambiguous or unusual claim terms by referencing
possible standard equivalents or synonyms found in the passages where
justifiable.
```

```
- Integrate complementary facts strictly from the passages, combining
evidence logically to reduce unresolved gaps while maintaining clarity
and confidence levels.
- Clearly state all remaining ambiguities, missing evidence, or
unresolved points that require further targeted investigation,
distinguishing them from conjecture.
- Exclude extraneous or peripheral information unrelated to directly
verifying the claim components.
- Avoid redundancy, excessive detail, or unsupported speculation;
include only facts grounded in the retrieved passages or justified
bridging inferences.
- Ensure the summary is precisely claim-focused, highlights evidence
strengths and gaps, and prepares the next query to actively resolve
missing or ambiguous claim facets.
Provide a summary that both synthesizes entities and core evidence and
recommends key unresolved aspects for follow-up.
</system_prompt_1>
<system_prompt_2>
You are given the fields 'claim' and 'summary_1'.
Your task is to generate the field 'query' that pinpoints and actively
targets all unresolved, ambiguous, or partially covered claim aspects
revealed or hypothesized in 'summary_1', refining and sharpening the
original claim semantics for precise second retrieval.
- Formulate a concise, compound query that explicitly includes all
necessary entities, attributes, relationships, or qualifiers needed
to resolve remaining ambiguities, disambiguate similar or competing
entities, or fill critical evidential gaps noted or inferred in
'summary_1'.
- Where helpful, incorporate negations, exclusions, or temporal
qualifiers to avoid conflation of similar entities or contexts,
enhancing retrieval specificity.
- Prioritize maximal specificity and logical combination of multiple
claim facets (e.g., entity + attribute + contextual relation) to
minimize retrieval noise and maximize relevance.
- Avoid vague, overly broad, or generic queries that do not directly
resolve outstanding ambiguities or missing evidence necessary for
claim verification.
- Utilize all known identifiers, distinctive details, temporal
markers, or normalized terms surfaced or hypothesized in 'summary_1'
to improve query precision without introducing unsupported facts.
Provide only the final query text, without explanation, formatting, or
additional content.
</system_prompt_2>
<system_prompt_3>
You are given the fields 'claim', 'context' (the first summary), and
'passages' retrieved in the second retrieval step.
Your task is to produce the field 'summary' as a concise (max 3
sentences), integrated synthesis that explicitly compares, reconciles,
and combines the new passage evidence with 'context', focusing tightly
on claim verification and information completeness.
- Summarize only the most relevant and novel information that
supports, refutes, clarifies, or resolves ambiguous or incomplete
points from the first summary, including any newly discovered evidence
or hypothesized clarifications.
- Where inconsistent, overlapping, or competing evidence exists,
explicitly weigh, rank, or differentiate candidate entities,
interpretations, or claim facets by evidential strength, likelihood,
or confidence.
- Clearly identify and articulate all remaining unresolved aspects,
contradictions, ambiguities, or uncertainties critical to the final
truth evaluation of the claim, explicitly stating what has been
resolved or narrowed down since the first summary.
```

```
- Exclude redundant, peripheral, or background information not
directly relevant to claim verification or query focus.
- Maintain clarity, factual accuracy, and logical coherence to guide
focused, effective final query generation.
- If multiple candidate entities, interpretations, or claim facets
remain ambiguous, list them explicitly to inform the final retrieval
focus.
- Include explicit confidence or evidence strength markers where
appropriate to indicate the reliability of conclusions or ongoing
gaps.
Provide a precise, integrative, and confidence-aware summary that
highlights evidence progress and remaining verification needs.
</system_prompt_3>
<system_prompt_4>
You are given the fields 'claim', 'summary_1', and 'summary_2' (the
two prior summaries).
Your task is to generate the field 'query' that precisely targets
all remaining unresolved, weakly supported, or ambiguous aspects
identified by synthesizing 'summary_1' and 'summary_2', aiming for a
final, comprehensive, and focused retrieval.
- Construct a concise, compound query that integrates all outstanding
claim facets, employing logical operators, exclusions, temporal
qualifiers, or specific attribute combinations as needed to maximize
retrieval relevance and minimize noise or redundancy.
- Explicitly prioritize disambiguation among similar or competing
candidate entities or interpretations, emphasizing terms that promote
retrieving novel or unique passages not covered in prior retrieval
steps to improve answer coverage and reduce redundancy.
- Avoid vague, generic, or overly broad queries that risk retrieving
irrelevant or overlapping passages.
- Use exact entity names, attributes, temporal markers, normalized
terms, or relationships surfaced in previous summaries, strictly
without introducing unsupported facts or assumptions.
- Where useful, prioritize query terms that target claim exclusivity,
identity or name disambiguation, term normalization, or completeness
gaps identified in prior summaries.
- Ensure the query is optimized to retrieve passages offering maximal
novel evidence for final claim verification.
Provide only the final query string without explanations or
formatting.
</system_prompt_4>
```

**C-Evolve:**
**Group Prompts 1:**
```
<system_prompt_1>
You are given the fields 'claim' and 'passages' which have been
retrieved as the first batch relevant to the claim.
Your goal is to produce a concise, factual, and coherent 'summary'
strictly focused on verifying all claim components, including all
relevant entities, roles, dates, and relations mentioned.
**Instructions:**
- Limit the summary to 3-4 well-structured sentences.
- Explicitly confirm, contradict, or report absence of evidence for
each distinct factual element in the claim (e.g., individuals, events,
roles, dates). If a component is not found, state "no direct evidence
found."
- Identify and enumerate alternative plausible entities or hypotheses
relevant to the claim especially if initial evidence is missing or
contradictory. For example, list other possible directors or films
connected to involved actors.
- Emphasize temporal, positional, and role-related consistencies or
discrepancies with clear statements.
```

```
- Synthesize all relevant evidence, integrating multiple claim parts,
and exclude unrelated or duplicate information.
- Avoid vague or hedging language; be precise and clear about
confirmations, contradictions, and gaps.
- Explicitly list any ambiguities or unresolved points and recommend
concrete focuses or targets for the next retrieval query (e.g.,
alternative entities, missing films, conflicting dates).
- Include a structured summary in terms of:  *Confirmed facts*,
*Contradicted claims*, *Ambiguities or unknowns*, *Evidence gaps*,
and *Next retrieval focus*.
- The summary should serve as a comprehensive basis for refining
search queries that aim to close evidence gaps and resolve
contradictions.
- Output the field:  'summary'.
</system_prompt_1>
<system_prompt_2>
You are given 'claim' and 'summary_1' (a concise summary of the first
retrieval).
Your task is to produce a clear, focused, and precise 'query' to
guide the second retrieval aimed at resolving all identified gaps,
ambiguities, and contradictions from 'summary_1'.
**Instructions:**
- Carefully analyze 'summary_1', especially the *Ambiguities*,
*Evidence gaps*, and *Next retrieval focus* sections, to identify
which claim components or alternative entities need targeted
exploration.
- Formulate a query that explicitly targets these unresolved aspects
using specific qualifiers such as exact names, dates, roles, film
titles, or locations.
- Include alternative hypotheses or related entities if suggested
by 'summary_1' to broaden evidence collection and confirm or refute
competing possibilities.
- Avoid vague, generic, or overly broad terms; do not reuse prior
queries verbatim but refine them to maximize relevance and new
evidence discovery.
- Ensure the query preserves the original claim's core semantics but
focuses the retrieval on bridging gaps and resolving contradictions.
- Output the field:  'query'.
</system_prompt_2>
<system_prompt_3>
You are given the fields 'claim', 'context' (which is 'summary_1'),
and 'passages' (the second retrieval batch).
Your goal is to produce a concise, factual, and coherent 'summary'
that meaningfully enriches or revises the evidence landscape regarding
the claim beyond what is captured in 'context'.
**Instructions:**
- Limit the summary to 3-4 well-structured sentences that present new,
relevant evidence supporting, refining, or contradicting individual
components of the claim.
- Clearly integrate and compare this new evidence against 'context'
and the claim, explicitly highlighting confirmations, contradictions,
or emerging ambiguities between summaries.
- Explicitly account for multi-entity relationships and temporal,
role, and positional consistencies or inconsistencies that emerge.
- If alternative hypotheses or entities were proposed in 'context',
report whether the new passages confirm, refute, or further complicate
them.
- Avoid restating unchanged information from 'context' or introducing
off-topic details.
- Summarize remaining gaps, ambiguities, or evidence still required
for thorough claim verification.
```

```
- Structure the output with discrete elements:  *New confirmed facts*,
*New contradictions or refinements*, *Remaining ambiguities or gaps*,
and *Recommendations for next retrieval focus*.
- Emphasize clarity, precision, and relevance to guide the final query
generation effectively.
- Output the field:  'summary'.
</system_prompt_3>
<system_prompt_4>
You are given 'claim', 'summary_1', and 'summary_2' (summaries from
previous retrievals).
Your task is to produce a final, narrowly targeted 'query' designed to
retrieve definitive, authoritative evidence that conclusively confirms
or refutes the claim by resolving all remaining contradictions, gaps,
and ambiguities identified in the combined summaries.
**Instructions:**
- Thoroughly analyze both summaries, focusing on unresolved questions,
contradictions, ambiguous or missing links crucial for final claim
verification.
- Formulate a precise, refined query that explicitly targets these
outstanding issues using specific qualifiers|such as exact personal
names, titles, dates, roles, film titles, locations, or other
contextual identifiers|to maximize retrieval relevance and precision.
- Include any alternative entities or hypotheses still unresolved to
ensure comprehensive coverage towards conclusive evidence.
- Avoid reusing broad or vague terms from prior queries without
refinement; strive for the narrowest effective scope that can yield
definitive passages.
- The query must be designed to enable retrievals that provide
final closure on the claim's truth status, isolating any remaining
uncertainty.
- Output the field:  'query'.
</system_prompt_4>
```

**Group Prompts 2:**

```
<system_prompt_1>
You are given the fields 'claim' and 'passages' retrieved in the first
retrieval step.
Your task is to produce the field 'summary' as a concise (max 3
sentences), coherent synthesis of the key information strictly
relevant to assessing the claim.
- Summarize only the most pertinent facts from the passages that
support or contradict the claim.
- Avoid repetition, duplication, and extraneous details unrelated to
the claim specifics.
- Focus on factual accuracy and clarity to enable precise query
generation in the next step.
- Do not include new information not present in the passages.
Provide a clear and concise summary suitable for query refinement.
</system_prompt_1>
<system_prompt_2>
You are given the fields 'claim' and 'summary_1'.
Your task is to generate the field 'query' that captures the core
semantics of the claim refined by insights from 'summary_1' to improve
retrieval focus.
- Formulate a precise and focused query that directly targets missing,
ambiguous, or critical aspects suggested by the first summary.
- Avoid mere paraphrasing; instead, narrow or clarify the claim-based
information to retrieve more relevant documents in the next retrieval
step.
- The query should be concise, specific, and constructed for effective
database search.
Provide the generated query only.
</system_prompt_2>
<system_prompt_3>
```

```
You are given the fields 'claim', 'context' (which is the first
summary), and 'passages' retrieved in the second retrieval step.
Your task is to produce the field 'summary' as a concise (max 3
sentences), integrated synthesis of the second retrieval passages,
focusing on information that complements or clarifies the first
summary regarding the claim.
- Summarize the most relevant evidence supporting or refuting the
claim, emphasizing new or refined facts emerging from these passages.
- Avoid redundant or irrelevant information.
- Your summary should prepare for precise query generation in the next
step, highlighting any remaining uncertainties or details needed.
Ensure factual clarity and avoid repetition.
</system_prompt_3>
<system_prompt_4>
You are given the fields 'claim', 'summary_1', and 'summary_2' (the
two prior summaries).
Your task is to generate the field 'query' to retrieve the final set
of passages relevant to claim verification.
- Construct a focused and precise query that targets unresolved
aspects or key evidence gaps identified by synthesizing 'summary_1'
and 'summary_2'.
- The query should be specific, concise, and designed to retrieve
non-duplicate, highly relevant passages that maximize coverage of the
true evidence needed to confirm or refute the claim.
- Avoid broad, vague, or redundant queries.
Provide only the final retrieval query.
</system_prompt_4>
Group Prompts 3:
<system_prompt_1>
You are given a 'claim' and a batch of retrieved 'passages' related to
it.
Your task is to produce a concise, coherent summary (up to 5
sentences) that synthesizes unique, claim-critical facts directly
relevant to verifying or refuting the claim.
- Aggressively exclude all tangential, duplicate, off-topic, or
unrelated information, especially facts involving similarly named
or ambiguous entities not related to this claim.
- Explicitly identify and list any such unrelated or ambiguous
entities excluded, to demonstrate clear disambiguation.
- Explicitly link all entities, numeric values, dates, and geographic
details to exact components of the claim for unambiguous clarity.
- Organize your summary strictly into three labeled sections:
(1) Facts Supporting the Claim,
(2) Facts Contradicting the Claim, and
(3) Remaining Ambiguities, Missing Information, or Conflicts.
- Use precise, focused language that highlights insightful nuances and
unresolved gaps.
- End with a clear, brief statement of the current evidential status:
whether the claim is supported, contradicted, or remains unresolved
based on this evidence.
- Your summary should serve as a clean factual foundation free
of noise, enabling sharp, targeted query generation for the next
retrieval step.
</system_prompt_1>
<system_prompt_2>
You receive a 'claim' and the first summary ('summary_1') synthesizing
initial evidence.
Generate a single, concise (one or two sentences), highly focused
query that targets the single most critical unresolved information
gap, ambiguity, or contradiction explicitly identified in 'summary_1'.
- The query must directly seek evidence to resolve that specific gap
or verify uncertain claim components, avoiding broad, general, or
multi-faceted questions.
```

```
- Do not repeat prior queries or restate the claim without narrowing
focus; aim solely to reduce uncertainty effectively.
- Formulate a query immediately actionable for database retrieval,
emphasizing precise entity linkages, numeric or date specifics, and
exact fact confirmations (e.g., "Which actress starring in *Money
Monster* also appeared in *One Percent More Humid*?" or "What is the
exact eldership subdivision containing Pervalka?").
- If 'summary_1' indicates no missing or ambiguous information,
produce a query that seeks explicit confirmation or refutation of
the claim's main assertion.
- Keep queries clear, unambiguous, and free of unnecessary verbosity
or multiple unrelated parts.
</system_prompt_2>
<system_prompt_3>
Given a 'claim', the first summary ('context' = 'summary_1'), and a
new batch of passages retrieved in the second retrieval:
- Produce a concise, coherent summary (up to 5 sentences) synthesizing
new, unique, directly relevant facts that were not covered in
'context'.
- Focus deeply on evidence that supports, contradicts, or further
disambiguates the claim and its key entities.
- Explicitly link all entities, numeric data, dates, or geographic
details to exact claim components.
- Exclude all facts already summarized in 'context' and remove any
tangential, redundant, irrelevant, or off-topic information.
- Clearly identify and list any passages or facts associated with
unrelated or ambiguous entities that have been excluded, especially if
already settled.
- Structure the summary into three labeled sections:
(1) Newly Supporting Facts,
(2) Newly Contradicting Facts, and
(3) Persisting Ambiguities, Missing Details, or Conflicts.
- End with a concise statement specifying which uncertainties
remain, prioritizing those that the final query should target for
verification.
- Ensure this summary provides a precise factual update that sharpens
the evidence base for final query formulation.
</system_prompt_3>
<system_prompt_4>
Given a 'claim' and the two prior summaries ('summary_1' and
'summary_2'), generate a single, sharply focused, and precise query
(one or two sentences) for the final retrieval step.
- Integrate evidence from both summaries to identify the single most
critical unresolved, ambiguous, or contradictory aspect of the claim.
- The query must explicitly target these remaining gaps or key
verification points, aiming to conclusively confirm or refute the
essential relationships or facts.
- Avoid repeating previous queries or broad restatements of the claim;
prioritize novel, decisive retrieval questions.
- Prefer queries that articulate explicit entity-to-claim mappings,
include precise numeric or geographic details, or request direct
confirmation or negation.
- Ensure the query is unambiguous, concise, and immediately usable for
retrieval.
- If the summaries indicate no remaining major ambiguities or
missing information, produce a final query that seeks authoritative
adjudication of the claim's overall truth or falsehood status.
</system_prompt_4>
```

## 9. MATH (GPT-4.1-mini)

**Baseline:**
<system_prompt_1>
Given the fields 'problem', produce the fields 'answer'.
</system_prompt_1>

**Alpha-Evolve:**
<system_prompt_1>
You will be given a mathematical problem in the input field named 'problem'. Your task is to produce the exact final answer by solving the problem step-by-step. Follow these instructions carefully:
– Provide a clear, logical, and fully justified step-by-step solution that guides from the problem statement directly to the final answer.
– Show all necessary calculations and explanations in sufficient detail to ensure full correctness and completeness.
– The final answer must be explicitly presented at the end, enclosed precisely in LaTeX \boxed{} notation, for example: \boxed{<answer>}, where <answer> is the exact final result.
– Make certain that the content inside \boxed{} matches exactly the mathematically correct and exact final answer|no approximation, rounding, or ambiguity is allowed.
– Do not include any irrelevant information, extraneous commentary, or outputs beyond the solution and the final boxed answer.
Your response will be evaluated for exact mathematical correctness and clarity. Prioritize accuracy, thoroughness, and clear reasoning to ensure the final answer is fully reliable.
</system_prompt_1>

**C-Evolve:**
**Group Prompts 1:**
<system_prompt_1>
You will be given a math problem as input in the field 'problem'. Your task is to produce a clear, logically structured, and rigorously accurate solution that directly leads to and includes the final answer.
Follow these instructions precisely:
1. Read the problem carefully and ensure full understanding before starting the solution.
2. Provide detailed, step-by-step reasoning with all necessary mathematical justifications, formulas, and calculations. Each step must be mathematically valid, complete, and clear.
3. Simplify all expressions fully, unless the problem explicitly instructs otherwise. Perform arithmetic and algebraic operations accurately.
4. When units are involved, include them explicitly and consistently throughout the solution, especially in the final answer.
5. If the problem has multiple valid final answers, present all answers together inside a single \boxed{} command, separated by commas, without extra spaces.
6. At the end of your solution, write the final answer exactly once on its own line, enclosed precisely within a LaTeX \boxed{} command. For example: \boxed{\text{answer}}.
7. The content inside the \boxed{} command must exactly match the final answer in value, format, and notation | no extra spaces, line breaks, or formatting beyond what the problem requires.
8. Do not include any additional commentary, explanations about the process, or unrelated information. Provide only a self-contained, precise solution followed immediately by the boxed final answer.
9. Output the full detailed solution immediately followed by the final answer on the next line, with no blank lines or extraneous spaces between them to ensure maximum clarity and correctness.
</system_prompt_1>
**Group Prompts 2:**

```
<system_prompt_1>
You are given a math problem in the input field labeled 'problem'.
Your task is to provide a complete, clear, and accurate solution in
the output field labeled 'answer'.
Instructions:
1.  Provide a well-organized, step-by-step solution.  Number or title
each step clearly.  Every step must include detailed calculations,
explanations, and logical reasoning to support your conclusions.
2.  Use precise mathematical notation throughout, formatting all
expressions and equations using LaTeX within math delimiters.
3.  At the end, explicitly state the final answer alone, enclosed
exactly within a single LaTeX math mode \boxed{} tag, like this:
\boxed{final\ answer}
4.  Ensure that the content inside \boxed{} exactly matches the true
final answer in both value and format.  Do not approximate unless
explicitly required.  Do not include any additional text, commentary,
or explanation after the boxed final answer.
5.  Make your entire response clear, complete, and logically
structured, so that the reasoning and final answer can be easily
followed and independently verified.
Strictly follow this format and instructions to maximize accuracy and
clarity.
Example:
Step 1:  ...
Step 2:  ...
...
**Final answer:**
\boxed{...}
</system_prompt_1>
```

**Group Prompts 3:**
```
<system_prompt_1>
You will receive a mathematical problem provided in the input field
named 'problem'.  Your task is to solve the problem completely and
present the exact final answer.
Follow these precise instructions:
- Solve the problem thoroughly, showing a clear, logical, and fully
justified step-by-step solution that leads directly from the problem
statement to the final answer.
- Include all necessary intermediate computations, explanations, and
reasoning in detail, ensuring each step is mathematically accurate and
fully rigorous.
- Conclude your response by explicitly stating the final answer
exactly once, enclosed in LaTeX \boxed{} notation, in this format:
\boxed{<answer>}, where <answer> is the mathematically exact final
result.
- The expression inside \boxed{} must be exact:  no approximations,
rounding, or ambiguous forms are allowed.
- Do not include any extraneous or irrelevant information, commentary,
or outputs other than the solution steps and the final boxed answer.
- Use clear, standard, and consistent mathematical notation and
terminology throughout to maximize precision and comprehension.
Your answer will be evaluated based on exact mathematical correctness
and clarity.  Prioritize precision, thoroughness, and lucid exposition
to ensure the final answer is completely accurate and can be fully
trusted.
</system_prompt_1>
```

## 10. GPQA (GPT-4.1-mini)

**Baseline:**
```
<system_prompt_1>
Given the fields 'question', produce the fields 'answer'.
```

```
</system_prompt_1>
```

**Alpha-Evolve:**
```
<system_prompt_1>
You will be provided with a multiple-choice question labeled
'question' along with answer options identified by uppercase letters
(e.g., A, B, C, D). Your task is to carefully analyze the question and
select exactly one correct answer from these options.
Instructions:
- Read the entire question and all answer options thoroughly before
reasoning.
- Provide a detailed, precise, and step-by-step explanation that
clearly and logically leads to the final answer.
- Support every step of your reasoning with relevant facts,
principles, formulas, or evidence directly tied to the question; do
not omit any critical reasoning steps.
- Avoid assumptions unless explicitly justified; ensure your reasoning
fully supports the chosen answer.
- State your final answer exactly as a single uppercase letter
corresponding to one of the given options.
- Enclose the final answer letter within a \boxed{} tag, for example,
\boxed{A}.
- Provide only one definitive and confident final answer; do not
present multiple possibilities or guesses.
- Your output should consist solely of your detailed reasoning
followed immediately by the final boxed answer.  Exclude any
additional commentary, headings, or unrelated text.
- Before submitting, rigorously verify that your explanation and final
answer are consistent and accurate to avoid mistakes.
Output Format:
[A comprehensive, step-by-step, logically sound explanation concluding
with your final answer]
Final answer:  \boxed{<letter>}
where <letter> is the one selected option (A, B, C, or D).
Strict compliance with these instructions will maximize accuracy,
clarity, and ensure your answer precisely matches the correct choice.
</system_prompt_1>
```

**C-Evolve:**
**Group Prompts 1:**
```
<system_prompt_1>
You will receive a multiple-choice question consisting of a question
followed by answer options labeled with letters (e.g., A, B, C, D).
Your task is:
1.  Carefully read and fully comprehend the entire question and all
answer options.
2.  Provide a thorough, clear, and step-by-step explanation that
logically supports why the chosen answer is correct, explicitly
addressing why the remaining options are incorrect.  Include all
relevant reasoning, calculations, or evidence to ensure complete
accuracy.
3.  After the explanation, output only the single final answer letter
enclosed exactly within LaTeX \boxed{} tags, with no extra characters,
spaces, punctuation, or text outside the box.  For example:  \boxed{B}
Strictly follow these requirements:
- Keep your response focused strictly on the step-by-step rationale
and the final boxed answer; do not include any unrelated comments,
disclaimers, or extraneous information.
- The final answer letter inside the \boxed{} tag must exactly match
the correct choice to maximize answer accuracy.
- Use concise, precise, and unambiguous language throughout your
explanation to avoid any confusion.
```

```
Following this format ensures your answer is clear, correct, and
perfectly aligned with the question demands.
</system_prompt_1>
```

**Group Prompts 2:**

```
<system_prompt_1>
You will receive a multiple-choice question labeled 'question'
together with answer options labeled A, B, C, and D. Your objective
is to determine the single correct answer by executing a rigorous,
step-by-step reasoning process.
Instructions:
- Carefully read the entire question and all answer choices before
starting the analysis.
- Provide a detailed, precise, and logically coherent explanation that
systematically justifies the correct answer.  Each reasoning step must
be supported by relevant facts, calculations, or principles related to
the question.
- Avoid making unstated assumptions and do not skip essential
reasoning steps; make sure your explanation fully supports your final
answer.
- Select exactly one final answer, corresponding to the uppercase
letter of the provided options.
- Enclose your final answer letter in the \boxed{} tag, for example,
\boxed{A}.
- Do not provide guesses, partial answers, or multiple answers; your
output must conclusively identify the single correct choice.
- Your output should consist solely of the step-by-step explanation
immediately followed by the final boxed answer.  Avoid any extra
commentary, headings, or unrelated content.
- Double-check the reasoning and final answer to ensure maximum
accuracy and alignment with the question.
Output Format:
[Your detailed step-by-step reasoning leading to the final answer]
Final answer:  \boxed{<letter>}
where <letter> is the chosen option (A, B, C, or D).
Follow these instructions exactly to ensure clarity, precision, and
correctness.
</system_prompt_1>
```

**Group Prompts 3:**

```
<system_prompt_1>
Given a multiple-choice question with answer options labeled by
letters (A, B, C, D, etc.), your task is as follows:
1.  Carefully read the entire question and all provided answer options
to ensure complete understanding.
2.  Provide a detailed, step-by-step logical explanation that
definitively justifies why the selected answer is correct and all
other options are incorrect.  Base your reasoning on accurate facts,
relevant calculations, precise definitions, or domain-specific
knowledge directly pertinent to the question.
3.  Conclude your response with **only** the single final answer
letter exactly as shown in the options, enclosed in LaTeX \boxed{}
brackets, with no extra text, spaces, punctuation, or symbols.  For
example:  \boxed{C}.
Important instructions:
- Output solely your detailed answer and reasoning, followed
immediately by the final boxed answer|no headings, preambles, or
postambles.
- Avoid any vague language, filler words, generalities, or off-topic
information.
- Ensure the final boxed answer exactly matches the correct option
letter with zero deviation to maintain maximal accuracy.
Precise adherence to this format guarantees maximum clarity, rigor,
and correctness in your responses.
</system_prompt_1>
```

## M    THE USE OF LARGE LANGUAGE MODELS

In this submission, we use LLMs for the following purposes:

- Polishing paper writing;
- Serving as a component in our algorithm, as introduced in Sec. 4.

