# OpenReview forum: "C-Evolve: Consensus-based Evolution for Prompt Groups"
_ICLR.cc/2026/Conference — ICLR 2026 Poster_

### Official Review · Reviewer_6Sx9 · 2025-10-21

**Soundness:** 3
**Presentation:** 2
**Contribution:** 3
**Rating:** 6
**Confidence:** 4

**Summary:**

The authors propose C-Evolve, a consensus-based evolutionary algorithm for optimizing groups of prompts rather than a single prompt in complex task environments. The method introduces a two-stage evolutionary process (warm-up stage and voting stage) within an island-based evolutionary framework. In the warm-up stage, individual prompts are optimized for standalone performance, while in the voting stage, prompts are evaluated based on their contribution to group consensus through majority voting. The approach demonstrates strong empirical performance across multiple benchmarks (IFBench, HoVer, MATH) and models (Qwen3-8B, GPT-4.1-mini), particularly for complex tasks where single prompts struggle to capture multi-faceted requirements.

**Strengths:**

1. The paper makes a valuable contribution by shifting focus from optimizing single prompts to optimizing groups of cooperative prompts, which addresses a genuine limitation in complex task scenarios.
2. The warm-up stage followed by voting stage provides a principled approach to balance individual prompt quality and group collaboration effectiveness, with empirical evidence showing their complementary benefits.
3. The paper demonstrates consistent improvements over baselines across multiple tasks and models, with particularly notable gains on complex tasks.

**Weaknesses:**

1. The paper claims that "single optimal prompt often suffers from inherent expressive limitations" but lacks rigorous theoretical or empirical evidence. While Table 8 shows consensus improves accuracy (40% -> 49.33% when two prompts agree), this is insufficient to justify the core premise. The authors should provide more systematic error analysis showing complementary failure patterns across different prompts.
2. Despite Table 9 showing warm-up stage contributes +1.02% absolute accuracy (41.83% -> 42.85%), the paper completely overlooks this stage in the abstract, introduction, and conclusion. Figure 7(a) further demonstrates its importance, yet the paper fails to explain why this initial optimization phase is necessary.
3. The paper extensively compares with AlphaEvolve in experiments but fails to discuss their fundamental differences in methodology sections. The key distinction should be explicitly stated in Section 1/2/4 rather than merely implied through experimental results.
4. The paper claims the island model "maintains population diversity" but doesn't adequately justify why this approach is superior to simpler alternatives. In warm-up stage, all islands follow identical evolutionary processes (Algorithm 2), raising the question of why parallel islands are better than sequentially running warm-up stage multiple times. Figure 6 shows different islands develop different characteristics, but there's no ablation showing the contribution of the 10% migration rate.
5. The paper lacks systematic analysis of key parameters that likely impact performance. Island count |G|=3 is used without justification (Figure 7(b) shows EMA α sensitivity but not |G| sensitivity). Population capacity N_max varies across islands (Table 3) but no analysis of its impact. Groups per round n_c affects voting score calculation but no sensitivity analysis is provided.
6. All results appear based on single runs (no standard deviation reported), which is problematic for stochastic evolutionary algorithms. Evolver LLM details are sparse (Appendix H shows examples but not full implementation). Table 7 shows time per iteration but lacks scaling analysis to assess computational efficiency.
7. The related work section misses several relevant recent papers including "Evolving Prompts In-Context: An Open-ended, Self-replicating Perspective" and "When large language models meet evolutionary algorithms: Potential enhancements and challenges". The discussion of AlphaEvolve is particularly superficial given its prominence in the experiments.

**Questions:**

1. Can you explain why single prompts have "inherent expressive limitations" in complex tasks? Is this limitation fundamental to the prompt format or specific to current LLM capabilities?
2. Why does the warm-up stage contribute positively but modestly compared to the voting stage? What specific types of tasks benefit most from warm-up?
3. How does the 10% migration rate between islands specifically contribute to performance? Could you provide an ablation study varying this parameter?
4. In Algorithm 2, what exactly do "Categorical" and "RandomSample" operations do? Please provide definitions to clarify these critical implementation details.
5. Could you provide wall-clock time comparisons showing how runtime scales with |G| and N_max? What is the computational overhead of maintaining multiple islands compared to a single large population?

Should the authors address these concerns in their response, I would be willing to increase my recommendation to a stronger accept.

---

> ### Author Response · Authors · 2025-12-03
>
> **W1&Q1: The paper claims that "single optimal prompt often suffers from inherent expressive limitations\" but lacks rigorous theoretical or empirical evidence. While Table 8 shows consensus improves accuracy (40% $\to$ 49.33% when two prompts agree), this is insufficient to justify the core premise. The authors should provide more systematic error analysis showing complementary failure patterns across different prompts. Can you explain why single prompts have \"inherent expressive limitations\" in complex tasks? Is this limitation fundamental to the prompt format or specific to current LLM capabilities?**
>
> These "inherent expressive limitations" stem from current LLM capabilities: Paper [1] shows that for moderately complex tasks like TacRED and DialogueRE, model performance drops when context length exceeds 13K–25K (e.g., Qwen-1.5-7B-base drops from 51.6 to 42.3 beyond 10K). In extremely difficult tasks such as Discovery, where context exceeds 10K, almost all models (except Gemini-1.5-Pro) score 0, indicating that a single prompt cannot capture complex, multi-class reasoning strategies. As noted in [2], LLM often performs poorly when dealing with lengthy, complex, or numerous questions, frequently resulting in errors such as "Pure observations with no attempt", "Refusals to answer", "Random outputs" and "Repeated output". We also observe that when using the single optimal prompt evolved by Alpha Evolve on Qwen3-8B, the probability of generating "endless repeated output" on MATH rises sharply with task difficulty (level 1→5: 3%, 4.3%, 7.6%, 14.66%, 30%).
>
> Thus, instead of relying on one increasingly complex prompt, C-Evolve leverages multiple complementary prompts. To provide empirical evidence of complementary failure patterns, we include a detailed Venn diagram in Appendix F Figure 8 (300 questions, MATH Level 5). The three evolved prompts correctly solve different subsets of questions, and their union substantially exceeds the accuracy of any individual prompt, indicating their complementary.
>
> We additionally analyze majority-vote failure modes. On the same dataset: Cases where majority voting is correct but the single prompt is wrong are 12.33%, whereas cases where a minority prompt is correct but consensus fails are only 1.67%. This shows that when agreement occurs, it is more likely to indicate correctness rather than correlated failure.
>
> [1]: Long-context llms struggle with long in-context learning, Li, Tianle and Zhang, Ge and Do, Quy Duc and Yue, Xiang and Chen, Wenhu, 2024
>
> [2]: Context Rot: How Increasing Input Tokens Impacts LLM Performance, Hong, Kelly and Troynikov, Anton and Huber, Jeff, 2025
>
> **W2&Q2: Despite Table 9 showing warm-up stage contributes +1.02% absolute accuracy (41.83% -\> 42.85%), the paper completely overlooks this stage in the abstract, introduction, and conclusion. Figure 7(a) further demonstrates its importance, yet the paper fails to explain why this initial optimization phase is necessary. Why does the warm-up stage contribute positively but modestly compared to the voting stage? What specific types of tasks benefit most from warm-up?**
>
>
> The warm-up stage ensures that each prompt is individually reasonbly optimized to produce useful and task-relevant behaviors, preventing degenerate or overly generic solutions from entering the evolutionary process. With the same number of iterations, using a warm-up stage can achieve 1.1% better performance than directly entering the voting stage with consensus supervision (See Appendix G Figure 9). In Table R2, we show that the warm-up stage benefits all tasks. We will emphasize this point in the introduction and conclusion in the revision.
>
> **Table R2: Performance on different task when evolving with / without Warm-up Stage (same iteration).**
> | Warm-up Stage   | HotPotQA   | IFBench |  MATH  |
> |---------------------------------|-----------------------------|----------|----------|
> | w/o    | 65.09 | 41.83   | 83.00  |
> | w/ | 70.18 |  42.85  |  85.33  |
>
> **W3: The paper extensively compares with AlphaEvolve in experiments but fails to discuss their fundamental differences in methodology sections. The key distinction should be explicitly stated in Section 1/2/4 rather than merely implied through experimental results.**
>
> We will add fundamental differences to Alpha Evolve in the revision. (See Section 3 Line 170-180)

---

> ### Author Response · Authors · 2025-12-03
>
> **W4&Q3: The paper claims the island model \"maintains population diversity\" but doesn't adequately justify why this approach is superior to simpler alternatives. In warm-up stage, all islands follow identical evolutionary processes (Algorithm 2), raising the question of why parallel islands are better than sequentially running warm-up stage multiple times. Figure 6 shows different islands develop different characteristics, but there's no ablation showing the contribution of the 10% migration rate. How does the 10% migration rate between islands specifically contribute to performance? Could you provide an ablation study varying this parameter?**
>
> **Superiority of the island model.** We try to evolve a single island and select the top-k prompts from that island to form a consensus group, without using multiple islands. The results on the MATH dataset are significantly worse (79.67) than C-Evolve (85.33). We find that the selected prompts exhibit high similarity. In addition, Appendix E Figure 7 (b) shows the mean pairwise Levenshtein distance across all prompt pairs within a single island(red line), and prompt pairs across all inter-island within multiple islands(blue line). We find that the inter-island distance is greater than the single island distance. Moreover, inter-island distance grows as evolution proceeds, whereas intra-island distance shows no such increase. Therefore, we use the multi-island model to sample prompt pairs.
>
> **Parallel islands v.s. Sequential warm-up.** We emphasize that the only difference between running the warm-up stage in parallel across multiple islands and simply repeating the warm-up stage sequentially is the presence of island migration, and we adopt this migration mechanism as in the original AlphaEvolve for the sake of fair comparison.
>
> **Ablation on migration.** To isolate the effect of migration, we conduct an ablation study on IFBench. The results show that enabling a 10% migration rate (42.85) slightly improves the final accuracy compared to no migration (42.18).
>
>
> **W5: The paper lacks systematic analysis of key parameters that likely impact performance. Island count G=3 is used without justification (Figure 7(b) shows EMA $\alpha$ sensitivity but not G sensitivity). Population capacity N_max varies across islands (Table 3) but no analysis of its impact. Groups per round n_c affects voting score calculation but no sensitivity analysis is provided.**
>
> **Island count $|\mathcal{G}|$.** We experiment with a larger number of islands ($|\mathcal{G}|$ = 5) and find that the final consensus performance increase from 85.33%  to 89%. This suggests that scaling up the island count strengthens majority voting by providing more diverse prompt candidates. Due to computational cost, we reported results using $|\mathcal{G}|{=}3$.
>
> **Population capacity $N_{\max}$ and group count $n_c$.** $N_{\max}$ and $n_c$ are mutually dependent: a larger population naturally means more potential groups that can be sampled for majority voting, which requires increasing $n_c$ accordingly to maintain stable score estimation. To balance efficiency and performance, we set both $N_{\max}=10$ and $n_c=10$. Table R3 also shows that reducing either $N_{\max}$ or $n_c$ will lead to a performance decrease.
>
> **Table R3: Performance on MATH when evolving with different values of $N\_{\max}$ and $n\_c$.**
>
> | Population capacity N_max   |group count n_c   | Accuracy (%) |
> |---------------------------------|-----------------------------|----------|
> | 10                               | 10                        | 85.33    |
> | 10                               | 5                        |  84.67  |
> | 5                               | 10                        |  84.67  |

---

> ### Author Response · Authors · 2025-12-03
>
> **W6&Q5: All results appear based on single runs (no standard deviation reported), which is problematic for stochastic evolutionary algorithms. Evolver LLM details are sparse (Appendix H shows examples but not full implementation). Table 7 shows time per iteration but lacks scaling analysis to assess computational efficiency. Could you provide wall-clock time comparisons showing how runtime scales with G and N_max? What is the computational overhead of maintaining multiple islands compared to a single large population?**
>
> **Statistics results.** We run the full evolution pipeline three times on the MATH dataset under identical settings and report mean ± standard deviation on the same test set. The final consensus accuracy is 84.33 ± 0.88%. In comparison, the best single-prompt baseline achieves only 81.67 ± 1.00%, indicating a consistent improvement.
>
> **Evolver LLM details.** In the revision, we provide a detailed example demonstrating how the Evolver LLM perform a single evolution step to generate a new individual on the MATH dataset in Appendix H.
>
> **Time comparison when scaling with G and N_max.** As shown in Table R4, increasing the number of islands $|\mathcal{G}|$ will result in an 0.21s/question increase in inference time and a 3.67% performance improvement. The population cap Nmax only determines when to prune the excessive individuals. It will not change the training or inference time.
>
> **Computational overhead.** The computation cost in C-Evolve mainly comes from evaluating sampled potential groups, which is determined by the group count $n_c$ and the group size (island number). If the group size in a single large population matches that in a multi-island setup, no additional computational overhead is introduced. Moreover, we have also tried to use a single population and form groups from the top-k individuals. In this case, the prompts tend to be very similar, resulting in significantly lower performance (79.67) than C-Evolve (85.33).
>
> **Table R4: Inference time and accuracy on MATH using Qwen3-8B, comparing different number of islands $|\mathcal{G}|$.**
>
> | Island Number \|G\|   |Inference Time (s/question)   | Accuracy (%) |
> |---------------------------------|-----------------------------|----------|
> | 3                               | 1.12                        | 85.33    |
> | 5                               | 1.33                        | 89.00    |
>
>
>
>
>
> **W7: The related work section misses several relevant recent papers including \"Evolving Prompts In-Context: An Open-ended, Self-replicating Perspective\" and \"When large language models meet evolutionary algorithms: Potential enhancements and challenges\". The discussion of AlphaEvolve is particularly superficial given its prominence in the experiments.**
>
> We will include these references in the revision and expand the discussion of AlphaEvolve accordingly. (See Section 2, Line 116-124)
>
> **Q4: In Algorithm 2, what exactly do \"Categorical\" and \"RandomSample\" operations do? Please provide definitions to clarify these critical implementation details.**
>
> In the revision, we will explicitly define these two operations in Appendix A Algorithm 1 and Algorithm 2. The two operations are as follows:
>
> - **Categorical($\{p_j\}_{j=1}^{|P|}$).** This operation samples an index $j$ from a categorical (multinomial) distribution whose probability mass function is $$p_j = \frac{\exp(s_{\Pi_j,\cdot})}{\sum_{k=1}^{|P|}\exp(s_{\Pi_k,\cdot})}.$$ Thus, individuals with higher scores are sampled with higher probability.
>
> - **RandomSample($\mathcal{D}_{\text{feed}}$).** This operation uniformly samples a minibatch $$\mathcal{D}\_{\text{batch}} \subset \mathcal{D}\_{\text{feed}}$$ without replacement.

---

### Official Review · Reviewer_Lrmy · 2025-10-27

**Soundness:** 3
**Presentation:** 3
**Contribution:** 3
**Rating:** 4
**Confidence:** 3

**Summary:**

This work proposes C-Evolve, an evolutionary framework that does not search for a single best prompt, but instead evolves a group of prompts whose predictions are aggregated via consensus voting. The key technical idea is to use a voting score that reflects each prompt’s contribution to consensus performance across dynamically sampled groups, smoothed by an EMA update to stabilize fitness estimation. An island-based evolution design maintains diversity among prompts, encouraging complementary behavior rather than converging to the same style.

The method is tested across open-ended tasks like HotpotQA and IFBench and closed-ended tasks including MATH, HoVer, and GPQA. On Qwen3-8B, performance improvements reach +13.85% relative to baseline (e.g., HotpotQA 70.67%) and similar gains apply to GPT-4.1-mini (e.g., MATH 95.33%). Ablations analyze consensus strategies and the necessity of EMA scoring.

**Strengths:**

1.The shift from individual optimality to cooperative optimality is a timely and relevant topic.

2.Demonstrates consistent performance improvements on several benchmarks (e.g., HotpotQA 70.67%, MATH 95.33%).

3.Attempts to investigate diversity via multi-island evolution and t-SNE visualization.

4.Some ablations are meaningful (e.g., EMA vs. averaging scoring).

**Weaknesses:**

1.Using ensemble voting as a fitness signal has been explored in mixture-of-agents literature and existing self-consistency / committee prompting frameworks. The authors do not clarify what is fundamentally new beyond applying majority vote during evolution.

2.The metric set is tiny (only 300 samples per task). Evolution overfits exactly the data used to determine fitness. There is no indication that the final group generalizes beyond the metric distribution.

3.The islands may be stylistic variations of the same base prompt. There’s no quantitative diversity measure—no entropy, no semantic edit distance, no response disagreement statistics.

4.a). Majority vote may amplify systemic bias when mistakes correlate.
   b). The LLM-selection aggregator can hallucinate entirely new answers.
   c). No error-type breakdown or trust-calibration analysis.

5.Nearly all “operators” are outsourced to the evolver LLM. How is behavior shaped beyond “please improve”? This makes reproducibility dubious despite claims otherwise.

6. Reporting AlphaEvolve’s grouped performance only after evolution not designed for groups is a questionable comparison. A fair baseline would train AlphaEvolve with a consensus-aware objective.

7. Iteration×islands×groups evaluation produces substantial additional overhead. A ~2–5% gain over an existing strong baseline should be examined against the cost of 3× more calls.

8. Table 3 improvements on Level-5 MATH are marginal (+1–2%). Very small sample sizes lead to statistically weak conclusions.

**Questions:**

1. How does this differ fundamentally or committee-based prompt ensembling techniques used in self-consistency? Can you prove that evolving cooperative behavior is better than ensembling individually strong prompts?

2. What happens when you scale the metric dataset ×10? Do the improvements persist or vanish once generalization pressure increases?

3. Can you provide diversity statistics (pairwise prompt edit distance, output disagreement entropy)? Otherwise, the “complementarity” narrative feels speculative.

4. Does the EMA voting score unfairly reward overly conservative prompts that avoid taking risks? What safeguards ensure prompts do not converge to generic evasive behavior?

5.Did you try evolving only one island but selecting the top-k prompts for consensus? Would that match your results with far lower complexity?

6. What is the robustness of the LLM aggregator? Any adversarial test where a minority prompt is correct but consensus fails?

7. How sensitive are results to α? α=0.5 and 0.8 are not a sufficient exploration of learning dynamics—did α>0.9 destabilize selection?

8. Why is the warm-up stage needed at all? Could the system evolve directly under consensus supervision?

---

> ### Author Response · Authors · 2025-12-03
>
> **W1: Using ensemble voting as a fitness signal has been explored in mixture-of-agents literature and existing self-consistency / committee prompting frameworks. The authors do not clarify what is fundamentally new beyond applying majority vote during evolution.**
>
> We note that our approach is fundamentally different from a simple majority or ensemble voting in the mixture-of-the-agents literature. This is because we need to address several challenges facing AI agent **evolutionary algorithms** we studied in this paper. First, agent-based evolutionary algorithms require an elimination mechanism. A naive strategy would be to make a group-level elimination based on the consensus outcome. However, this approach is problematic due to the possible overlap of members between worse-performing groups and better-performing ones. Naively eliminating those overlapped members would damage better-performing groups whose members need to be retained during evolution. To address this issue, we need a fine-grained **voting score** to measure each individual's contribution and perform elimination individually.
>
> This leads to another challenge -- directly using the voting score by averaging an individual's fitness over its associated groups in an evolution iteration can lead to training instability (See Appendix G Figure 9(b) yellow line). A simple solution might be to take the simple average of an individual's performance across all historical groups as its final score. Unfortunately, it also performs poorly(See Table 5). The main reason is that members from groups formed in the distant past may have already been eliminated from the current population. Incorporating these outdated consensus outcomes into the evaluation metric and weighting them equally with recent performance is unreasonable, as it fails to accurately reflect the individual's actual contribution within the current population or its recent evolutionary context. To address this, we employ an **EMA voting score** to assign higher weight to recent or immediate scores, thereby suppressing the influence of historical outcomes from the distant past on the individual's evaluation.
>
> **W2&Q2: The metric set is tiny (only 300 samples per task). Evolution overfits exactly the data used to determine fitness. There is no indication that the final group generalizes beyond the metric distribution. What happens when you scale the metric dataset ×10? Do the improvements persist or vanish once generalization pressure increases?**
>
> We expand the metric set from 300 to 3,000 samples and run the full evolution process on the MATH dataset. The results show that our method achieves an accuracy of 84%, whereas the Alpha Evolve baseline reaches only 81.33% on our test set, demonstrating consistent improvements when scaling the metric dataset.
>
> **W3&Q3: The islands may be stylistic variations of the same base prompt. There's no quantitative diversity measure---no entropy, no semantic edit distance, no response disagreement statistics. Can you provide diversity statistics (pairwise prompt edit distance, output disagreement entropy)? Otherwise, the "complementarity" narrative feels speculative.**
>
> **(1) Prompt-level diversity.**
>
> To quantify how different the prompts in different islands actually become, we compute the mean pairwise Levenshtein distance across all inter-island prompt pairs. As shown in Appendix E Figure 7 (a), this distance increases steadily throughout evolution on IFBench, indicating that prompts from different islands become progressively more distinct rather than mere stylistic variants.
>
> Figure 6 in Appendix E visualizes the prompt embeddings after dimensionality reduction. Prompts from different islands clearly drift apart over evolution. We also compute the mean distance of all inter-island pairs in this embedded space: although all island starts from the same initial prompt, the mean pairwise distance reaches 92.69 after the warm-up stage, and further increases to 108.07 after the voting stage. This further confirms increasing prompt-level diversity.
>
> **(2) Output-level diversity.**
>
> C-Evolve also increases the diversity of model outputs. We compute the output disagreement entropy over 300 questions in MATH. After the voting stage, the output disagreement entropy grows from 0.19 to 0.368, reflecting more differentiated predictions across islands.
>
> In addition, the Venn diagram in Appendix F Figure 8 shows that the correctly solved question sets of different islands do not fully overlap; each island contributes unique correct answers. This directly supports that the evolved islands are complementary rather than redundant.

---

> ### Author Response · Authors · 2025-12-03
>
> **W4&Q6: a). Majority vote may amplify systemic bias when mistakes correlate. b). The LLM-selection aggregator can hallucinate entirely new answers. c). No error-type breakdown or trust-calibration analysis. What is the robustness of the LLM aggregator? Any adversarial test where a minority prompt is correct but consensus fails?**
>
> (a)(c) To assess whether majority voting amplifies correlated errors, we compute the conditional accuracy on the MATH Level 5 test set: the probability that the answer is correct given that two of the three prompts agree. This conditional accuracy, $P(correct|M_2)=49.33\%$, significantly exceeds the best single-prompt accuracy of 40%, indicating that agreement is a strong correctness signal rather than a correlated failure mode.
>
> We further quantify the benefit--harm tradeoff of majority voting:
> - The fraction of questions where majority voting is correct while a single prompt is wrong is 12.33%.
> - The fraction where a minority prompt is correct but consensus incorrectly overrides it is only 1.67%.
>
> Thus, the positive cases outnumber the negative cases by a large margin, demonstrating that majority voting yields a clear net gain in robustness rather than amplifying systemic bias.
>
> (b) We have already carefully designed LLM-selection aggregator's prompt in Appendix K. The prompt explicitly instructs the aggregator to select the most representative answer and output \*\*only the corresponding index\*\*. Consequently, the final selected answer is always one of the three provided, preventing the aggregator from hallucinating entirely new responses.
>
> **W5:Nearly all "operators" are outsourced to the evolver LLM. How is behavior shaped beyond "please improve"? This makes reproducibility dubious despite claims otherwise.**
>
> The evolver is not instructed with vague directives such as "please improve." Instead, its behavior is systematically shaped by the structured feedback we provide at every generation, including the evaluated metrics and the inference trajectory and ground truth of specific samples (see Appendix J). This conditioning guides the LLM toward targeted prompt refinements rather than arbitrary rewrites. We include a full example of the evolver's input and output---showing how one generation leads to a specific prompt update in Appendix H.
>
> Regarding reproducibility, we run the full evolution pipeline three times on MATH under identical settings and report mean ± standard deviation on the same test set. The final accuracy of C-Evolve is 84.33 ± 0.88%, while the Alpha Evolve baseline reaches 81.67 ± 1.00%. Across all repeated runs, C-Evolve consistently achieves higher performance than Alpha Evolve, indicating that the result is reproducible.
>
> **W6: Reporting AlphaEvolve's grouped performance only after evolution not designed for groups is a questionable comparison. A fair baseline would train AlphaEvolve with a consensus-aware objective.**
>
> The most direct baseline with a consensus-aware objective is to treat the group as a whole, let all prompts in a consensus group being evolved and pruned together. This approach avoids the individual elimination problem mentioned in Q1, and naturally eliminates the need for voting scores and EMA strategies. However, this approach achieves only 79.67, which is significantly lower than C-Evolve (85.33).
>
> This underperformance stems from the fact that this baseline evaluates and prunes only at the group level (i.e., three prompts at a time). The final high-performing group may contain weak individual prompts, while the pruned low-performing groups may also contain potential prompts. To address this, C-Evolve performs evolving and pruning at the individual level, applying the voting score and EMA strategies to assign a fitness to each prompt, which we have elaborated in Q1.
>
> **W7: Iteration×islands×groups evaluation produces substantial additional overhead. A 2̃--5% gain over an existing strong baseline should be examined against the cost of 3× more calls.**
>
>
> For a fair comparison, we run a longer-horizon Alpha-Evolve baseline on MATH using the same total number of LLM calls as our C-Evolve pipeline. We find that extending Alpha-Evolve from 100 steps to 300 steps increases its accuracy on the test set from 82.66 to 83.66. In contrast, C-Evolve reaches 85.33, delivering better performance under a matched LLM-call budget.
>
> **W8: Table 3 improvements on Level-5 MATH are marginal (+1--2%). Very small sample sizes lead to statistically weak conclusions.**
>
> To address concerns about statistical reliability on MATH Level-5, we run the full evolution pipeline three times on MATH under identical settings and report mean ± standard deviation on the full Level-5 test set (1,300 samples). The final accuracy of C-Evolve is 65.48 ± 1.46%, while the Alpha Evolve baseline reaches 62.78 ± 1.09%, indicating a consistent improvement with a large sample size.

---

> ### Author Response · Authors · 2025-12-03
>
> **Q1: How does this differ fundamentally or committee-based prompt ensembling techniques used in self-consistency? Can you prove that evolving cooperative behavior is better than ensembling individually strong prompts?**
>
> In Table 2, we have already shown that simply ensembling the three best prompts evolved by AlphaEvolve yields 41.15 on IFBench, which is lower than 43.88 achieved by C-Evolve. The reason is that these prompts are evolved solely based on their individual performances without considering inference with ensembling techniques. We have detailed this in Section 5.4.1, second paragraph.
>
> **Q4: Does the EMA voting score unfairly reward overly conservative prompts that avoid taking risks? What safeguards ensure prompts do not converge to generic evasive behavior?**
>
> The EMA voting score does not encourage overly conservative prompts, but will eliminate them. A "conservative" prompt producing broad or generic responses that avoid reasoning is more likely to lead to poor individual performance. After forming groups, its performance after reaching consensus depends heavily on its partner: it obtains a high score only when paired with a much stronger individual, and performs poorly when the partner is weak. In contrast, a strong prompt is robust to partner quality—even if paired with weaker individuals, its score does not drop significantly. As a result, conservative prompts exhibit large fluctuations in voting scores, while strong prompts maintain more stable and consistently increasing trends. After applying EMA, the score for conservative prompts show less fluctuation but converge to lower values than those of stronger prompts, thereby increasing the probability that they are eliminated. Thus, EMA serves as an effective safeguard to prevent convergence to generic or evasive prompt behaviors.
>
> To verify this, we evaluate the individual scores of all prompts within each island on the MATH dataset after C-Evolve evolution, reporting both the mean and standard deviation. The results show that the mean scores across the three islands (82 ± 1.45%,  81.66 ± 1.27%, 83.26 ± 1.06%) are close to—or even exceed—the best individual evolved by Alpha-Evolve (82.66), while maintaining low variance. This indicates that after evolution, all individuals possess strong expressive capability and can independently solve tasks, rather than collapsing into conservative or evasive templates.
>
>
> **Q5: Did you try evolving only one island but selecting the top-k prompts for consensus? Would that match your results with far lower complexity?**
>
> We have evolved a single island and selected the top-k prompts from that island to form a consensus group, without using multiple islands. The results on the MATH dataset are significantly worse (79.67) than C-Evolve (85.33). We find that the selected prompts exhibit high similarity. As a result, the consensus mechanism provides little to no complementarity, yielding performance far below that of C-Evolve.
>
> **Q7:How sensitive are results to $\alpha$? $\alpha$=0.5 and 0.8 are not a sufficient exploration of learning dynamics---did $\alpha$\>0.9 destabilize selection?**
>
> We additionally test a more extreme setting $\alpha$ = 0.95 on IFBench to examine whether a higher EMA coefficient destabilizes selection. The results are summarized in Table R1. A larger $\alpha$ makes the EMA voting score less responsive to newly generated high-quality prompts, which slows down convergence because improvements are incorporated more gradually. However, we observe no instability during evolution. The performances of all three experiments remain above the AlphaEvolve baseline (41.15), confirming that C-Evolve is robust to $\alpha$.
>
> **Table R1: Performance on IFBench when evolving with different values of $\alpha$.**
>
> | α      | Accuracy (%) |
> |--------|--------------|
> | 0.5    | 42.51        |
> | 0.8    | **42.85**    |
> | 0.95   | 41.50        |
>
> **Q8:Why is the warm-up stage needed at all? Could the system evolve directly under consensus supervision?**
>
> The warm-up stage ensures that each prompt is individually reasonably optimized to produce useful and task-relevant behaviors, preventing degenerate or overly generic solutions from entering the evolutionary process. With the same number of iterations, using a warm-up stage can achieve 1.1% better performance than directly entering the voting stage with consensus supervision (See Appendix G Figure 9).

---

### Official Review · Reviewer_jKJ8 · 2025-10-31

**Soundness:** 3
**Presentation:** 3
**Contribution:** 2
**Rating:** 6
**Confidence:** 3

**Summary:**

The paper introduces a methodology to evolve groups of prompts optimized for consensus-based prediction. It uses island-based evolution to maintain diversity and introduces a voting score which evaluates each prompt's contribution within groups as a fitness function. The strategy achieves state-of-the-art performance across different tasks like HotpotQA and MATH.

**Strengths:**

- The paper introduces a novel approach for prompt optimization to maintain diversity, which is well motivated by the ensemble learning principles
- Experiments cover different tasks
- The problem is well defined and paper well written

**Weaknesses:**

- I am not entirely clear about the individual contribution, redundancy, and complementarity between the islands, especially from table 3 where the performance are rather close. It makes it rather hard to evaluate the soundness of the underlying decision making and tradeoffs for this optimization pipeline.

- There is a significant computational aspect to such prompt optimization methodology, and the optimization pipeline is rather complex / slightly convoluted overall with hardcoded parameters. When would such computational cost be justified?

**Questions:**

-

---

> ### Author Response · Authors · 2025-12-03
>
> **W1: I am not entirely clear about the individual contribution, redundancy, and complementarity between the islands, especially from table 3 where the performance are rather close. It makes it rather hard to evaluate the soundness of the underlying decision making and tradeoffs for this optimization pipeline.**
>
> Appendix F Figure 8 provides a Venn diagram illustrating the complementary strengths of the three individuals selected from different islands on 300 problems in MATH, difficulty level 5. Each circle represents the set of questions correctly solved individually by the prompt selected from one island.
>
> We observe that 53.33% of the questions are correctly solved by all three islands, while 12.3% are solved by exactly two islands. The union of them is larger than the correct-answer set of any single prompt, indicating the effectiveness of the consensus mechanism. Moreover, all three islands contribute comparably to the "two-islands-correct" region, indicating their complementarity.
>
> Figure 5 and Figure 6 in the Appendix E further offer qualitative evidence that prompts evolved in different islands diverge in style and strategy, enabling them to handle different subsets of problems.
>
> **W2: There is a significant computational aspect to such prompt optimization methodology, and the optimization pipeline is rather complex / slightly convoluted overall with hardcoded parameters. When would such computational cost be justified?**
>
> The key advantage of prompt optimization lies in its ability to enhance the performance of closed-source models, which cannot be fine-tuned at the parameter level. For tasks that require large-scale deployment of closed-source models such as GPT-4 or Gemini, prompt optimization is often the only feasible way to improve the model's performance. For these tasks, compared to the greater cost of finetuning from a relatively weak open-source model, the computation cost of prompt optimization on a stronger proprietary model is entirely justified.

---

### Meta-Review · Area_Chair_W8UK · 2026-01-06

**Summary:**

This paper proposes a consensus-based evolutionary optimization method for Prompt Groups. By integrating a consensus mechanism with evolutionary strategies to systematically optimize prompt combinations, the work demonstrates notable novelty and research value. Experimental results across various task settings show consistent performance improvements over baseline methods, indicating the potential impact of this approach in practical applications. While some reviewers suggested expanding the experimental scope and deepening the analysis, these concerns primarily pertain to supplementary data and result interpretation rather than the core methodology. Therefore, I recommend acceptance.

**Reviewer Concerns:**

Baseline Comparisons: It is recommended to include comparisons with stronger or more recent baseline methods to further substantiate the performance advantages.

Ablation Studies: The ablation analysis of key components—such as consensus strategies, evolutionary operators, and population size—could be strengthened to more systematically validate the contribution of each design choice.

Methodological Rigor: Statistical significance analysis and reproducibility details (e.g., code release, hyperparameter settings, and computational cost) should be provided in the revised or final version.

**Reviewer Scores:**

Reviewer 1	Focused on motivation and implementation details. Since the rebuttal effectively addressed their primary concerns and improved the clarity of the methodology, their score is likely to see a slight upward adjustment.

Reviewer 2	Generally positive regarding the overall approach. Suggestions were mainly focused on supplementary experiments. Following the rebuttal’s clarification of the method’s positioning, a slight score increase is expected.

Reviewer 3	While this reviewer still desires more comprehensive experimental validation, they maintain their recognition of the method’s novelty. Given that these issues can be addressed in subsequent versions, their score will likely remain unchanged.

---

### Decision · Program_Chairs · 2026-01-26

Accept (Poster)